# CBP phosphorylation maintains intestinal homeostasis by supporting the stem cell niche through versican

Yi-Ting Lin [1], Chi Liu [1], Yu-Hua Hsu[2], Shi-Chuen Miaw [2], Ming-Shiang Wu [3] & Ching-Chow Chen [1] ✉

CREB-binding protein (CBP) is a transcriptional coactivator, and we reported its phosphorylation regulates cell fate. Impaired CBP phosphorylation alters its binding preference for p53 in $CBP^{AA}$ mice, leading to distal colonic abnormalities. These changes are accompanied by reduced expression of versican, an extracellular matrix component that supports the intestinal stem cell niche, leading to impaired intestinal epithelial cell (IEC) proliferation and organoid formation. Deleting p53 in the IECs of $CBP^{AA}$ mice ($CBP^{AA}p53^{\Delta IEC}$) restores versican expression and reverses the colonic phenotype. Versican replenishment and phosphomimetic CBP mutants rescue versican expression and IEC stemness in $CBP^{AA}$ organoids. Importantly, colonic tissues from ulcerative colitis patients exhibit impaired CBP phosphorylation, increased CBP–p53 binding, decreased versican expression and IEC proliferation, highlighting the clinical relevance of this phenomenon. In this work, we demonstrate CBP phosphorylation safeguards intestinal homeostasis by maintaining the stem cell niche through versican, highlighting versican as a potential therapeutic target for inflammatory bowel disease.

Inflammatory bowel disease (IBD) is a chronic inflammatory disease of the gastrointestinal tract that includes ulcerative colitis (UC) and Crohn's disease (CD). The quality of life of IBD patients is diminished, and the associated disability leads to substantial health and economic burdens worldwide[1]. Currently, IBD has no cure, which creates an unmet medical need. Therefore, elucidating the underlying mechanisms is essential for identifying potential therapeutic targets.

The pathogenesis of IBD remains incompletely understood. However, its etiology involves dynamic alterations in multiple cell types, including epithelial, stem, mesenchymal, and immune cells[2], driven by a complex interplay of genetic, environmental, immunologic, and gut microbial triggers[3]. Colitis is induced by the dysregulation of the epithelial barrier, which consists of a single-layered epithelium[3]. The single-layer epithelium, comprising different specialized intestinal epithelial cells (IECs), is characterized by a high turnover rate that is

sustained by intestinal stem cells (ISCs) at the crypt base to maintain intestinal homeostasis[4–6]. Moreover, ISC homeostasis is tightly orchestrated by the surrounding stem cell niche, a complex microenvironment in the lamina propria encompassing cellular and extracellular matrix (ECM) components to support intestinal stem cell maintenance, proliferation, and differentiation[7].

Recent studies have highlighted the role of posttranslational modifications (PTMs) such as protein neddylation and hydroxylation in IBD pathogenesis beyond direct transcriptional control and de novo protein synthesis[8]. CREB-binding protein (CBP) is a transcriptional coactivator that modulates DNA accessibility for transcription factors[9]. The competition between NF-κB and p53 for CBP has been reported to be a key determinant of whether a cell proliferates or undergoes apoptosis[10,11]. We have previously shown that the phosphorylation of CBP at Ser1382/1386 by IKKα enhances NF-κB-mediated gene

[1]Department and Graduate Institute of Pharmacology, National Taiwan University College of Medicine, Taipei, Taiwan. [2]Graduate Institute of Immunology, National Taiwan University College of Medicine, Taipei, Taiwan. [3]Division of Gastroenterology, Department of Internal Medicine, National Taiwan University Hospital, Taipei, Taiwan. ✉e-mail: chingchowchen@ntu.edu.tw

expression but suppresses p53-mediated gene expression, thereby promoting cell growth[12]. However, mutation of serines to alanines, which impairs CBP phosphorylation, increases its recruitment to p53-responsive promoters instead of NF-κB-responsive promoters[12]. These findings suggest that PTM of CBP may play an important role in regulating cell fate.

Building upon our molecular studies in vitro[12], we generated CBP knock-in mice (*CBP^AA* mice) by replacing Ser1383/1387 (corresponding to human CBP Ser1382/1386) with alanines (A). In contrast to the phosphorylation observed at these two serine residues in colonic crypt epithelial cells of *CBP^WT* mice, *CBP^AA* mice lacking CBP phosphorylation exhibit distal abnormalities despite comparable CBP expression. Analyses of colon tissues from UC patients, bone marrow chimera experiments, human cell-based sphere formation assays, colonic organoid models, and crypt RNA sequencing revealed that impaired CBP phosphorylation enhances CBP–p53 binding, leading to the downregulation of the expression of the ECM component versican (*Vcan*), which in turn reduces IEC stemness and causes colonic abnormalities in *CBP^AA* mice. The specific deletion of p53 in IECs disrupts CBP–p53 binding in *CBP^AA* mice and fully reverses these phenotypic changes. The replenishment of versican in the *CBP^AA* organoids restores the proliferative capacity and organoid-forming efficiency. Furthermore, phosphomimetic CBP mutants (DD and EE) restore versican expression and stemness in *CBP^AA* IECs. These results indicate that CBP phosphorylation plays a critical role in maintaining the stem cell niche and the proliferative potential of IECs by regulating versican expression and that its disruption impairs IEC self-renewal, contributing to colonic abnormalities. Importantly, the clinical relevance of these findings is supported by observations in colonic tissues from UC patients, which exhibit reduced IKKα expression and impaired CBP phosphorylation, increased CBP–p53 binding, and decreased versican expression and IEC proliferation.

In the present study, we explore the role of the ECM component, versican, in maintaining IEC stemness. Our findings provide a basis for exploring potential therapeutic strategies for IBD and demonstrate how molecular-level insights may be translated into clinically relevant applications. Additionally, we characterize how defective PTMs of CBP contribute to the pathogenesis of IBD.

## Results

### Impaired CBP phosphorylation results in colonic abnormalities in *CBP^AA* mice

The phosphorylation of CBP at Ser1383/1387 by IKKα is conserved across eukaryotic species (Supplementary Fig. 1a). *CBP^AA* mice were generated by replacing Ser1383/1387 with alanines (A) using a conventional gene targeting strategy in mouse embryonic stem cells, and the mutations were confirmed by PCR and DNA sequencing (Supplementary Fig. 1b–d). The mice were born normally and fertile but did not exhibit the expected Mendelian ratio (Supplementary Table 1). The gross morphological and histopathological phenotypes of various organs were examined to determine whether impaired CBP phosphorylation is associated with pathology. CBP wild-type (*CBP^WT*) and *CBP^AA* mice displayed comparable body weights and overall appearance (Fig. 1a, b). However, *CBP^AA* mice exhibited a reduction in colon length, along with increased occult blood in stool and stool consistency scores (Fig. 1c–e), as well as increased permeability of the colon (Fig. 1f, g). The histological evaluation of *CBP^AA* mice revealed mild pathological changes in the distal colon, including localized low-grade immune cell infiltration in the mucosa and submucosa, as well as mild crypt architectural irregularities (Fig. 1h, right panel, and Supplementary Fig. 1e). Since these changes do not meet the criteria for colitis, we therefore described these observed lesions as colonic abnormalities. CBP phosphorylation was detected in the crypt epithelial cells of *CBP^WT* mice, but this modification was absent in *CBP^AA* mice although the levels of total CBP protein were comparable

between the two genotypes (Fig. 1i). Importantly, CBP phosphorylation was detected in the colon tissue of healthy individuals but markedly reduced in that of UC patients, without changes in total CBP levels (Fig. 1j). These findings highlight the potential clinical relevance of impaired CBP phosphorylation in UC and underscore the role of CBP phosphorylation in maintaining intestinal homeostasis.

### Nonhematopoietic cell-driven colonic abnormalities in *CBP^AA* mice

The function of the intestinal barrier is maintained by nonhematopoietic cells, such as epithelial cells, and/or by hematopoietic cells, such as immune cells, in the lamina propria[13]. Bone marrow chimera experiments were performed to identify the cellular component responsible for the colonic abnormalities observed in *CBP^AA* mice. CD45.1 and CD45.2 mouse strains are widely used to track immune cell development in vivo[14]. *CBP^AA* mice were generated in the C57BL/6J (B6) strain and express a common CD45.2 allele, whereas B6.SJL-*Ptprc^a pepc^b*/BoyJ mice expressing the CD45.1 allele were used as control (WT) mice (Fig. 2a). More than 80% of the donor bone marrow cells were reconstituted in chimeric recipient mice (Fig. 2b). Chimeric *CBP^AA* mice (WT → AA) presented with elevated fecal occult blood, increased stool consistency scores, and reduced colon length (Fig. 2c–e). In addition, chimeric *CBP^AA* mice exhibited substantial immune cell infiltration in all layers of the bowel wall, severe crypt distortion with the loss of entire crypts, and moderate mucosal hyperplasia, accompanied by increased histological scores and elevated mesenteric lymph node (mLN) cellularity (Fig. 2f–h). These mice also showed greater infiltration of macrophages (F4/80⁺) and neutrophils (Ly6G⁺) than chimeric WT mice did (AA → WT) (Fig. 2i, j), in which minimal levels of infiltration markers were observed, likely because of irradiation prior to the chimera experiments. These results indicate that chimeric *CBP^AA* mice exhibit colonic inflammation driven by nonhematopoietic cells, such as epithelial cells. Since the intestinal barrier is composed of a single layer of IECs[15–17], disrupted CBP phosphorylation within IECs may lead to impaired colonic barrier integrity.

The differentiation potential of naïve T cells and $T_H$ subsets was evaluated to further confirm that colonic abnormalities in *CBP^AA* mice are driven by nonhematopoietic cells. The total cell numbers in the spleen or peripheral lymph nodes (pLNs) were comparable between *CBP^WT* and *CBP^AA* mice (Supplementary Fig. 2a, b). However, the numbers of CD4⁺ and CD8⁺ T cells in the mLNs, the key local sites of gut inflammation[18], were increased in *CBP^AA* mice (Supplementary Fig. 2c–f). In the spleen, a decrease in the number of naïve CD4⁺ and CD8⁺ T cells but an increase in the number of effector or memory T cells were observed in *CBP^AA* mice (Supplementary Fig. 2g, h), suggesting alterations in T cell properties despite unchanged total cell numbers. In addition, the production of cytokines and mRNA expression levels of $T_H$-specific transcription factors in naïve CD4⁺ T cells were comparable between *CBP^AA* and *CBP^WT* mice under both neutral and $T_H$-skewed conditions (Supplementary Fig. 2i–n and 3). Although increased expression of the $T_H$2-associated transcription factors GATA3 and cMaf was observed in *CBP^AA* mice, neither mRNA expression nor secretion of IL-4 differed (Supplementary Fig. 3b). These findings further indicate that the colonic abnormalities in *CBP^AA* mice are attributed to impaired CBP phosphorylation in IECs per se, rather than to the regulation of baseline immune homeostasis.

### Impaired CBP phosphorylation hinders the renewal of colonic epithelial cells, and the mutant protein exhibits a binding preference for p53 in *CBP^AA* mice

Because impaired CBP phosphorylation in IECs contributes to colonic abnormalities in *CBP^AA* mice, the status of epithelial cells was examined by staining for Ki-67, a widely used marker of cell proliferation. A reduction in the number of proliferating IECs was observed in *CBP^AA* mice compared with control mice (Fig. 3a).

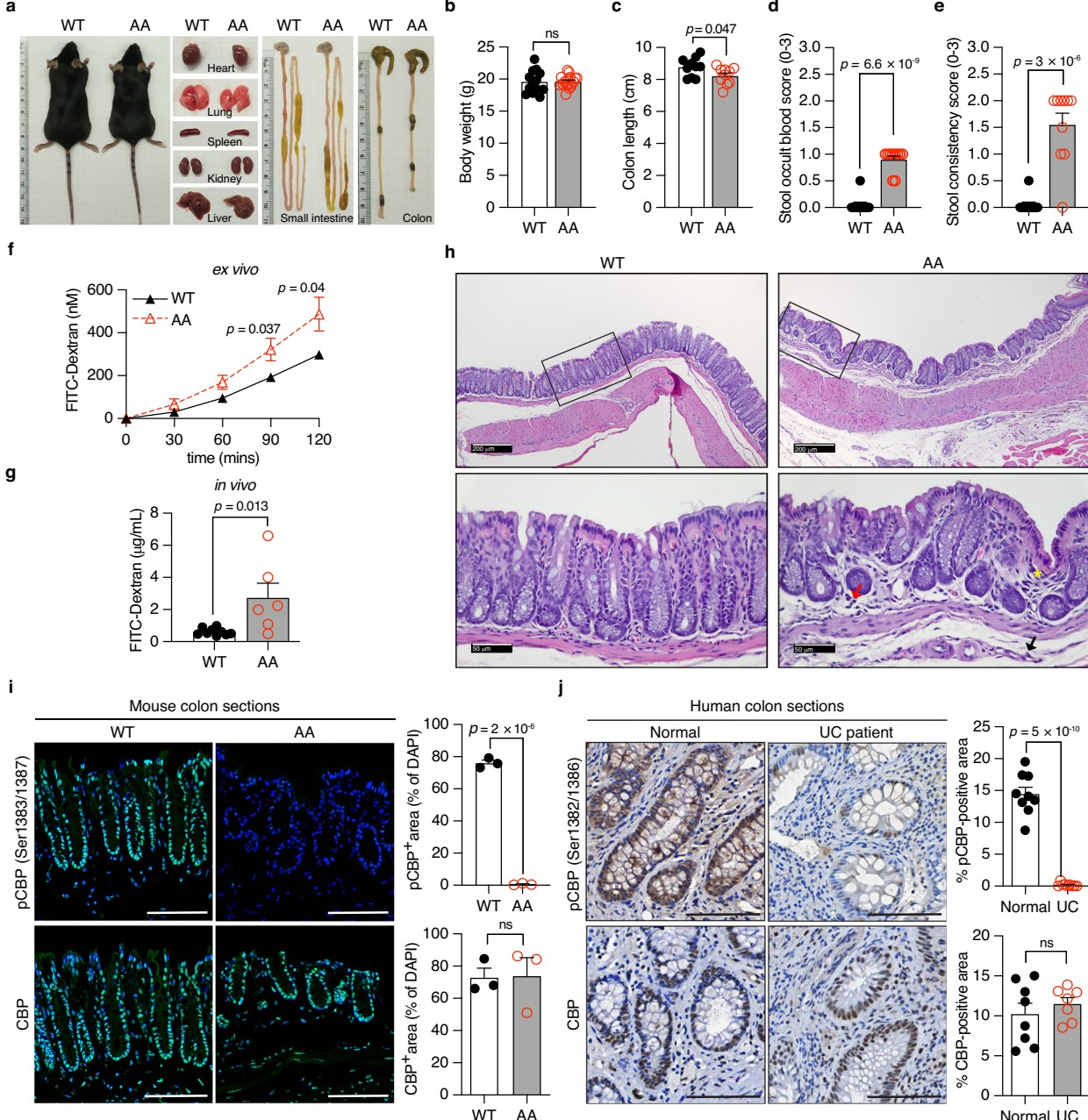

**Fig. 1 | Impaired phosphorylation of CBP at Ser1383/1387 results in colonic abnormalities in *CBP*^AA mice. a** Macroscopic images of organs from *CBP*^WT mice and their *CBP*^AA littermates. Body weight (**b**), colon length (**c**), stool occult blood score (**d**), and stool consistency score (**e**) of the *CBP*^WT and *CBP*^AA mice. *n* = 10 mice per group. **f** Ex vivo intestinal permeability measured by luminal-to-serosal FITC-dextran flux. *n* = 6 mice per group. **g** In vivo intestinal permeability was assessed by measuring the serum FITC-dextran concentration at 4 h postgavage. *n* = 9 mice for *CBP*^WT, *n* = 6 mice for *CBP*^AA. **h** Representative images of H&E-stained colon sections. Compared with *CBP*^WT mice, *CBP*^AA mice showed mild pathological changes in the distal colon, including slight infiltration of mononuclear cells (red arrow) and neutrophils (black arrow) in the mucosa and submucosa, along with slight crypt distortion (yellow asterisk). **i** Representative images of immunofluorescence staining for phosphorylated CBP (pCBP) and total CBP in colon sections from CBP

mice. The nuclei were counterstained with DAPI for normalization. Expression levels were calculated as the percentage of the pCBP^+ area to the DAPI^+ area per image field within a fixed region of interest (ROI) using ImageJ. Values are presented as %. *n* = 3 mice per group. **j** Representative images of IHC staining for pCBP and CBP in human colon sections. The positively stained area was quantified as a percentage of a fixed ROI using ImageJ. For anti-pCBP staining, *n* = 9 individuals per group; for anti-CBP staining, *n* = 8 individuals for normal controls, and *n* = 7 individuals for UC patients. Data are presented as the means ± SEM. All *p*-values were calculated using an unpaired two-tailed Student's *t*-test and are indicated on the graphs. ns not significant. Scale bars: 100 μm. Scale bars for images of H&E staining (**h**) are 200 μm (upper panels) and 50 μm (lower panels). Source data are provided as a Source Data file.

The integrity of the colonic epithelial barrier is challenged by the high turnover rate of IECs[16]. BrdU pulse-chase experiments were performed to assess IEC turnover in vivo. Two hours after the BrdU injection, BrdU-labeled cells were restricted to the crypt base, reflecting the baseline proliferation of IECs[19]. The subsequent migration of BrdU^+ cells from the crypt base toward the lumen was examined by performing immunohistochemical (IHC) staining at 24 h, 48 h, and 96 h after the BrdU injection. The BrdU^+ cells in *CBP*^WT mice had already reached the luminal surface by 96 h, whereas those in *CBP*^AA mice were still located within the crypt region and had not yet

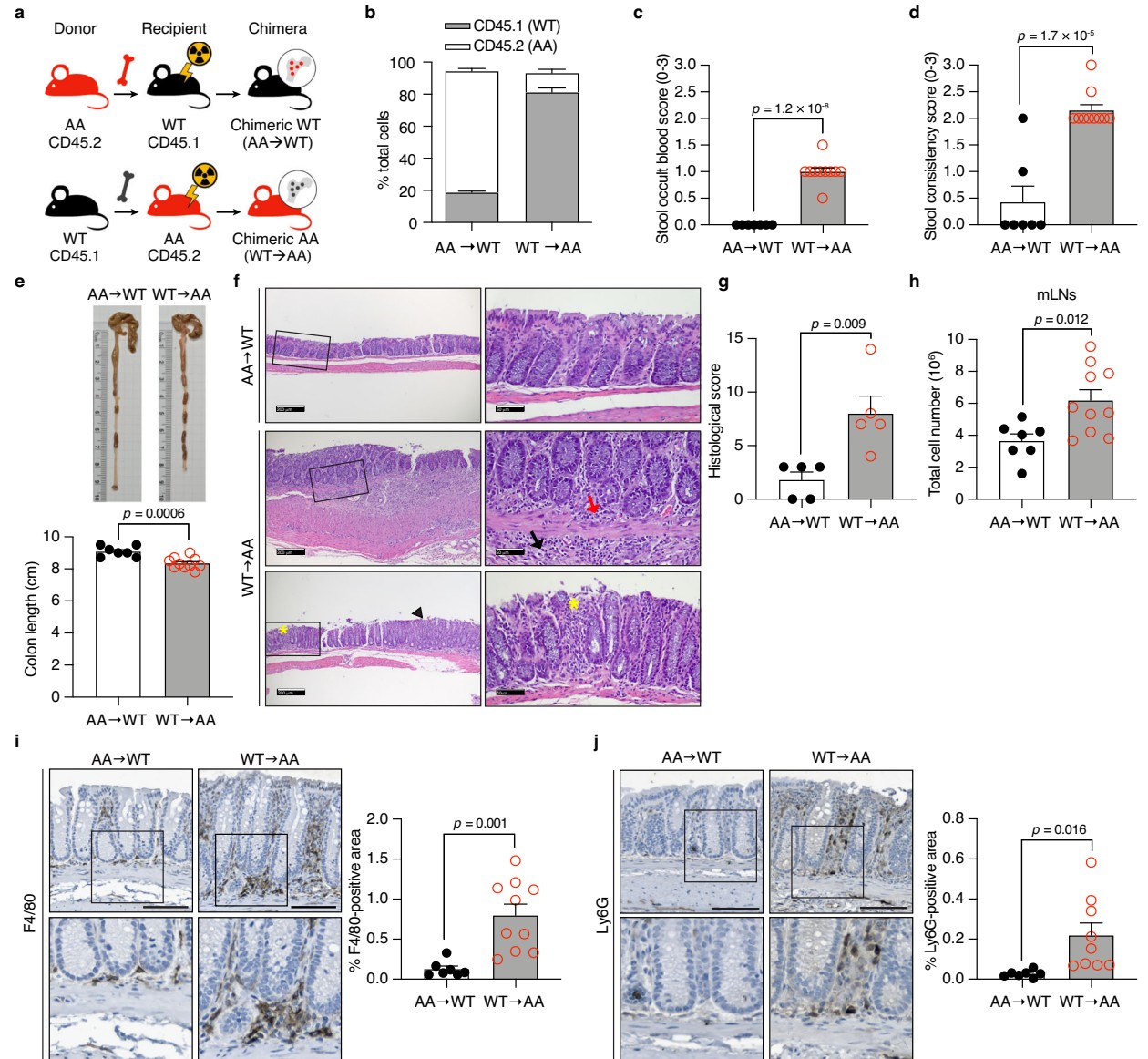

**Fig. 2 | Nonhematopoietic cell-driven colonic abnormalities in *CBP^AA* mice.** Bone marrow chimera experiments were performed using both male and female mice. The data were pooled for analysis. **a** Schematic of the bone marrow chimera experiment. Bone marrow-reconstituted mice were evaluated 6 weeks after transplantation. **b** The percent contributions of donor WT (CD45.1) or *CBP^AA* (CD45.2) bone marrow cells to γ-irradiated recipient mice. Stool occult blood scores (**c**) and stool consistency scores (**d**) in bone marrow-reconstituted mice. **e** Gross morphological images of the colons and measurement of the colon length. For (**b**–**e**), *n* = 7 biological replicates for AA → WT mice and *n* = 10 for WT → AA mice. **f** Representative images of H&E-stained colon sections. Sections from AA → WT mice show a normal histological architecture. Sections from WT → AA mice show pathological features, including the substantial infiltration of mononuclear cells (red arrow) and neutrophils (black arrow) across all layers of the bowel wall, severe crypt distortion with the loss of entire crypts (yellow asterisk), and moderate mucosal hyperplasia (black arrowhead). **g** Histopathological scores from blinded evaluations of H&E-stained colon sections. *n* = 5 mice per group. **h** The total cell number in mLNs was analyzed by flow cytometry. *n* = 7 biological replicates for AA → WT mice and *n* = 10 for WT → AA mice. **i, j** Representative images of IHC staining for F4/80 and Ly6G in colon sections. The positive staining area was quantified as a percentage of a fixed region of interest (ROI) using ImageJ. For (**i**), *n* = 7 biological replicates for AA → WT mice and *n* = 10 for WT → AA mice. For (**j**), *n* = 7 biological replicates for AA → WT mice and *n* = 9 for WT → AA mice. Data are presented as the means ± SEM. All *p*-values were calculated using an unpaired two-tailed Student's *t*-test and are indicated on the graphs. Scale bars: 100 μm (**i**, **j**); images of H&E staining (**f**) 200 μm (left panels) and 50 μm (right panels). Source data are provided as a Source Data file.

migrated to the lumen (Fig. 3b), suggesting a potential delay in IEC turnover.

Since the impairment in CBP phosphorylation induced by substituting serines with alanines increased its recruitment to *p53*-responsive promoters instead of to *NF-κB*-responsive promoters at the molecular level[12], we next sought to determine whether this event occurred in *CBP^AA* mice. A proximity ligation assay (PLA, Duolink®) evaluating protein–protein interactions in colonic IECs based on antibody detection was employed to examine CBP binding to p53 in *CBP^AA*

and *CBP^WT* mice. CBP in the IECs of *CBP^AA* mice preferentially bound to p53, and the expression of the p53 target p21 was increased (Fig. 3c, d). The expression levels of p53 (Fig. 3e) and CBP (Fig. 1i, lower panel) were comparable between *CBP^WT* and *CBP^AA* mice, indicating that the binding preference of CBP for p53 in *CBP^AA* mice is not due to alterations in p53 and CBP levels. CUT&RUN assays revealed that chromatin precipitated by both CBP and p53 was enriched at the *p21* promoter (Fig. 3f, g), supporting that enhanced recruitment of CBP by p53 regulates p21 expression in *CBP^AA* mice.

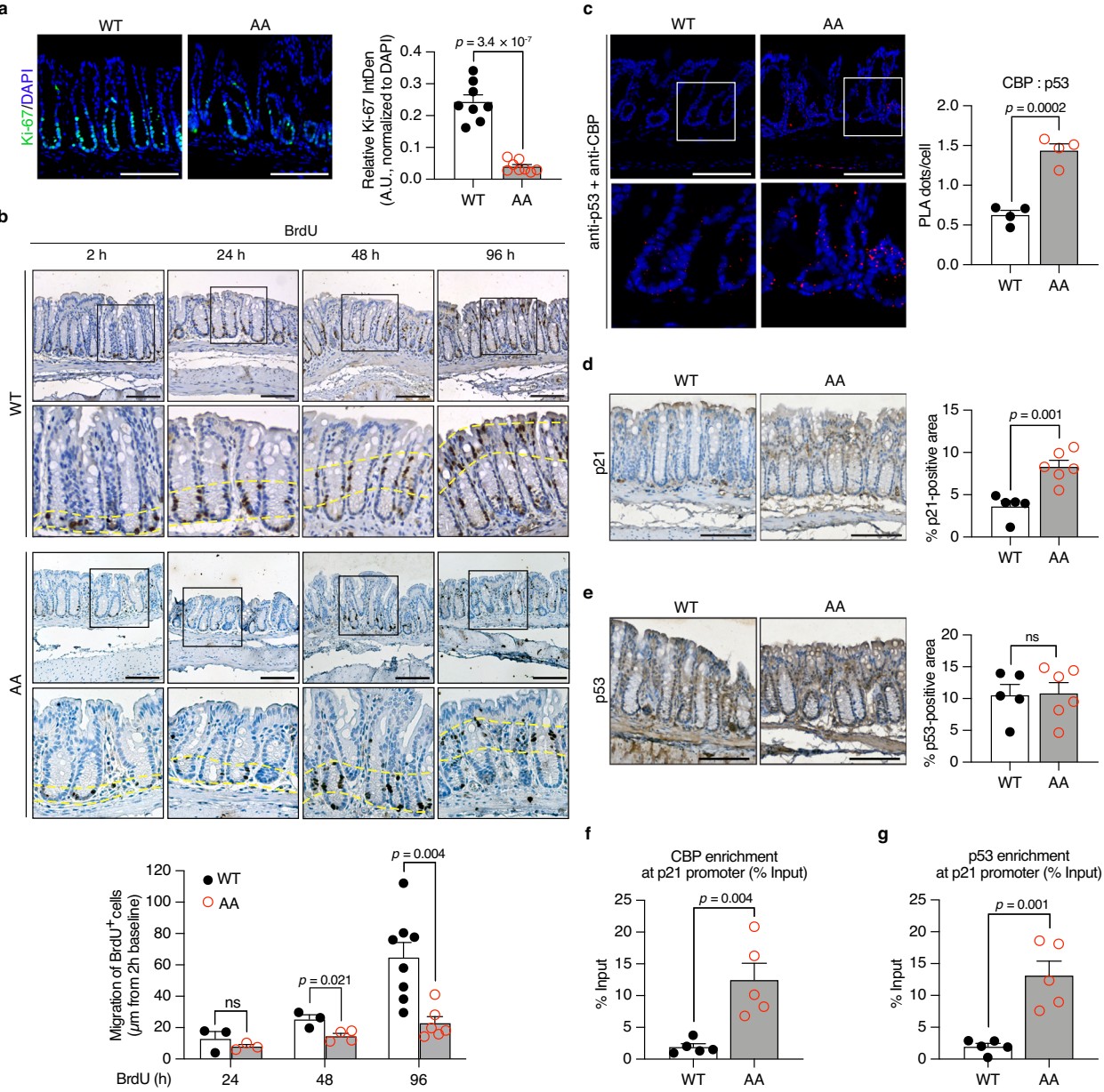

**Fig. 3 | Impaired phosphorylation of CBP retards the renewal of colonic epithelial cells, and the mutant protein shows a binding preference for p53 in *CBP^AA* mice. a** Representative images of immunofluorescence staining for Ki-67 in colon sections. The nuclei were counterstained with DAPI for normalization. Ki-67 is shown in green pseudo-color (originally captured as red Alexa Fluor 594) to maintain consistency across figures. Ki-67 expression was quantified as the integrated density (IntDen) normalized to the DAPI IntDen within a fixed region of interest (ROI) using ImageJ and is presented as arbitrary units (A.U.). $n = 8$ mice per group. **b** Representative images of IHC staining for BrdU in colon sections at 2, 24, 48, and 96 h after the BrdU injection. A 2-h pulse was used to mark the baseline position of proliferating crypt cells. Yellow dashed lines in magnified views indicate the upward migration trend of BrdU+ cells. The cell migration distance was quantified using ImageJ. For *CBP^WT* mice: $n = 3$ mice at 24 and 48 h, $n = 8$ mice at 96 h. For *CBP^AA* mice:

$n = 3$ mice at 24 h, $n = 4$ mice at 48 h, $n = 6$ mice at 96 h. **c** Representative images and quantification of PLA used to detect the CBP−p53 association in colon sections. Nuclei were counterstained with DAPI to identify individual cells. PLA signals (red dots) per cell were quantified using ImageJ. $n = 4$ mice per group. Representative images of IHC staining for p21 (**d**) and p53 (**e**) in colon sections. The positive staining area was quantified as a percentage of a fixed region of interest (ROI) using ImageJ. For (**d**) and (**e**), $n = 5$ mice for *CBP^WT*, and $n = 6$ mice for *CBP^AA*. The enrichment of CBP (**f**) and p53 (**g**) at the p21 promoter in *CBP^AA* mice was measured by the CUT&RUN assay using anti-CBP and anti-p53 antibodies, respectively. The data are shown as % input normalized to the input DNA of each sample. $n = 5$ mice per group. Data are presented as the means ± SEM. All *p*-values were calculated using an unpaired two-tailed Student's *t*-test and are indicated on the graphs. ns not significant. Scale bars: 100 μm (**a**, **b**, **d**, **e**), 50 μm (**c**). Source data are provided as a Source Data file.

## Impaired CBP phosphorylation reduces spheroid formation by HCT116 cells

The constantly self-renewing IECs are replenished by ISCs at the crypt base to maintain intestinal barrier integrity[4]. Dysregulation of ISCs results in delayed IEC turnover[20], leading to epithelial barrier dysfunction and colitis[21]. Since impaired CBP phosphorylation in the colonic IECs of *CBP^AA* mice resulted in a binding preference

of mutant CBP for p53 that paralleled the reduced IEC proliferation and delayed epithelial turnover (Fig. 3), WT and $p53^{-/-}$ human HCT116 cells[22–24] were first utilized to examine whether and how CBP−p53 binding dysregulates stemness. Spheroids are highly enriched in stem cell populations[25], and the functions of normal stem cells and cancer stem cells are conceptually similar[26].

An IKK inhibitor, CYL-19s, has been shown to abolish TNF-α-induced CBP phosphorylation in HeLa cells and increase CBP binding to *p53*-responsive promoters instead of *NF-κB*-responsive promoters[12,27]. Similar results were also observed in WT HCT116 cells in the present study (Supplementary Fig. 4a–c). The inhibition of basal CBP phosphorylation by CYL-19s led to increased CBP–p53 binding without affecting p53 levels, along with increased protein expression of the p53 target p21 (Supplementary Fig. 4a–e). In addition, the viability of WT HCT116 cells decreased in response to CYL-19s treatment (Supplementary Fig. 4f). None of these changes were observed in *p53⁻/⁻* HCT116 cells (Supplementary Fig. 4b–d, f).

We determined whether and how increased CBP–p53 binding weakens HCT116 stemness by examining the effects of spheroid formation by WT and *p53⁻/⁻* HCT116 cells in the presence of CYL-19s. CYL-19s treatment significantly inhibited sphere formation by WT HCT116 cells (Supplementary Fig. 4g) and reduced the expression of the stemness marker ALDH1A1 (Supplementary Fig. 4d), suggesting that

increased CBP–p53 binding suppresses spheroid formation (Supplementary Fig. 4b, c, g).

The spheres formed by *p53⁻/⁻* HCT116 cells were larger than those formed by WT HCT116 cells (Supplementary Fig. 4g), and ALDH1A1 expression was increased (Supplementary Fig. 4d), indicating that increased CBP–p53 binding played a negative role in regulating HCT116 cell stemness since this event was relieved in *p53⁻/⁻* HCT116 cells (Supplementary Fig. 4b). Increased ALDH1A1 expression and larger spheres were sustained in *p53⁻/⁻* HCT116 cells in the presence of CYL-19s (Supplementary Fig. 4d, g), further indicating that increased CBP–p53 binding decreases HCT116 cell stemness.

### Impaired CBP phosphorylation hinders the formation of organoids from *CBPᴬᴬ* mice

Since increased CBP–p53 binding suppressed sphere formation and reduced the expression of stemness markers in HCT116 cells, confirming this event in colonic IECs by using ex vivo primary 3D colonic

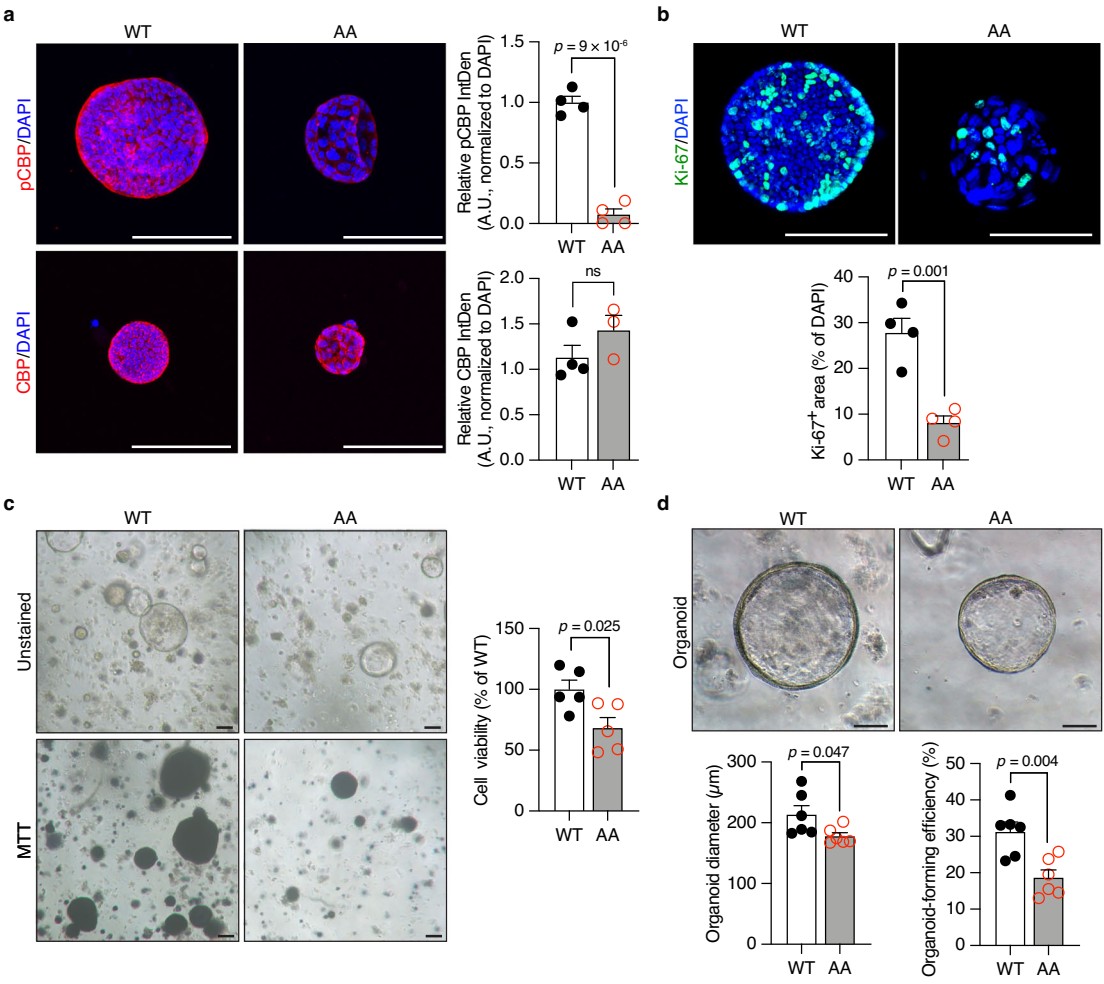

**Fig. 4 | Impaired phosphorylation of CBP hinders the formation of organoids from *CBPᴬᴬ* mice. a** Representative images of immunofluorescence staining for phosphorylated CBP (pCBP) and total CBP from cultured organoids on Day 6. The nuclei were counterstained with DAPI. Expression levels were quantified as the integrated density (IntDen) normalized to the DAPI IntDen per image field using ImageJ, and presented as arbitrary units (A.U.) relative to the mean of *CBPᵂᵀ* controls. For anti-pCBP staining, *n* = 4 biological replicates per group. For anti-CBP staining, *n* = 4 for *CBPᵂᵀ* and *n* = 3 for *CBPᴬᴬ* biological replicates. **b** Representative images of immunofluorescence staining for Ki-67. The nuclei were counterstained with DAPI. The Ki-67⁺ area was quantified as a percentage of the total DAPI⁺ area per field using ImageJ. *n* = 4 biological replicates per group. **c** Microscopy images (4×)

and organoid cell viability measured by the MTT assay. The cell viability was normalized to that of the *CBPᵂᵀ* organoids. *n* = 5 biological replicates per group. **d** Microscopy images of cultured organoids on Day 6 (10×). The organoid diameter and organoid-forming efficiency were quantified using ImageJ. Organoid-forming efficiency was calculated as: (number of organoids on Day 6 per well / 400 seeded crypts) × 100%. For each mouse, the values were averaged from at least three wells. *n* = 6 biological replicates per group. Data are presented as the means ± SEM. All *p*-values were calculated using an unpaired two-tailed Student's *t*-test and are indicated on the graphs. ns not significant. Scale bars: 100 μm. Source data are provided as a Source Data file.

organoids is essential. We established a near-physiological primary epithelial 3D organoid culture system from colonic crypts. Crypt-derived organoids consisting of various differentiated epithelial lineages were generated from single *Lgr5*[+] (Leucine-rich repeat-containing G protein-coupled receptor 5-expressing) stem cells[28]. CBP phosphorylation was detected in *CBP^WT* organoids using immunofluorescence staining, but this PTM was absent in *CBP^AA* organoids, despite the comparable total CBP levels between the two groups (Fig. 4a). Similar to colon tissues, *CBP^AA* organoids exhibited a reduced number of Ki-67[+] cells and decreased cell viability (Fig. 4b, c). Additionally, both the formation efficiency and the diameter of *CBP^AA* organoids were reduced (Fig. 4d). These findings indicate that the loss of CBP phosphorylation in *CBP^AA* mice is associated with impaired IEC proliferation and reduced stemness, likely through its preferential binding to p53, and contributes to colonic abnormalities in *CBP^AA* mice.

## Targeted deletion of p53 in IECs diminishes colonic abnormalities and restores epithelial stemness in *CBP^AA* mice

Because increased CBP–p53 binding, which decreases IEC stemness, was observed in vitro and ex vivo, *CBP^AA* mice with an IEC-specific p53 deletion (*CBP^AA p53^ΔIEC* mice) were generated to further examine whether the abnormal colonic phenotypes would be alleviated and colonic IEC stemness would be restored in the absence of CBP–p53 binding.

p53 was specifically knocked out in the IECs of *CBP^AA* mice by crossbreeding *CBP^AA* mice with *Villin-Cre; p53^f/f* mice. qPCR and Western blotting confirmed the effective deletion of p53 in the colonic epithelial cells of *CBP^AA p53^ΔIEC* mice (Fig. 5a, b). *CBP^AA p53^ΔIEC* mice exhibited significant improvements in the abnormal colonic phenotypes caused by the *CBP^AA* mutation, as evidenced by the restored colon length, reduced occult blood in stool and stool consistency scores, and decreased intestinal permeability (Fig. 5c–f), along with improvements in the histological abnormalities (Fig. 5g).

The preferential CBP–p53 binding observed in *CBP^AA p53*-floxed mice was comparable to that in *CBP^AA* mice (compare Fig. 5h, left two panels, and Fig. 3c). In contrast, CBP–p53 binding was lost in *CBP^AA p53^ΔIEC* mice (Fig. 5h, right two panels), which paralleled the restoration of the reduced numbers of Ki-67[+] cells, organoid diameter, and organoid-forming efficiency (Fig. 5i, j). These findings support that increased CBP–p53 binding impairs colonic IEC stemness and drives colonic abnormalities in *CBP^AA* mice, whereas the loss of this interaction restores stemness and prevents these abnormalities in *CBP^AA p53^ΔIEC* mice.

Enrichments of both CBP- and p53-precipitated chromatin at the *p21* gene promoter were observed in *CBP^AA* mice (Fig. 3f, g). However, CBP enrichment at the *p21* promoter was absent in *CBP^AA p53^ΔIEC* mice, accompanying a reduction in p21 expression (Supplementary Fig. 5), in which CBP–p53 binding was lost (Fig. 5h, right two panels). These results confirm that direct regulation of CBP on p53 is enhanced at the chromatin level by impaired CBP phosphorylation in *CBP^AA* mice, providing further evidence that differential phosphorylation of CBP regulates binding to the p53 locus and subsequent p21 expression.

## Versican expression is downregulated in *CBP^AA p53*-floxed mice and restored in *CBP^AA p53^ΔIEC* mice

RNA-seq was performed to profile the transcriptome of colonic crypt-enriched epithelial cells to investigate how impaired CBP phosphorylation results in decreased IEC stemness. The differentially expressed genes (DEGs) are depicted in heatmaps that show a distinct expression pattern in crypts between *CBP^WT p53*-floxed and *CBP^AA p53*-floxed mice (Fig. 6a), and a volcano plot shows that versican (*Vcan*), a proteoglycan in the ECM, was significantly downregulated in *CBP^AA p53*-floxed mice (Fig. 6b). A Venn diagram depicts the overlapping DEGs between the two experimental groups. Versican was also the only gene overlapping between the comparisons of the *CBP^AA p53*-floxed vs. *CBP^WT p53*-floxed and *CBP^AA p53^ΔIEC* vs. *CBP^AA p53*-floxed groups but was excluded from the

comparisons of the *CBP^AA p53^ΔIEC* vs. *CBP^WT p53^ΔIEC* and *CBP^WT p53^ΔIEC* vs. *CBP^WT p53*-floxed groups (Fig. 6c). RNA-seq revealed that versican expression was downregulated in *CBP^AA p53*-floxed mice but restored in *CBP^AA p53^ΔIEC* mice to levels comparable to those in *CBP^WT p53^ΔIEC* and *CBP^WT p53*-floxed mice (Fig. 6d), which was further verified by qPCR (Fig. 6e). Immunofluorescence staining showed reduced versican expression in the mucosal stroma of *CBP^AA p53*-floxed mice, which was restored in *CBP^AA p53^ΔIEC* mice to levels comparable to those in *CBP^WT p53^ΔIEC* mice (Fig. 6f, g).

Versican mediates the mesenchymal–epithelial transition by promoting the switch from N-cadherin expression to E-cadherin expression, thereby facilitating the formation of epithelial-specific adhesion junctions[29]. E-cadherin is required for maintaining epithelial integrity, but defects in E-cadherin expression are associated with IBD[30,31]. Notably, the expression levels of E-cadherin paralleled those of versican (Fig. 6h, i vs. 6f, g), suggesting that versican expression may influence cell–cell adhesion and epithelial barrier integrity.

The recombinant versican protein was added to the culture medium of *CBP^AA p53*-floxed organoids (AA + Vcan) to examine the role of versican in IEC stemness. The organoid diameter, organoid-forming efficiency, number of Ki-67[+] cells, and E-cadherin expression were restored to levels comparable to those in *CBP^WT p53*-floxed organoids (Fig. 6j–p), suggesting that versican contributes to the maintenance of IEC stemness and the regulation of intestinal epithelial cell behavior. Versican contains two EGF-like repeats in its G3 domain, and upon cleavage, EGF may be released to promote proliferation[32,33]. The restorative effect of recombinant versican was abolished by gefitinib (Fig. 6j–p), suggesting the involvement of the activated EGFR pathway in its function.

Versican is produced by various cell types, including epithelial, endothelial, and stromal cells, as well as leukocytes[34]. In addition to the RNA-seq and qPCR results (Fig. 6a, d, e), the epithelial-intrinsic defect and reduced versican expression observed in *CBP^AA* mice were reversed by epithelial-specific p53 deletion (Fig. 6f), suggesting that versican is derived primarily from epithelial cells. Consistent with these tissue-level changes (Fig. 6f), versican expression was also reduced in organoids derived from the crypts of *CBP^AA p53*-floxed mice and restored in those derived from *CBP^AA p53^ΔIEC* mice (Fig. 6q, r), further confirming its epithelial origin.

## Overexpression of phosphomimetic CBP mutants restores versican expression and rescues the phenotype of *CBP^AA p53^f/f* organoids

As mentioned, CBP phosphorylation is important for maintaining intestinal homeostasis, and versican contributes to the maintenance of IEC stemness and epithelial cell behavior. We performed gain-of-function experiments using phosphomimetic CBP mutants to further determine whether CBP phosphorylation affects versican levels. Single cells dissociated from crypt-derived *CBP^AA p53^f/f* (AA) organoids were transfected with either a control vector expressing EGFP alone (pCMV-puro-IRES2-*EGFP*) or a construct encoding HA-tagged CBP phosphomimetic mutant coexpressing EGFP (DD: pCMV-puro-IRES2-*EGFP mCBP^S1383D/S1387D*-HA; EE: pCMV-puro-IRES2-*EGFP mCBP^S1383E/S1387E*-HA) using Lipofectamine™ Stem Transfection Reagent. Transfected cells were identified by EGFP fluorescence. Successful delivery of HA-tagged CBP mutants into AA organoids was confirmed by HA-CBP expression detected by Western blotting (Supplementary Fig. 6). Compared with the vector control, AA organoids overexpressing DD or EE showed restored versican expression and a reversal of AA phenotype, as evidenced by the rescued organoid-forming efficiency and organoid diameter (Fig. 7a, b). Additionally, these organoids exhibited increased IEC proliferation and E-cadherin expression (Fig. 7c, d). These findings indicate that CBP phosphorylation plays important roles in maintaining versican expression and supporting the intestinal stem cell niche, thereby contributing to the preservation of epithelial homeostasis in the colon. Despite the low number of EGFP-positive cells,

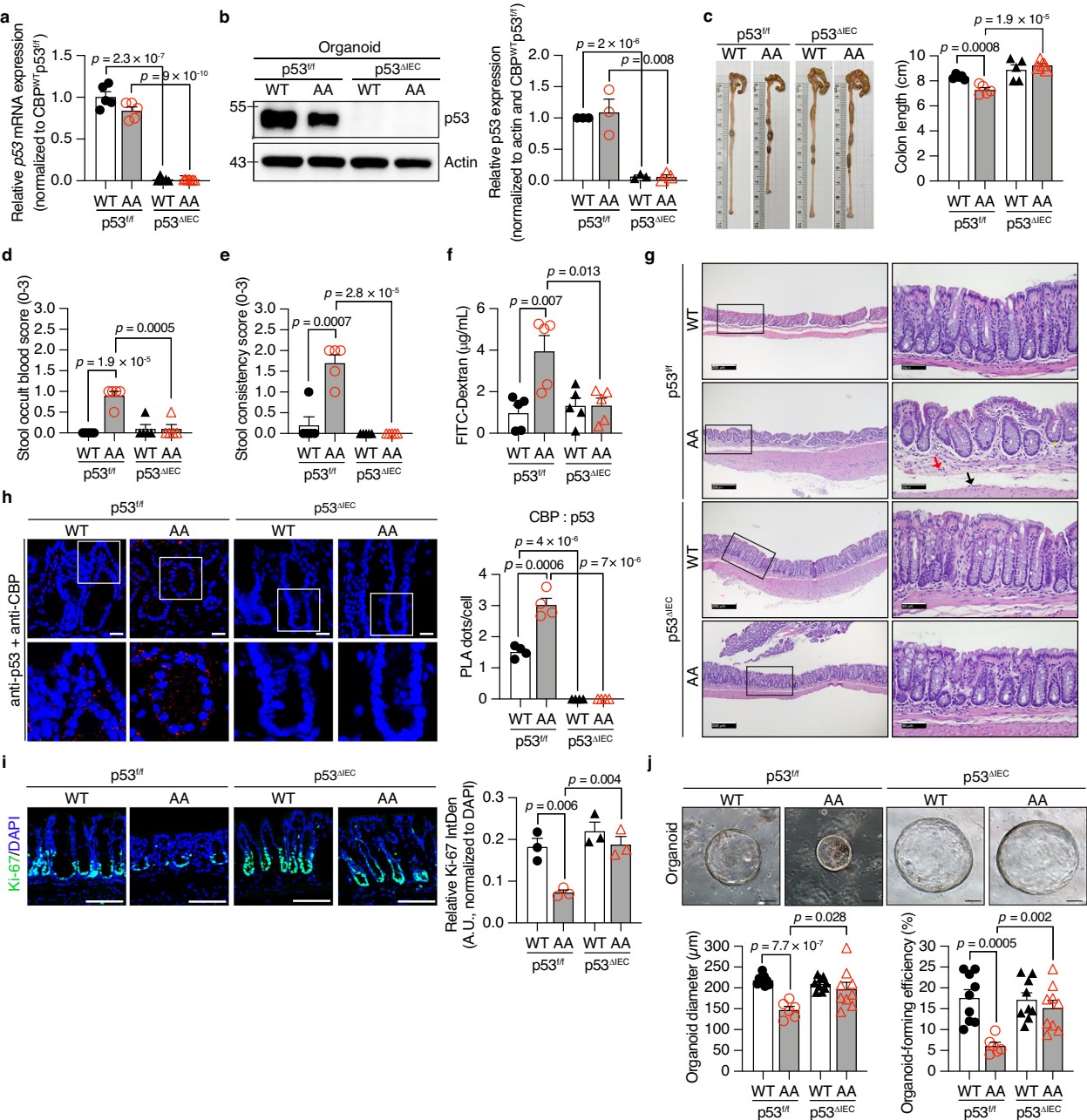

**Fig. 5 | Colonic abnormalities are attenuated, and the decreased IEC stemness is restored in $CBP^{AA}p53^{\Delta IEC}$ mice. a** Quantitative RT–PCR analysis of *p53* expression in isolated colonic crypts. The expression was normalized to that of $CBP^{WT}p53^{f/f}$ mice. *n* = 5 mice for all groups except $CBP^{AA}p53^{\Delta IEC}$ mice (*n* = 7). **b** Representative immunoblots showing p53 levels in total lysates from cultured organoids on Day 6. The p53 protein levels were quantified using ImageJ, normalized to those of β-actin, and further normalized to the mean value of $CBP^{WT}p53^{f/f}$ organoids. β-actin served as a loading control. Data represent *n* = 3 independent biological replicates (derived from different mice) per group. Full uncropped scans for all replicates, including the representative blot and additional independent experiments, are provided in the Source Data file. **c** Gross images of colons and the colon length. *n* = 5 mice per group. **d** Stool occult blood scores. *n* = 5 mice per group. **e** Stool consistency scores. *n* = 5 mice per group. **f** Intestinal permeability was assessed by determining the serum FITC-dextran concentration 4 h after oral gavage. *n* = 5 mice per group. **g** Representative images of H&E-stained colon sections. The colon of $CBP^{AA}p53^{f/f}$ mice showed slight infiltration of mononuclear cells (red arrow) and neutrophils (black arrow) in the mucosa and submucosa, along with slight crypt distortion (yellow asterisk). The colons from $CBP^{WT}p53^{f/f}$, $CBP^{WT}p53^{\Delta IEC}$, and $CBP^{AA}p53^{\Delta IEC}$ mice

exhibited a normal histological architecture. **h** Representative images and quantification of the results of the PLA for detecting CBP–p53 associations in colon sections. Nuclei were counterstained with DAPI to identify individual cells. PLA signals (red dots) per cell were quantified using ImageJ. *n* = 4 mice per group.
**i** Representative images of immunofluorescence staining for Ki-67 in colon sections. The nuclei were counterstained with DAPI for normalization. Ki-67 expression was quantified as the integrated density (IntDen) normalized to the DAPI IntDen within a fixed region of interest (ROI) using ImageJ and is presented as arbitrary units (A.U.). *n* = 3 mice per group. **j** Microscopy images of cultured organoids on Day 6 (10×). The organoid diameter and organoid-forming efficiency were quantified using ImageJ. The organoid-forming efficiency was calculated as follows: (number of organoids on Day 6 ÷ 400 seeded crypts per well) × 100%. For each mouse, the values were averaged from at least three wells. *n* = 9 mice for all groups except for $CBP^{AA}p53^{f/f}$ (*n* = 6 mice). Data are presented as the means ± SEM. All *p*-values were calculated using an unpaired two-tailed Student's *t*-test and are indicated on the graphs. Scale bars: 100 μm (**i**, **j**), 20 μm (**h**); images of H&E images (**g**) 200 μm (left panels) and 50 μm (right panels). Source data are provided as a Source Data file.

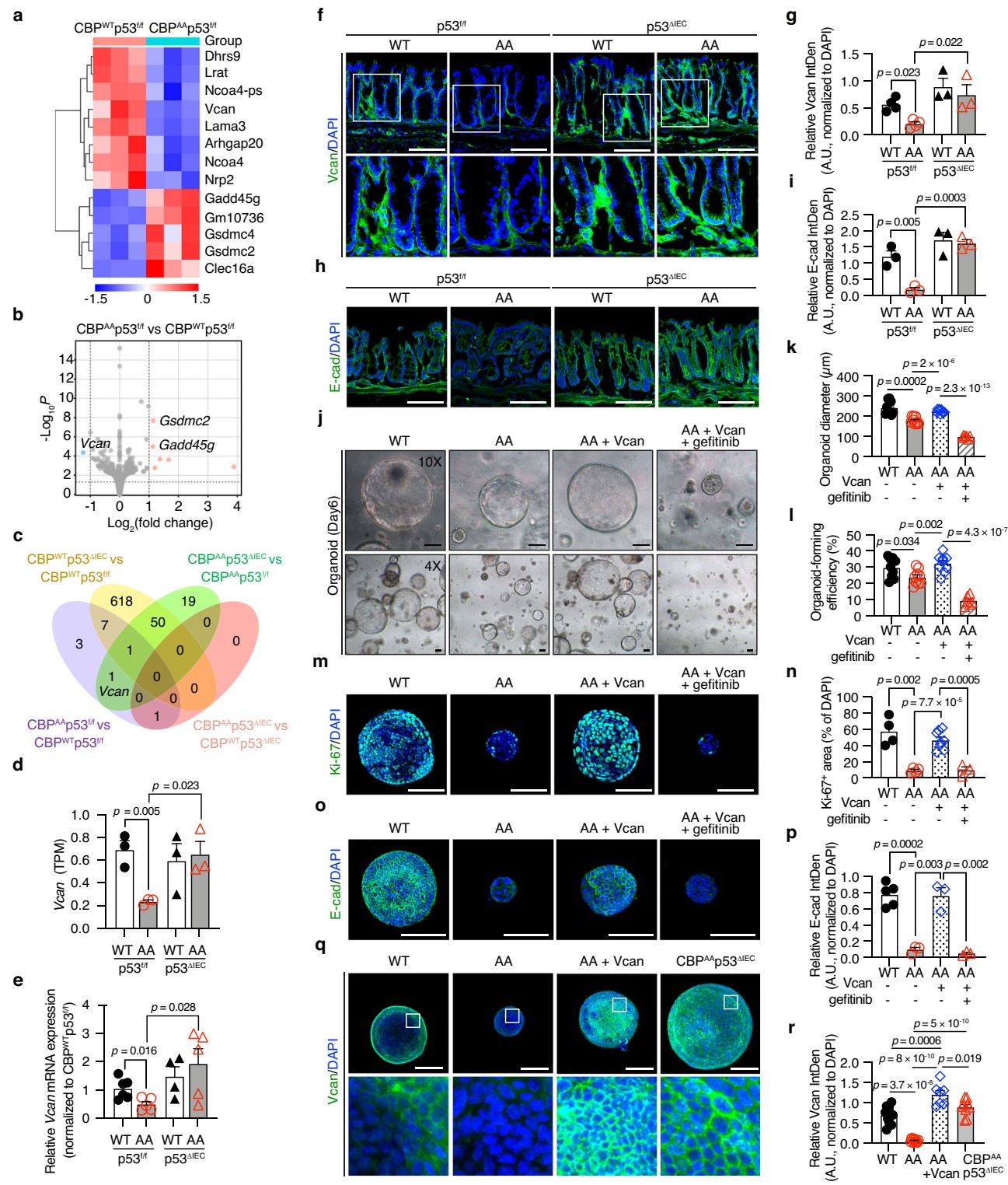

overexpression of DD or EE in AA organoids upregulated versican expression (Fig. 7a, e). This finding parallels the concept that localized alterations within the stem cell niche can have broad effects on epithelial structure and function[35].

**Binding preference of CBP for p53 and reduced versican expression in colon tissues from UC patients**

The observation of impaired CBP phosphorylation without changes in total CBP levels in UC patients underscores its clinical relevance (Fig. 1j). Consistently, increased CBP–p53 binding, reduced versican

levels, and fewer Ki-67+ proliferative IECs were observed in UC patients (Fig. 8a–c), along with decreased IKKα expression (Fig. 8d). These findings strengthen the hypothesis that impaired CBP phosphorylation enhances CBP–p53 binding, which in turn suppresses versican expression and may ultimately contribute to IBD pathogenesis in humans.

Moreover, healthy individuals presented preserved IKKα expression, intact CBP phosphorylation, maintained versican levels, and abundant Ki-67+ IECs (Fig. 8b–d and Fig. 1j). Together, these results underscore the essential role of CBP phosphorylation in maintaining

**Fig. 6 | Versican expression is reduced in *CBP^AA^p53^f/f^* mice but restored in *CBP^AA^p53^ΔIEC^* mice. a–d** RNA-seq analysis of colonic crypts from *CBP^WT^p53^f/f^*, *CBP^AA^p53^f/f^*, *CBP^WT^p53^ΔIEC^*, and *CBP^AA^p53^ΔIEC^* mice. *n* = 3 mice per group. Heatmap (**a**) and volcano plot (**b**) of differentially expressed genes (DEGs; |log₂FC| > 0.5; *p*_adj_ value < 0.05) between *CBP^WT^p53^f/f^* and *CBP^AA^p53^f/f^* samples. Red, high expression; blue, low expression. The graphs in (**a**) and (**b**) were generated using the SR plot. **c** Venn diagram of DEGs (same cutoff) generated using Venny 2.1. **d** *Vcan* (versican) expression levels are presented as TPMs (transcripts per million). **e** Quantitative RT–PCR analysis of *Vcan* (versican) expression in the colonic crypts. Relative expression levels are shown as the fold changes compared with those in the *CBP^WT^p53^f/f^* group. *n* = 6 mice for *CBP^WT^p53^f/f^*, *n* = 5 mice for *CBP^AA^p53^f/f^*, *n* = 4 mice for *CBP^WT^p53^ΔIEC^*, *n* = 5 mice for *CBP^AA^p53^ΔIEC^*. Representative images of immuno-fluorescence staining for versican (**f**) and E-cadherin (**h**) in colon sections. The nuclei were counterstained with DAPI. Quantification of versican (**g**) and E-cadherin (**i**) expression in colon sections, calculated as the integrated density (IntDen) normalized to the DAPI intensity within a fixed ROI using ImageJ and presented as arbitrary units (A.U.). For (**g**), *n* = 4 mice per *p53^f/f^* group; *n* = 3 mice per *p53^ΔIEC^* group. For (**i**), *n* = 3 mice per group. **j–p** Organoids were collected on Day 6. The experimental groups included *CBP^WT^p53^f/f^* (WT), *CBP^AA^p53^f/f^* (AA), AA organoids treated with 100 ng/mL versican from Day 0 (AA + Vcan), and AA organoids treated with 1 μM gefitinib from Day 3 (AA + Vcan + gefitinib). **j** Microscopy images of cultured organoids on Day 6 captured at 10× (upper panels) and 4× (lower panels). The organoid diameter (**k**) and organoid-forming efficiency (**l**) were quantified using ImageJ. Organoid-forming efficiency was calculated as: (number of organoids on Day 6 per well / 400 seeded crypts) × 100%. For each mouse, the values were averaged from at least three wells. *n* = 9 biological replicates (WT, AA, AA + Vcan), *n* = 6 biological replicates (AA + Vcan + gefitinib). **m, n** Representative images (**m**) and quantification (**n**) of the Ki-67⁺ area, calculated as the percentage of total DAPI⁺ area per field using ImageJ. *n* = 4 biological replicates (WT, AA), *n* = 8 biological replicates (AA + Vcan), *n* = 3 biological replicates (AA + Vcan + gefitinib). **o, p** Representative images (**o**) and quantification of E-cadherin expression in organoids. *n* = 5 biological replicates (WT), *n* = 3 biological replicates (AA, AA + Vcan, AA + Vcan + gefitinib). **q, r** Representative images (**q**) and quantification (**r**) of versican expression in organoids on Day 6. The experimental groups included WT, AA, AA + Vcan, and *CBP^AA^p53^ΔIEC^* organoids. *n* = 10 biological replicates (WT, AA), *n* = 7 biological replicates (AA + Vcan), and *n* = 9 biological replicates (*CBP^AA^p53^ΔIEC^*). For **m–r**, the nuclei were counterstained with DAPI. Except for Ki-67⁺, expression levels were calculated as IntDen normalized to DAPI intensity using ImageJ and presented as A.U. Data are presented as the means ± SEM. All *p*-values were calculated using an unpaired two-tailed Student's *t*-test and are indicated on the graphs. Scale bars: 100 μm (**f, h, j, m, o, q**). Source data are provided as a Source Data file.

versican expression and the epithelial stem cell niche, thereby sustaining intestinal homeostasis and preventing IBD.

## Discussion

Protein PTMs significantly expand the regulatory possibilities of certain signaling pathways[8]. CBP and its homolog p300 are transcriptional coactivators that modulate DNA accessibility for transcription factors. These coactivators mediate communication between transcription factors and the transcriptional machinery, playing a key role in gene transcription[36]. Since CBP/p300 are essential for numerous signal transduction pathways, their availability is limited for the proper execution of biological processes, such as the functional antagonism between p53 and NF-κB, which compete for binding to CBP or p300[11]. Both p53 and NF-κB physically interact with CBP and require this interaction to maximize their activities[37,38]. However, increased CBP expression alone cannot completely rescue these reciprocal inhibitory effects[11], suggesting that the limited quantity of CBP cannot fully explain the crosstalk between p53 and NF-κB. Based on these findings, we showed that IKKα-induced phosphorylated CBP switches its binding preference from p53 to NF-κB, leading to increased NF-κB-mediated gene expression but decreased p53-mediated gene expression, which may promote cell proliferation[12]. However, the mutation of Ser1382 and Ser1386 to Ala impairs CBP phosphorylation, leading to its preferential recruitment to *p53*-responsive promoters such as the *p21* promoter. In the present study, we demonstrated that our previous mechanistic findings at the molecular level[12] have translational significance in *CBP^AA^* mice and UC patients. This clinical relevance is supported by the findings that colon tissues from UC patients show reduced IKKα expression and impaired CBP phosphorylation, increased CBP–p53 binding, and decreased versican expression and IEC proliferation. In healthy individuals, we observed intact IKKα expression, CBP phosphorylation, and versican expression, and increased numbers of Ki-67⁺ IECs. Therefore, CBP phosphorylation plays a key role in maintaining the stem cell niche and IEC self-renewal through the regulation of versican expression to maintain intestinal homeostasis.

The homeostasis of IECs is tightly orchestrated by the rapid self-renewal of ISCs[4]. ISCs are a small population of *Lgr5*⁺ stem cells residing at the crypt base that are responsible for the continual self-renewal and rapid recovery of the intestinal epithelium[4]. *Lgr5*⁺-expressing ISCs are highly proliferative, with a cell cycle of 12–24 h, and are critical for supporting intestinal homeostasis. ISCs are also tightly regulated by the surrounding stem cell niche in the lamina propria through multiple signaling pathways to control cell proliferation and differentiation[39]. The ISC niche contains both cellular and extracellular components[39].

The cellular niche includes various stromal cells that secrete essential growth factors and contribute to the production of ECM components. The ECM, which is composed of laminins, fibronectins, collagens, noncollagenous glycoproteins, proteoglycans, and glycosaminoglycans, is secreted locally to form a complex macromolecular meshwork[7]. It can also function as a rich reservoir for growth factors and cytokines, which can be bound within its structure and released after proteolysis to modulate cellular proliferation and differentiation, thereby maintaining ISC homeostasis[7,39]. The ECM has been reported to be essential for maintaining epithelial barrier integrity, and its dysregulation can lead to the pathogenesis of IBD[40,41].

In the present study, the number of *Lgr5*⁺ stem cells in the IECs did not differ between *Lgr5-eGFP-creERT2^+/Tg^; Villin-Cre; CBP^AA^* (*Lgr5-CBP^AA^*) mice and *Lgr5-eGFP-creERT2^+/Tg^; Villin-Cre; CBP^WT^* (*Lgr5-CBP^WT^*) mice (Supplementary Fig. 7). However, intestinal stem cell function was compromised in *CBP^AA^* mice, as evidenced by the reduced organoid formation, decreased number of Ki-67⁺ IECs, and reduced versican expression. The unchanged number of *Lgr5*⁺ stem cells suggests that the observed impairment in stemness may be attributed to a disruption in the stem cell niche, likely caused by alterations in the ECM, rather than a direct loss of the stem cell population. Currently, epithelial-derived versican may contribute to the local regulation of the stem cell niche, as replenishing versican into the culture medium of *CBP^AA^* organoids restored organoid formation, the organoid diameter, and the number of Ki-67⁺ cells to levels comparable to those of *CBP^WT^* organoids. The loss of CBP–p53 binding in *CBP^AA^p53^ΔIEC^* mice also restored versican expression and IEC stemness to prevent colonic abnormalities. In contrast, impaired CBP phosphorylation in *CBP^AA^* mice increases CBP–p53 binding, leading to the downregulation of versican expression and contributing to colonic abnormalities. An analysis of the versican gene promoter and enhancer regions using the PROMO and EPD databases did not reveal classical p53 binding sites. The precise mechanism by which CBP–p53 binding negatively regulates versican expression in IECs seems to be indirect and remains to be investigated.

Versican is a highly versatile molecule associated with a proliferative cell phenotype through its direct or indirect interactions with other molecules[33]. This finding is also consistent with the reports that versican can bind to cell surface proteins, such as CD44, integrin β1, EGFR, and P-selectin glycoprotein ligand-1, to influence the behavior of stem cells, such as ISC proliferation, cell adhesion/migration, crypt fission, and intestinal growth[7,28,41]. Furthermore, versican can influence proliferation by acting as a mitogen through the two EGF repeats in its G3 domain[32,33]. EGF exerts a strong mitogenic effect on stem cells upon

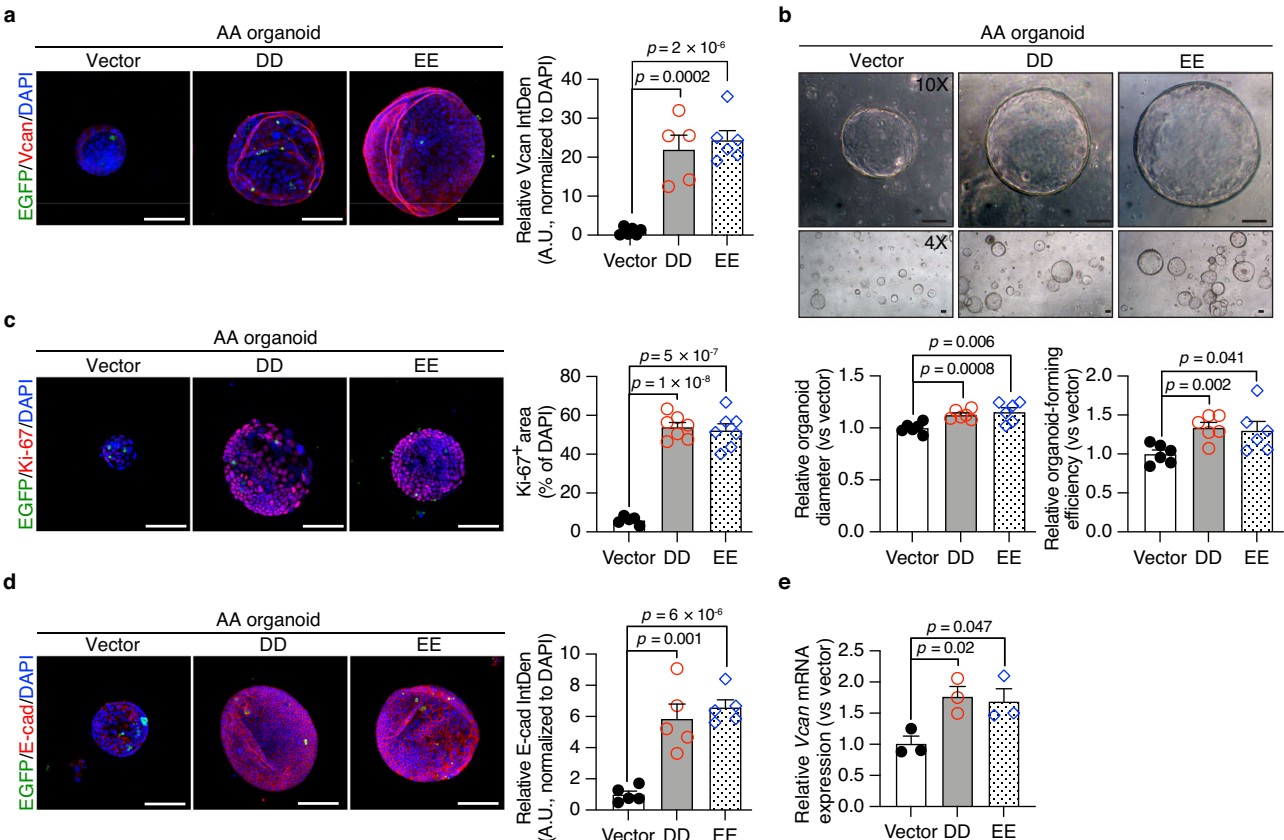

**Fig. 7 | Phosphomimetic CBP mutants rescue versican expression and stemness of $CBP^{AA}pS3^{f/f}$ organoids.** Single cells derived from dissociated $CBP^{AA}pS3^{f/f}$ (AA) organoids were transfected with either a control vector expressing EGFP alone (pCMV-puro-IRES2-*EGFP*) or a construct encoding HA-tagged CBP phosphomimetic mutant coexpressing EGFP (DD: pCMV-puro-IRES2-*EGFP mCBP$^{S1383D/S1387D}$*-HA; EE: pCMV-puro-IRES2-*EGFP mCBP$^{S1383E/S1387E}$*-HA) using Lipofectamine™ Stem Transfection Reagent. Transfected cells were detected by EGFP expression (green) and subsequently formed organoids de novo. HA-CBP expression detected by Western blot verified the successful delivery of the DD or EE mutant (Supplementary Fig. 6). **a** Representative images of immunofluorescence staining for versican. The nuclei were counterstained with DAPI. The expression levels were quantified as the integrated density (IntDen) normalized to the DAPI intensity using ImageJ and are presented as arbitrary units (A.U.). $n = 6$ biological replicates for vector, $n = 5$ biological replicates for DD, $n = 6$ biological replicates for EE. **b** Microscopy images of cultured organoids on Day 5 at 10× (upper panels) and 4× (lower panels)

magnification. The relative organoid diameter and formation efficiency (normalized to those of the vector group) were quantified using ImageJ. $n = 6$ biological replicates per group. **c** Representative images of immunofluorescence staining for Ki-67. The nuclei were counterstained with DAPI. The Ki-67$^+$ area was quantified as a percentage of the total DAPI$^+$ area per field using ImageJ. $n = 5$ biological replicates for vector, $n = 7$ biological replicates for DD and EE. **d** Representative images of immunofluorescence staining for E-cadherin. The nuclei were counterstained with DAPI. The expression levels were quantified as IntDen normalized to the DAPI intensity using ImageJ and are presented as A.U. $n = 5$ biological replicates per group. **e** Quantitative RT–PCR analysis of *Vcan* (versican) expression in colonic organoids. Relative expression levels are shown as the fold changes compared with those in the vector group. $n = 3$ biological replicates per group. Data are presented as the means ± SEM. All *p*-values were calculated using an unpaired two-tailed Student's *t*-test and are indicated on the graphs. Scale bars: 100 μm (**a**–**d**). Source data are provided as a Source Data file.

binding EGFR[28]. Its downstream Ras/Raf/Mek/Erk signaling axis has been reported to be active in the crypt epithelium, and this pathway is blocked in Mek-ablated intestinal stem cells[28]. The effect of versican supplementation on increasing $CBP^{AA}$ organoid formation was abrogated by gefitinib, suggesting the involvement of the activated EGFR pathway in mediating the effect of versican on IEC stemness, potentially through the EGF-like motifs within its G3 domain[32].

Abundant CBP–NF-κB (p65) binding was observed in healthy people and $CBP^{WT}$ mice under basal (unstimulated) conditions (Supplementary Fig. 8a, b). In contrast, UC patients and $CBP^{AA}$ mice exhibited reduced CBP–NF-κB (p65) binding (Supplementary Fig. 8a, b). These findings suggest that environmental triggers[3], such as dietary antigens, may increase CBP phosphorylation (Fig. 1i, j), thereby promoting its interaction with NF-κB in normal individuals and $CBP^{WT}$ mice. Upon TNF-α stimulation, only the $CBP^{WT}$ organoids exhibited robust CBP–NF-κB (p65) binding, whereas the $CBP^{AA}$ organoids remained unresponsive (Supplementary Fig. 8c), indicating that UC patients and $CBP^{AA}$ mice are unresponsive to environmental stimulation. Given that CBP phosphorylation is required for efficient binding

to NF-κB (p65), the lack of responsiveness to TNF-α in $CBP^{AA}$ organoids indicates impaired CBP binding to NF-κB in this context. This defect may compromise the NF-κB-mediated protective functions in the intestinal epithelium. Epithelial NF-κB was reported to play a prominent role in protecting against colitis, likely through the expression of antiapoptotic genes in IECs, as the conditional deletion of the NF-κB pathway in IECs led to increased susceptibility to colitis in mice[42]. Furthermore, the IEC-specific loss of NF-κB activation in mice induced by the conditional ablation of intestinal epithelial IKKγ or both IKKα and IKKβ resulted in a proinflammatory phenotype and loss of barrier integrity[43–46]. Therefore, epithelial NF-κB signaling may perform critical "peace-keeping" functions in the colon epithelial barrier by regulating cell survival, barrier integrity, and the immunological and antimicrobial responses of epithelial cells[46,47].

Our $CBP^{AA}$ knock-in mouse model was established through the replacement of Ser1383/1387 with alanines by conventional embryonic stem cell technology. Most well-established colitis mouse models, such as $Il10^{-/-}$ mice, are based on the disruption of immune cell functions. IL-10 knockout ($IL-10^{KO}$) mice develop spontaneous enterocolitis

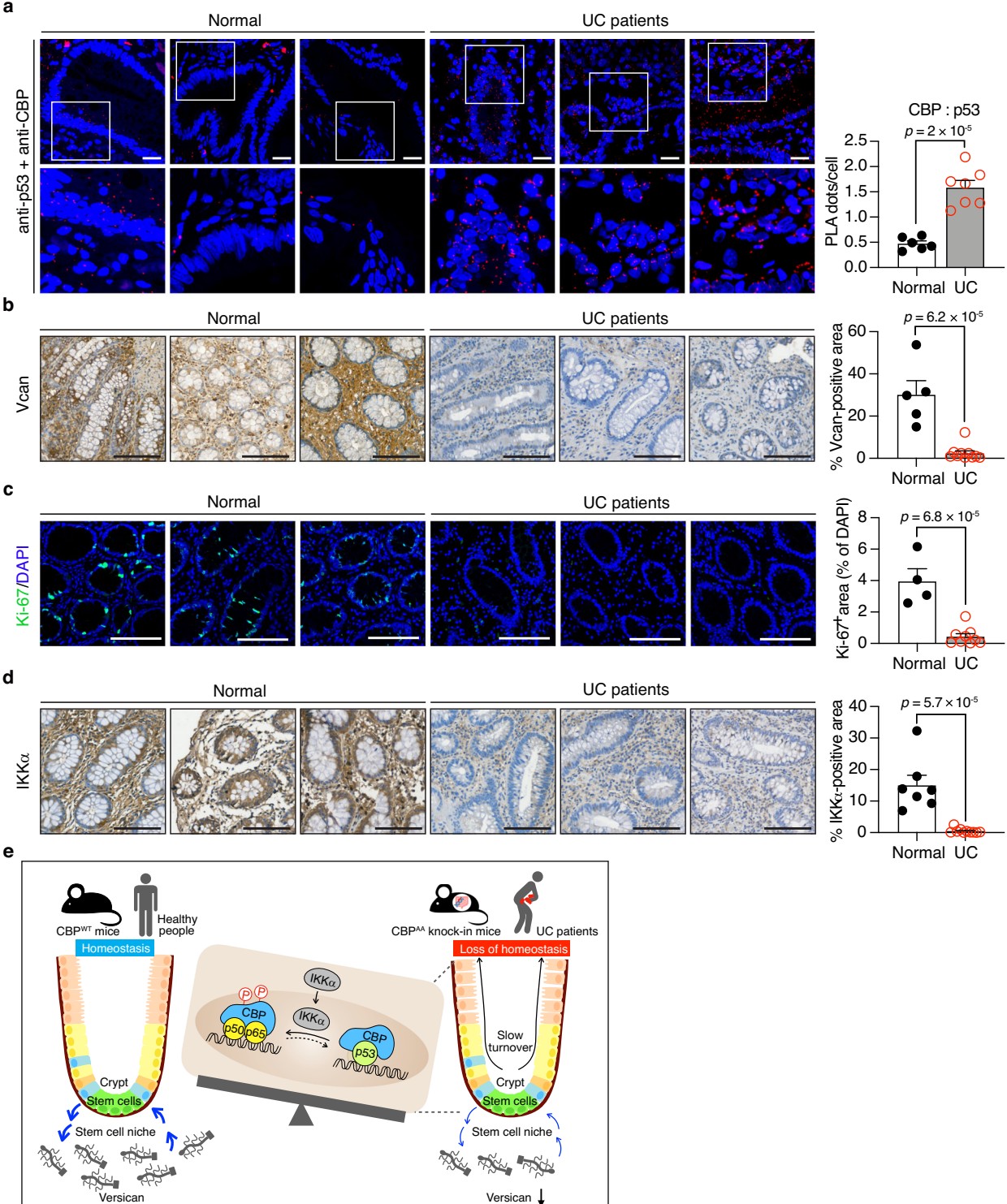

**Fig. 8 | Increased CBP–p53 binding and reduced versican expression in colon tissues from UC patients. a** Representative images and quantification of PLA signals detecting the association between CBP and p53 in colon sections from healthy controls and UC patients. Nuclei were counterstained with DAPI to identify individual cells. PLA signals (red dots) per cell were quantified using ImageJ. $n = 6$ individuals for normal controls, and $n = 7$ individuals for UC patients. **b** Representative images of IHC staining for versican, $n = 5$ individuals for normal controls, and $n = 10$ individuals for UC patients. **c** Representative images of immunofluorescence staining for Ki-67. The nuclei were counterstained with DAPI for normalization. The Ki-67+ area is expressed relative to the total DAPI+ area per field. $n = 4$ individuals for normal controls, and $n = 9$ individuals for UC patients.

**d** Representative images of IHC staining for IKKα. $n = 7$ individuals for normal controls, and $n = 10$ individuals for UC patients. For (**b**–**d**), positively stained areas were quantified as a percentage of a fixed region of interest (ROI) using ImageJ. **e** Schematic summary illustrating how impaired CBP phosphorylation at Ser1383/1387 (human: Ser1382/1386) promotes CBP–p53 binding while reducing its interaction with p65/p50. This shift leads to decreased versican expression in the stem cell niche, thereby disrupting epithelial stemness and crypt turnover, and ultimately contributing to the loss of colonic homeostasis. Data are presented as the means ± SEM. All $p$-values were calculated using an unpaired two-tailed Student's $t$-test and are indicated on the graphs. Scale bars: 20 μm (**a**), 100 μm (**b**–**d**). Source data are provided as a Source Data file.

affecting the duodenum, proximal jejunum, and proximal colon, which resembles Crohn's disease[48]. However, the *CBP^AA* mouse model displays abnormalities primarily in the distal colon, which is attributed to an intrinsic impairment of CBP phosphorylation in colonic epithelial cells. Unlike *IL-10^KO* mice, which develop growth retardation and significant body weight loss[48], *CBP^AA* mice exhibit no obvious developmental or growth defects; thus, this strain is an ideal model for investigating the role of IEC stemness in maintaining gut homeostasis without confounding growth issues.

Our study demonstrates that versican is a key component of the ECM and that its reduction, driven by increased CBP–p53 binding due to impaired CBP phosphorylation, ultimately decreases colonic stemness and affects epithelial homeostasis (Fig. 8e). CBP phosphorylation by IKKα is essential for maintaining versican expression and the stem cell niche and preserving IEC stemness and intestinal homeostasis to potentially prevent IBD development in humans. Additionally, these insights highlight versican as a candidate therapeutic target for IBD. Furthermore, our findings characterize how defective PTMs of CBP contribute to the pathogenesis of IBD. Further research and clinical investigations are needed to evaluate the efficacy of versican-based therapies in restoring IEC stemness for IBD treatment.

## Methods

### Mice

CBP mutation "knock-in" mice (*CBP^AA* mice) were generated using traditional embryonic stem cell technology. These mice were generated via the targeted insertion of the mutated allele (S1383/1387A) at exon 25 with a neomycin selection cassette into mouse embryonic stem cells on a 129 background, followed by subsequent backcrossing to C57BL/6 mice over 10 generations. *CBP^WT* mice were used as littermate controls. The mice were designed and generated by the Transgenic Mouse Models Core of the National Taiwan University Center of Genomic and Precision Medicine. *Lgr5-eGFP-creERT2^{+/Tg}; Villin-Cre* mice and *p53^{f/+}* mice were gifts from Dr. Jr-Wen Shui and Dr. Kai-Chien Yang, respectively. *p53^{f/+}* mice were intercrossed to generate *p53^{f/f}* mice. These mice were then bred to *CBP^AA* mice to generate heterozygous *CBP–p53^{f/+}* mice. Heterozygous *CBP–p53^{f/+}* mice were intercrossed to generate *CBP^{WT}p53^{f/f}* and *CBP^{AA}p53^{f/f}* mice. *CBP^{WT}p53^{f/f}* mice were used as littermate controls. These mice were then bred to *Villin-Cre* mice to generate IEC-specific p53-knockout mice (*CBP^{AA}p53^{ΔIEC}* and *CBP^{WT}p53^{ΔIEC}*) for our experiments. *CBP^{WT}p53^{ΔIEC}* mice were used as littermate controls. The *p53^{ΔIEC}* mice were used to verify whether the colitis phenotype was diminished and whether the decreased stemness of colonic IECs was recovered. The mouse genotypes were determined by the extraction and amplification of DNA from mouse toe tissues (KAPA Mouse Genotyping Kit; KAPA Biosystems, Cape Town, South Africa; KK7352). All three mouse strains were on a C57BL/6 background and were raised and bred under specific pathogen-free conditions at the Laboratory Animal Center, National Taiwan University College of Medicine. Female mice at 8 weeks of age were used in all experiments, unless indicated otherwise. All the mice were housed in isolated ventilated cages (a maximum of five mice per cage) in a barrier facility at the Laboratory Animal Center, National Taiwan University College of Medicine. The mice were maintained on a 12-h light–dark cycle at 22–26 °C, with sterile pellet food and water available ad libitum. All the mice were sacrificed between 10 am and 3 pm. All animal experiments were approved by the National Taiwan University College of Medicine, Institutional Animal Care and Use Committee.

### Cell lines and reagents

The cell lines were cultured at 37 °C with 5% $CO_2$. WT and $p53^{-/-}$ HCT116 cells were obtained from Prof. Chang ZF and grown in DMEM supplemented with 10% FBS with pen/strep. CYL-19s were synthesized by our collaborator, Prof. Cherng-Chyi Tzeng, a medicinal chemist[27], and dissolved in DMSO to prepare a stock solution (50 mM). The recombinant human TNF-α protein was purchased from Peprotech (#300-01A) and dissolved in $ddH_2O$. The recombinant mouse TNF-α protein was purchased from Peprotech (#315-01A) and dissolved in 3% BSA/PBS. The recombinant mouse versican protein (Abbexa, abx069663), produced in a bacterial (*E. coli*) expression system, was dissolved in $ddH_2O$. The final working concentration of 100 ng/mL was selected based on a recent study[49] and further supported by our preliminary dose–response tests (50, 100, and 200 ng/mL), in which 100 ng/mL elicited a robust biological effect with no additional increase observed at 200 ng/mL. The recombinant mouse epidermal growth factor (mEGF) protein was purchased from Peprotech (#315-09) and dissolved in 0.1% BSA/PBS. Gefitinib was purchased from LC Laboratories (G-4408) and dissolved in DMSO.

### Construction of the targeting vector and genotyping of mice expressing the mutant CBP allele

A construct containing the mutated exon 25 was used to target the mouse *CBP* gene. A targeting vector was generated by the recombineering method to achieve the knock-in of this allele[50,51]. Briefly, a 9.6 kb fragment of mouse genomic DNA containing the region from exon 25 of the mouse *CBP* gene was retrieved from a B6 strain-derived bacterial artificial chromosome clone (bMQ451N12, Children's Hospital Oakland Research Institute). The fragment was inserted at the *Not*I–*Spe*I sites of pL253, in which the *MC1-TK* (thymidine kinase) gene served as a negative selection marker. The resulting construct was used as a backbone for *CBP* and a *neo* cassette (5′–*loxP–PGK–Neo*bpA–*loxP*-3′) from pL451 downstream of exon 25. The final targeting construct contained a homologous 5′ arm of 4.7 kb and a 3′ arm of 4.9 kb.

The primer sequences used for PCR to generate the 300-bp homologous arms for the retrieval of exon 25 of mouse *CBP* genomic DNA (AU, BD, YU and ZD) with the insertion of *loxP* and *FRT* sites (CU, DD, EU and FD) and the *neo* cassette were as follows: *CBP*-KI AU 114200, 5′-ATAGCGG CCGCAGTAACTTGAAGTGCTAATATC-3′; *CBP*-KI BD 114500, 5′-ATAAAGCTTATACTTAA GAGAACAAAGTAATTACTGCTAG-3′; *CBP*-KI CU 118130, 5′-ATAGTCGACTCCATTTTA AGTGAGCCTAGG-3′; *CBP*-KI DD 118890, 5′-ATATCTAGAAAAGAAAGTCACATGCTAGC-3′; *CBP*-KI EU 119300, 5′-ATAGGGCCCTAAGCTGTATATTTTGGTGCTAG-3′; *CBP*-KI FD 120000, 5′-ATAGCGGCCGCAAAAGCAGAGTAGA-CACTGTG-3′; *CBP*-KI YU 123900, 5′-ATAAAGCTTTAGAA-CATTGATGGTGAAG-3′; and *CBP*-KI ZD 124200, 5′-ATAACTAGTGTA CTACAGCGCCCATACAG-3′. The knock-in targeting vector was confirmed by sequencing, linearized by DNA digestion at the unique *Not*I site, and electroporated into C57BL/6N ES cells. Correctly targeted ES cell clones were selected by culture with G418 (240 μg/mL) and ganciclovir (2 μM). Correctly targeted clones were identified by Southern blotting to confirm homologous recombination at both 5′ and 3′ arms. Validated ES cells were subsequently injected into blastocytes of C57BL/6J origin (National Laboratory Animal Breeding and Research Center, Taiwan) to establish knock-in mice.

The mouse genotypes were determined by the extraction and amplification of DNA from mouse toe tissues (KAPA Mouse Genotyping Kit; KAPA Biosystems). Routine genotyping was performed by PCR with the primer 5′-CCCTTCCCTAAGAAACAGGACT-3′ (*Fok*I-F) paired with 5′-CCCTTCCTCCCCTAAGAGTTTA-3′ (*Fok*I-R) for the wild-type (WT) *CBP* allele or paired with 5′-CCAAAATATACAGCTTAGGGCC-3′ (AA-R) for the mutant *CBP* allele. The PCR conditions were 95 °C for 3 min; 35 cycles of 95 °C for 15 s, 60 °C for 15 s and 72 °C for 15 s; and 72 °C for 1 min.

### Assessment of colon length, stool consistency, and occult blood in stool under baseline conditions

Colon length was measured from the ileocecal junction to the anus using a plastic ruler. Disease severity was evaluated by an observer blinded to the experimental groups based on stool consistency and fecal occult blood scores. Stool consistency was assessed by pressing a

stick onto the feces and scored as follows: 0, well-formed pellets; 1, semi-formed stools; 2, pasty stools; and 3, liquid stools adhering to the anus. Fecal occult blood was determined using a Hemoccult Sensa test (Beckman Coulter, #64151) and scored as follows: 0, negative hemoccult; 1, positive hemoccult; 2, positive hemoccult with visible traces of blood; and 3, gross rectal bleeding. For intermediate severity, half-scores (e.g., 0.5, 1.5) were assigned.

## Histological evaluation and scoring

Histological scoring was performed in a blinded manner by a trained veterinary pathologist, Yi-Ting Tsai (Laboratory Animal Center, National Taiwan University College of Medicine, Taipei, Taiwan). For tissue preparation, a 2-cm colon fragment was taken from the distal colon above the anus and submerged in a buffered 10% formalin solution. Hematoxylin and eosin (H&E) staining was performed on 5-μm sections from 10% formalin-fixed, paraffin-embedded samples using the appropriate procedures. Scoring was based on seven parameters: the severity of inflammation, location of the inflammatory lesion, crypt architecture, lesion area in the crypts, mucosal hyperplasia, goblet cell depletion, and crypt dilation and abscesses. Each parameter was scored on a scale from 0 to 3, where 0 indicated normal tissue and 3 indicated severe pathology. For example, the severity of inflammation was scored as 0 (normal), 1 (slight infiltration of mononuclear cells and neutrophils, <30%), 2 (moderate infiltration, 30–60%), or 3 (severe infiltration, >60%). The lesion location was scored as 0 (normal), 1 (mucosa), 2 (mucosa and submucosa), or 3 (all layers of the bowel wall). The remaining parameters were similarly graded based on the severity and extent of tissue involvement. Representative images were captured using a Leica DM2500 upright microscope equipped with MicroVisioneer Manual Whole-Slide Imaging Software.

## Assessment of ex vivo intestinal permeability

A 2-cm segment of colon tissue was opened longitudinally along the mesenteric border and mounted in a Ussing chamber (World Precision Instruments, Sarasota, FL). The tissue in each chamber was exposed to 5 mL of circulating oxygenated Krebs buffer (37 °C) containing either glucose (10 mM, serosal buffer) or mannitol (10 mM, mucosal buffer). Fluorescein isothiocyanate (FITC)-dextran (12.5 mM; molecular weight, 3000–5000 Da; Sigma–Aldrich, FD4) was added to the mucosal buffer at a final concentration of 500 μM. Samples were collected from the serosal buffer at different time points (0, 30, 60, 90, and 120 min) and measured using a multimode plate reader (Paradigm Detection Platform; Beckman Coulter, Indianapolis, IN) at an excitation wavelength of 485 nm and an emission wavelength of 535 nm.

## Assessment of intestinal permeability in vivo

The mice were starved for 4–6 h and then orally gavaged with 0.5 mg/g body weight FITC-dextran (molecular weight, 3000–5000 Da; Sigma–Aldrich, FD4). After 4 h, 300–400 μL of blood was collected by cardiac puncture and centrifuged at $900 \times g$ for 20 min at room temperature. Serum was diluted with an equal volume of PBS, and the serum FITC-dextran concentrations were measured with a multimode plate reader (Paradigm Detection Platform; Beckman Coulter) at an excitation wavelength of 485 nm and an emission wavelength of 535 nm using a FITC-dextran standard curve (0, 0.125, 0.25, 0.5, 1, 2, 4, 6, 8, and 12.5 μg/mL). Serum from mice not administered FITC-dextran was used as the background sample.

## Immunohistochemical (IHC) assay

Tissues were fixed in 10% neutral buffered formalin, embedded in paraffin, and sectioned at 4 μm thickness. Immunohistochemistry (IHC) staining was performed using the TnAlink Polymer Detection System (BioTnA, #TAHC04D) according to the manufacturer's instructions. Heat-induced epitope retrieval (HIER) was performed using Tris-EDTA

HIER Solution (pH 9.0; ScyTek, #TES500). The following primary antibodies were used: anti-pCBP (Ser1383/1387 for mouse samples; Ser1382/1386 for human samples; 1:100; GeneTex, customized, GTX90661), anti-CBP (1:50; Cell Signaling Technology, #7389), anti-F4/80 (1:250; Cell Signaling Technology, #70076), anti-Ly6G (1:50; R&D, MAB1037), anti-p53 (1:100; Santa Cruz Biotechnology, sc-126), anti-p21 (1:100; Cell Signaling Technology, #2947), anti-versican (1:100; Abcam, ab19345) and anti-IKKα (1:100; Santa Cruz Biotechnology, sc-7606). Whole-slide images were acquired with a TissueFAXS microscopy system (TissueGnostics GmbH, Vienna, Austria). Quantification of positive staining was performed using ImageJ as described in the corresponding figure legends.

## Generation of bone marrow chimeras

Bone marrow transplantation was performed using bone marrow cells isolated from donor mice, which were subsequently transplanted into γ-irradiated recipient mice. B6.SJL-$Ptprc^{\alpha}$ $pepc^{\beta}$/BoyJ mice expressing the CD45.1 alleles were used as WT mice; $CBP^{AA}$ mice (C57BL/6J strain) expressed the CD45.2 allele. Recipient mice were sublethally irradiated with 6 Gy from a cesium source two times at 4 h intervals on the day before transplantation. Bone marrow cells were isolated from the femurs and tibias of the donor mice, and a total of $5 \times 10^6$ bone marrow cells were injected into the tail veins of the irradiated recipient mice. The mice were evaluated and sacrificed 6 weeks after reconstitution. Bone marrow reconstitution was evaluated by staining mLNs with anti-CD45.1 (1:200; BUV395; BD Biosciences, #565212) and anti-CD45.2 (1:200; BUV737; BD Biosciences, #612778) antibodies and performing flow cytometry (LSRFortessa, BD).

## Frozen sections for immunostaining

The distal colon (approximately 2 cm from the anus) was collected, flushed with cold PBS to remove fecal content, and fixed in 4% paraformaldehyde (PFA) for 30 min at room temperature. The tissues were then dehydrated in 15% (w/v) sucrose in PBS until they sank (6–12 h), followed by 30% (w/v) sucrose in PBS overnight at 4 °C. Subsequently, the colon segments were embedded in FSC 22 Frozen Section Media (Leica Biosystems, #3801480) and frozen at −80 °C.

Frozen sections were cut at 5 μm thickness. Sections were permeabilized with 0.25% Triton X-100 for 20 min and blocked with 5% bovine serum albumin (BSA) in PBS for 1 h at room temperature. After incubation with primary antibodies overnight at 4 °C and secondary antibodies for 1 h at room temperature, slides were mounted using ProLong Gold Antifade Mountant with DAPI (Invitrogen, P36935). Confocal laser scanning microscopy was performed on a Zeiss LSM 780 confocal microscope (Carl Zeiss, Jena, Germany). Quantification of positive staining was performed as described in the corresponding figure legends.

The following primary antibodies were used: anti-pCBP (Ser1383/1387 for mouse, Ser1382/1386 for human; 1:150; GeneTex, customized, GTX90661), anti-CBP (1:200; Cell Signaling Technology, #7389), anti-versican (1:200; Abcam, ab19345), anti-Ki-67 (1:200; Cell Signaling Technology, #9129S), anti-E-cadherin (1:200; Invitrogen, #33-4000), and anti-p21 (1:150; Cell Signaling Technology, #2947).

The following secondary antibodies were purchased from Invitrogen and used at a 1:500 dilution: Alexa Fluor® 488-conjugated goat anti-rabbit IgG (A-11034), Alexa Fluor® 488-conjugated goat anti-mouse IgG (A-11029), Alexa Fluor® 594-conjugated goat anti-rabbit IgG (A-11037), and Alexa Fluor® 594-conjugated goat anti-mouse IgG (A-11005).

## Confocal microscopy and image analysis

Confocal imaging was performed using a Zeiss LSM 780 or LSM 880 confocal laser scanning microscope (Carl Zeiss, Jena, Germany) equipped with 405 nm, 488 nm, and 561 nm laser lines and detectors, operated with ZEN software. All images were acquired using a 20×/0.8 NA Plan-Apochromat objective in sequential scanning mode to

eliminate spectral crosstalk. Within each comparative experimental set, core acquisition parameters were maintained consistently, including pixel resolution (1024 × 1024), pixel dwell time, and pinhole size. To ensure quantitative accuracy, acquisition parameters (e.g., laser power and detector gain) were optimized per staining batch to maintain the fluorescence signal within the linear dynamic range without pixel saturation, and were carefully controlled and kept within a comparable range across all samples within the same comparative experiment. For three-dimensional structures such as organoids, Z-stack images were acquired with optimal step sizes automatically calculated by the ZEN software based on the objective numerical aperture. Image processing and quantification were performed using ImageJ/Fiji. Any adjustments to brightness or contrast were applied uniformly across images within the same experiment and were performed solely for visualization purposes. No nonlinear image manipulation was applied. The original raw image files (.czi) and complete acquisition metadata are available from the corresponding author upon request due to patient privacy laws regarding non-de-identifiable metadata.

### Pulse-chase BrdU labeling and IHC detection of BrdU staining

The mice were injected intraperitoneally with 100 mg/kg body weight 5-bromo-2′-deoxyuridine (BrdU; Sigma–Aldrich, St. Louis, MO; B5002) at 2 h, 24 h, 48 h, or 96 h before sacrifice. The colon was excised from each mouse, fixed overnight with Formalde-Fresh solution (Fisher Scientific, SF-94), and embedded in paraffin. Colon sections were cut at a thickness of 5 µm for IHC staining using a TAlink Polymer Detection System (BioTnA; Kaohsiung, Taiwan; TAHC04D) and probed with an anti-BrdU antibody (1:100; Abcam, ab6326). A flash label at 2 h was used to designate the baseline position of the proliferating crypt cells. The migration of BrdU-positive (BrdU$^+$) cells from the crypt base to the lumen was measured using ImageJ software.

### CUT&RUN assay for CBP and p53 enrichments at the *p21* promoter

The CUT&RUN assay was performed using the CUT&RUN Assay Kit (Cell Signaling Technology, #86652) according to the manufacturer's protocol, with minor modifications. Briefly, $1–3 × 10^5$ freshly isolated colonic crypt cells were fixed with 0.1% formaldehyde, permeabilized with digitonin-containing buffer, and incubated with anti-CBP (1:50; Cell Signaling Technology, #7389) and anti-p53 (1:50; Cell Signaling Technology, #2524) antibodies to capture CBP-bound and p53-bound chromatin, respectively. Nuclei were immobilized with concanavalin A-coated magnetic beads. Targeted chromatin was cleaved by Protein A–micrococcal nuclease (pA-MNase) upon calcium exposure and released at 37 °C. Purified DNA was analyzed by qPCR using primers specific for the p53 binding site within the mouse p21 promoter (forward: 5′-CCACAGCAGAGGGAGAAAGAAG-3′; reverse: 5′-GCTGCTCA-GAGTCTGGAAATC-3′). Enrichment was calculated as % input, defined as $100 × 2^{[Ct(input) – Ct(CUT\&RUN)]}$ for each sample. qPCR was performed with SYBR Green (Thermo Fisher Scientific, A46109) on the ABI Quant-Studio 5 Real-Time PCR System (Thermo Fisher Scientific). All reactions were run in at least three biological replicates with corresponding matched input controls.

### Colonic crypt isolation and organoid culture

The colons of 8-week-old female *CBP$^{WT}$* and *CBP$^{AA}$* mice were opened longitudinally. The tissue was minced into 5-mm pieces, washed three times with cold PBS, and incubated with 30 mM EDTA in PBS for 5 min at 37 °C in a tube roller at 100 rpm. After the incubation, the tissue fragments were vortexed three times in 10 mL of cold PBS containing 0.1% bovine serum albumin (BSA), and the resulting supernatant, which contained few crypts and debris, was discarded. The tissue pieces were resuspended in 10 mL of cold PBS containing 0.1% BSA and vortexed eight times, and the supernatant enriched in crypts was

collected in a clean 50 mL tube. After crypt-enriched supernatants were collected by vortexing three to four times, the samples were centrifuged at 50 × g for 5 min at 4 °C to pellet intact crypt fragments. After the crypts were resuspended in 1–10 mL of cold PBS containing 0.1% BSA, the crypts were counted (400 crypts/well), transferred to 1.5 mL Eppendorf tubes, and centrifuged at 9600 × g for 10 s. The supernatant was discarded, and 0.5 mL of TrypLE (Gibco, #12563-011) containing 10 µM Y-27632 (LC Laboratories, Y-5301) was then added and incubated at 37 °C for 8 min. After the incubation, the crypts were pipetted fifteen times and centrifuged at 9600 × g for 10 s. For culture, the isolated crypts were resuspended in Matrigel (Corning, #356231) (40 µL/well) and seeded in prewarmed 24-well plates. The polymerized Matrigel was overlaid with organoid growth medium (1:1 mixture of basal culture medium and WNR conditioned medium). The basal culture medium was Advanced DMEM/F12 (Gibco, #12634-034) supplemented with penicillin/streptomycin (Gibco, 15140-122), GlutaMAX (Gibco, 35050-061), 10 mM HEPES (Gibco, #15630-080), N2 (Gibco, #17502-048), B27 (Gibco, #17504-044), 1.25 mM N-acetylcysteine (Sigma–Aldrich, A9165-5G), and 10 mM nicotinamide (Sigma–Aldrich, N0636). WNR conditioned medium containing WNT, Noggin, and R-spondin derived from stable cell lines was a kind gift from Dr. Jr-Wen Shui and supplemented with 10 µM Y-27632 (a ROCK inhibitor) and 50 ng/mL mEGF. The overlay medium was changed every 3 days, and the organoids were harvested on Day 6. The organoid-forming efficiency was calculated as the number of organoids formed in the Day 6 culture divided by the total number of isolated crypts seeded in the well and presented as a percentage.

### Organoid viability assay

Organoid viability was assessed using an MTT assay[52]. Crypt-derived organoids from 8-week-old female mice were seeded into 96-well plates at a density of 200 crypts/well in 7 µL of Matrigel. After 6 days of organoid culture, 0.5 mg/mL MTT (Sigma–Aldrich, M2128) was added to each well, and the cells were further incubated for 3 h at 37 °C. After the incubation, the medium was discarded, and 20 µL of a 2% sodium dodecyl sulfate (SDS; Invitrogen, #15525017) solution in ddH$_2$O was added to solubilize the Matrigel at 37 °C for 2 h. Then, the purple–blue MTT formazan precipitate was dissolved in 100 µL of DMSO for 2 h 20 min at 37 °C, and the absorbance was measured at 562 nm in a microplate reader (BioTek Synergy HT, USA). Cell viability was normalized to that of the *CBP$^{WT}$* organoids by dividing the OD$_{562}$ value of each sample by the mean OD$_{562}$ of the *CBP$^{WT}$* controls.

### Whole-mount staining of organoids

Crypt-derived organoids generated from 8-week-old female mice were seeded in 8-well chamber slides (Merck, PEZGS0816) or 8-well chambered coverglasses (Thermo Scientific, #155411) at a density of 40 crypts/well in 40 µL of Matrigel. After 6 days of organoid culture, the organoids were fixed with 4% paraformaldehyde for 30 min at room temperature on an orbital shaker at 50 rpm, permeabilized with TBS containing 0.1% Triton X-100 and 1% BSA for 30 min at room temperature, immersed in acid–alcohol for 10 min at −20 °C, and blocked with TBS containing 0.1% Triton X-100 and 1% BSA for 1 h at room temperature. Organoids were incubated with primary antibodies overnight at 4 °C. The primary antibodies used included anti-pCBP (Ser1383/1387) (1:150; GeneTex, customized, GTX90661), anti-CBP (1:200; Cell Signaling Technology, #7389), anti-Ki-67 (1:200; Cell Signaling Technology, #9129S), and anti-E-cadherin (1:200; Invitrogen, #33-4000). After the incubation with the primary antibodies, the organoids were incubated with secondary antibodies at a 1:500 dilution, including Alexa Fluor® 488-conjugated goat anti-mouse IgG (Invitrogen, A-11029) and Alexa Fluor® 594-conjugated goat anti-rabbit IgG (Invitrogen, A-11037), for 30 min at room temperature and then with ProLong Gold Antifade Reagent with DAPI (Invitrogen, P36935) for nuclear staining. Images of most whole-mount sections were

obtained via z-stack reconstruction using a Zeiss LSM 880 or LSM 780 confocal microscope (Carl Zeiss, Jena, Germany). Images were acquired and analyzed as described in the confocal microscopy and image analysis section.

## Molecular cloning and plasmid construction

The plasmids utilized in this study, including a control vector expressing *EGFP* alone (pCMV-puro-IRES2-*EGFP*) or a construct encoding HA-tagged *CBP* phosphomimetic mutants coexpressing *EGFP* (DD mutant: pCMV-puro-IRES2-*EGFP mCBP^{S1383D/S1387D}*-HA; EE mutant: pCMV-puro-IRES2-*EGFP mCBP^{S1383E/S1387E}*-HA), were all developed at the Biomedical Resource Core (BMRC) of the First Core Labs, National Taiwan University College of Medicine. All plasmids have a total size of 14,236 bp and utilize the *Mus musculus* CREB-binding protein (*Crebbp*; NCBI Reference Sequence: NM_001025432.1) as the backbone. To ensure proper expression and facilitate cloning, a silent mutation (V292V; c.876G>T) was introduced into the *mCbp* sequence. For the phosphomimetic constructs, site-directed mutagenesis was performed to introduce the following amino acid substitutions:

- S1383D/S1387D (DD): c.4147T>G, c.4148C>A (S1383D) and c.4159T>G, c.4160C>A, c.4161G>C (S1387D).
- S1383E/S1387E (EE): c.4147T>G, c.4148C>A, c.4149T>A (S1383E) and c.4159T>G, c.4160C>A (S1387E).

The oligonucleotide sequences for cloning and sequencing verification are provided in Supplementary Table 2. All constructs were validated by Sanger sequencing through the BMRC facility to confirm the presence of intended mutations and the integrity of the plasmid backbone.

## Lipofectamine-mediated plasmid transfection of organoids dissociated into single cells

Lipofectamine transfection of the organoids was performed with modifications to the established protocol[53]. Briefly, organoids that were dissociated into single cells were transfected with plasmids using Lipofectamine and subsequently re-embedded in Matrigel for organoid reformation. The detailed steps are described below.

The culture medium was removed, and the organoids were briefly washed with prewarmed PBS. The organoids were then collected in cold PBS and centrifuged at $9600 \times g$ for 10 s. After the PBS was aspirated, 500 µL of prewarmed TrypLE (Gibco, #12563-011) containing 10 µM Y-27632 (LC Laboratories, Y-5301) was added to the tube. Organoids were mechanically disrupted by pipetting up and down ten times using a P1000 pipette and then incubated at 37 °C for 5 min. Following this incubation, further dissociation was performed by pipetting up and down approximately 20 times using a P200 tip. The enzymatic reaction was terminated by the addition of cold Advanced DMEM/F12 supplemented with GlutaMAX, HEPES, and penicillin–streptomycin. The cells were pelleted by centrifugation at $9600 \times g$ for 10 s.

Approximately $10^5$ cells were resuspended in 450 µL of organoid growth medium (as described in the "colonic crypt isolation and organoid culture" section) supplemented with 10 µM Y-27632 and kept on ice until transfection. One microgram of plasmid DNA (i.e., pCMV-puro-IRES2-*EGFP* (vector), pCMV-puro-IRES2-*EGFP mCBP^{S1383D/S1387D}*-HA (DD), or pCMV-puro-IRES2-*EGFP mCBP^{S1383E/S1387E}*-HA (EE)) was mixed with 1 µL of Lipofectamine Stem Transfection Reagent (Thermo Fisher Scientific, STEM00001) according to the manufacturer's protocol. Fifty microliters of reaction mixture were added to $10^5$ single organoid cells suspended in 450 µL of organoid growth medium supplemented with 10 µM Y-27632 (without Matrigel) in a single well of a 24-well plate. The plate was then centrifuged at $600 \times g$ for 1 h at 32 °C, followed by an incubation at 37 °C for 2 h. The cells were then harvested, pelleted, seeded in Matrigel in two wells of a 24-well plate, and grown in organoid growth medium supplemented with 10 µM Y-27632.

The overlay medium was changed every 3 days, and the organoids were harvested on Day 5 post-transfection.

Transfected cells were identified by EGFP fluorescence. Successful delivery and expression of DD and EE mutants were confirmed by Western blotting detection of HA-CBP (Supplementary Fig. 5).

## Proximity ligation assay (PLA)

PLA was performed according to the manufacturer's instructions (Sigma–Aldrich, #DUO92101). Primary antibodies were diluted in Duolink antibody diluent as follows: anti-CBP (Cell Signaling Technology, #7389) at 1:200, anti-p65 (Cell Signaling Technology, #6956) at 1:200, and anti-p53 (Santa Cruz Biotechnology, sc-126) at 1:200. Imaging was performed with a Zeiss LSM 780 confocal microscope (Carl Zeiss, Jena, Germany).

## Western blot analysis

Cells or organoids were lysed on ice for 30 min using RIPA buffer (Thermo Scientific, #89901) supplemented with Protease and Phosphatase Inhibitor Cocktail (Thermo Scientific, #78442) at a 1:100 dilution. Cell lysates were then centrifuged at $16,200 \times g$ for 20 min at 4 °C. The supernatant was collected, and protein concentration was determined using Bio-Rad Protein Assay Dye Reagent (Bio-Rad Laboratories, #500-0006). Equal amounts of protein (30 µg) were resolved on 6–10% SDS-PAGE gels and then transferred to nitrocellulose membranes (Thermo Scientific, #88018). After transfer, membranes were blocked for 1 h at room temperature with 5% (w/v) non-fat milk in TTBS (50 mM Tris-HCl, pH 7.4, 150 mM NaCl, and 0.05% Tween 20). The membranes were then incubated with the appropriate primary antibodies overnight at 4 °C. Following washing, membranes were incubated with horseradish peroxidase (HRP)-conjugated secondary antibodies for 30 min at room temperature. Immunoblots were visualized using the T-Pro LumiDura Chemiluminescence Detection Kit (T-Pro Biotechnology, #JT96-K006M). The following primary antibodies were used: anti-p53 (1:2000; Santa Cruz Biotechnology, sc-126; Leica, NCL-L-p53-CM5p), anti-p21 (1:1000; Santa Cruz Biotechnology, sc-6246; Cell Signaling Technology, #2947), anti-ALDH1A1 (1:1000; Cell Signaling Technology, #12035), anti-HA-tag (1:1000; Cell Signaling Technology, #3724), anti-Lgr5 (1:1000; Abcam, ab75850), anti-vinculin (1:1000; Santa Cruz Biotechnology, sc-25336), anti-GAPDH (1:2000; Proteintech, #60004-1-Ig), and anti-actin (1:5000; Proteintech, #66009-1-Ig). HRP-conjugated secondary antibodies included anti-mouse IgG (#7076S, 1:10,000) and anti-rabbit IgG (#7074S, 1:10,000) from Cell Signaling Technology.

## Spheroid formation

HCT116 cells ($4 \times 10^3$ cells/mL in 6-well plates) were cultured in serum-free DMEM/F12 supplemented with 1% antibiotic/antimycotic Solution, 1% insulin–transferrin–selenium, 20 ng/mL EGF, and 25 ng/mL bFGF. After 5 days, the formation of spheres was evaluated.

## Reverse transcription and cDNA preparation

Total RNA was isolated from various samples, including colonic crypts, cultured colonic organoids, and T cells of the indicated mice, using TRIzol reagent (Thermo Scientific, #15596018) according to the manufacturer's protocol. Briefly, tissues were homogenized in TRIzol, and phase separation was induced by chloroform addition. RNA was then precipitated with isopropanol, washed with 75% ethanol, and finally resuspended in nuclease-free water. For cDNA synthesis, 2 µg of total RNA from each sample was mixed with 1 µL of Oligo(dT)$_{20}$ primer and nuclease-free water to a total volume of 15 µL. The mixture was incubated at 70 °C for 5 min to denature secondary structures within the RNA template, followed by immediate cooling to 4 °C for at least 1 min to prevent renaturation. Subsequently, 10 µL of Master Mix containing M-MLV RT 5× Reaction Buffer (5 µL, Promega, #M531A), 10 mM dNTP (1 µL, FocusBio, #BTCP-10M), M-MLV Reverse Transcriptase (1 µL, Promega, #M1705), and RNase inhibitor (0.625 µL, Promega, #N2518)

was added to each tube. The reverse transcription reaction was performed at 42 °C for 1 h using a Biometra T3 thermocycler (Biometra). The resulting cDNA was stored at −20 °C until further use.

## Real-time PCR and primer sequences

Real-time PCR was performed using Applied Biosystems (ABI) Power-Track SYBR Green Master Mix (Thermo Fisher Scientific, #A46109) on ABI QuantStudio 5 Real-Time PCR System (Thermo Fisher Scientific). All samples were analyzed in technical duplicate, and target gene expression levels were normalized to *Actin* or *Gapdh* using the $2^{-\Delta\Delta C_T}$ method. The mouse primer sequences used were as follows:

- Figures 6d, 6e and 7e: *Vcan* (5′-GGACCAAGTTCCACCCTGACAT-3′ and 5′-CTTCACTGCCAAGGTTCCTCTTCT-3′); *Actin* (5′-CATTGCT-GACAGGATGCAGAAGG-3′ and 5′-TGCTGGAAGGTGGACAGTGA GG-3′).
- Supplementary Figs. 2 and 3: *Tbx21* (5′-CAA-CAACCCCTTTGCCAAAG-3′ and 5′-TCCCCCAAGCAGTTGACAGT-3′); *Ifng* (5′-CAAGTGGCATAGATGTGGAAG-3′ and 5′-GAA-GAAGGTAGTAATCAGGTGTG-3′); *Gata3* (5′-AGAACCGGCCCCT-TATCAA-3′ and 5′-AGTTCGCGCAGGATGTCC-3′); *Cmaf* (5′-GAGCAGCGACAACCCTTC-3′ and 5′-CCCACGGAGCATTTAA-CAAG-3′); *Il4* (5′-TAGTTGTCATCCTGCTCTTCTT-3′ and 5′-GTGTTCTTCGTTGCTGTGAG-3′); *Rorc* (5′-CGCCTCACCTGACC-TACC-3′ and 5′-TTGCCTCGTTCTGGACTATAC-3′); *Il17* (5′-AGG-CAGCAGCGATCATCC-3′ and 5′-TGGAACGGTTGAGGTAGTCTG-3′); and *Il2rb* (5′-GTCCATGCCAAGTCGAACCT-3′ and 5′-GGATGCCTGCCTCACAAGAG-3′); and *Gapdh* (5′-CATCACTGC-CACCCAGAAGACTG-3′ and 5′-ATGCCAGTGAGCT TCCCGTTCA G-3′).

## RNA sequencing and bioinformatics

Total RNA samples (3–5 µg) were sent to Genomics BioSci & Tech (New Taipei City, Taiwan) for mRNA enrichment using oligo(dT) magnetic beads, and indexed dsDNA libraries were constructed using the Illumina TruSeq RNA Sample Preparation Kit (Illumina) for further processing and bioinformatics analysis.

## Flow cytometry analysis

CD4⁺ and CD8⁺ T cells were isolated from the spleens, peripheral lymph nodes, and mesenteric lymph nodes (mLNs) of *CBP^WT* and *CBP^AA* mice and were then stained with anti-CD4 (1:100; PE; eBioscience, #12-0042-82), anti-CD8α (1:200; APC; BioLegend, #100711), and anti-TCR-β (1:100; FITC; BioLegend, #109205) antibodies and gated on CD4⁺ T cells (TCR-β⁺CD4⁺) and CD8⁺ T cells (TCR-β⁺CD8⁺). Naïve CD4⁺ T cells were isolated from the peripheral lymph nodes of *CBP^WT* and *CBP^AA* mice, stained with anti-CD4 (1:200; FITC; BioLegend, #100510) and anti-CD62L (1:200; PE; BioLegend, #104407) antibodies, and gated on naïve CD4⁺ T cells (CD4⁺CD62L^hi). The cells were analyzed by flow cytometry (FACSAria III, BD). Representative gating strategies and isotype controls for splenic CD4⁺ and CD8⁺ T cell characterization are provided in Supplementary Figs. 9 and 10, respectively. Data analysis and gating were performed using FlowJo software.

## Isolation of colonic epithelial cells from crypts for flow cytometry analysis

The colons of *Lgr5-eGFP* reporter mice were removed, cut longitudinally, and washed five times with cold PBS. Following an incubation for 5 min at 37 °C in a 30 mM EDTA solution, the colonic crypts were released from the colon tissue by mechanical vortexing in wash buffer (0.1% BSA in PBS), and this process was repeated three times. The supernatant, which was enriched with colonic crypts, was centrifuged 2 times at 80 × *g* and 40 × *g* for 10 min at 4 °C. Isolated colonic crypts were resuspended in 5 mL of TrypLE Express supplemented with Y-27632 (1:1000 stock dilution) and transferred into C-tubes.

Using the program m-intestine-1, the crypts were dissociated with the GentleMACS dissociator at room temperature, and the c-tube was incubated for 8 min in a water bath at 37 °C. After the incubation, the program m-intestine-1 was run again. Dissociated crypt cells were resuspended in organoid medium and filtered through a 40-µm cell strainer, and colonic single epithelial cells were analyzed by flow cytometry (LSRFortessa, BD) through the service provided by the Flow Cytometric Analysis and Sorting Core (the First Core Laboratory, National Taiwan University College of Medicine). The sequential gating hierarchy for the isolation of colonic *Lgr5-GFP⁺* stem cells, including debris removal and doublet exclusion, is detailed in Supplementary Fig. 11. *Lgr5-GFP⁺* cells were gated using GFP-negative crypt cells (from mice without the *Lgr5-eGFP* reporter) as a negative control to define background autofluorescence. Data analysis and gating were performed using BD FACSDiva software.

## Statistical analysis

The sample size (*n*) refers to the number of independent biological replicates in each group, specifically representing individual mice or individual human donors/patients as indicated in the respective figure legends. For all organoid experiments, *n* denotes independent biological replicates, with each replicate representing an organoid line established from a separate donor mouse. For organoid diameter and organoid-forming efficiency experiments, each data point represents the average value derived from at least three technical replicates (wells) per mouse. To ensure reproducibility and minimize batch effects, all samples within a comparative set were processed, stained, and imaged in parallel. Data were analyzed using unpaired two-tailed Student's *t*- tests with GraphPad Prism 10 software. Data distribution was assumed to be normal, and variances were similar between groups being compared. All quantitative data are presented as mean ± SEM. Statistical significance was defined as $p < 0.05$, with exact *p*-values provided directly on the graphs.

## Study approval

Human tissue samples were obtained with informed consent from all participants, and the study protocol was approved by the Institutional Review Board (IRB) of National Taiwan University Hospital. All studies involving human paraffin-embedded colon sections were conducted under the approval of the IRB (Nos. 201705031RIND and 202404149RINC). All animal experiments were approved by the National Taiwan University College of Medicine Institutional Animal Care and Use Committee (IACUC; Nos. 20170357, 20170454, 20200054, 20210223, 20220486, 20230281, and 20240106).

## Reporting summary

Further information on the research design is available in the Nature Portfolio Reporting Summary linked to this article.

## Data availability

The RNA sequencing data generated in this study have been deposited in the Gene Expression Omnibus (GEO) database under accession code GSE295556. The processed figures and source data are available in the Figshare database under https://doi.org/10.6084/m9.figshare. 28837331. The raw confocal immunofluorescence images are protected and are not available in a public repository due to data privacy laws, as the metadata contains non-de-identifiable patient identifiers; however, these data are available from the corresponding author upon request. The source data generated in this study are provided in the Source Data file. Source data are provided with this paper.

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

## Acknowledgements

We thank Dr. Jr-Wen Shui at the Institute of Biomedical Sciences, Academia Sinica, for providing technical assistance with organoid culture, *Lgr5-eGFP-creERT2^{+/Tg}; Villin-Cre* mice, and insightful guidance and research consultation. We thank Dr. Kai-Chien Yang at the Department of Pharmacology, National Taiwan University College of Medicine, for the kind gift of *p53^{f/+}* mice, editing assistance, and invaluable consultation on the research. We thank Dr. Wei-Chien Huang at the School of Pharmacy, China Medical University, for his valuable guidance on this study. We thank Dr. Chien-Kuo Lee at the Graduate Institute of Immunology, National Taiwan University College of Medicine, for the kind gift of B6.SJL-*Ptprc^α pepc^β*/BoyJ (CD45.1) mice, and Yu-Ling Hsiao and Wei-Yao Chin for the technical assistance with the bone marrow chimera experiment. We thank Dr. Li-Chung Hsu at the Institute of Molecular Medicine, National Taiwan University College of Medicine, for valuable research consultation and editorial advice. We also thank Dr. Wen-Pin Chen at the Department of Pharmacology, National Taiwan University College of Medicine, for his research consultation. We thank the Transgenic Mouse Model Core Facility of the National Core Facility for Biopharmaceuticals, Ministry of Science and Technology, Taiwan, and the Animal Resources Laboratory of the National Taiwan University Center of Genomic and Precision Medicine for the technical services provided. We thank the staff of the Biomedical Resource Core at the First Core Labs, National Taiwan University College of Medicine, for providing technical assistance. We thank the imaging core at the First Core Labs, National Taiwan University College of Medicine, for providing technical support in image acquisition and analysis. We also acknowledge the service provided by the Flow Cytometric Analysis and Sorting Core of the First Core Laboratory, National Taiwan University College of Medicine. We thank veterinarian Yi-Ting Tsai of the Laboratory Animal Center, National Taiwan University College of Medicine, Taipei, Taiwan, for her technical assistance in the histopathological assessment and interpretation of H&E-stained tissue sections. This work was financially supported by the core consortia of National Taiwan University from the Higher Education Sprout Project by the Ministry of Education (MOE) in Taiwan (107L890402, 108L890402, 109L890402, 110L893201, and 111L892601 to C.C.C.), the National Science and Technology Council (NSTC113-2320-B002-012 and NSTC114-2320-B002-005 to C.C.C.), and Mr. Yua-Yue Tsay's donation to the Ching Kang Foundation for Pharmacy Promotion (to C.C.C.).

## Author contributions

Y.T.L., C.L., and C.C.C. were responsible for the study concept, design, and data interpretation. Y.T.L., C.L., and Y.H.H. conducted the experiments, data acquisition, and analysis. M.S.W. provided materials for human samples and clinical consultation. Y.T.L. and C.C.C. drafted the manuscript. C.C.C. supervised the study and acquired the funding. S.C.M. supervised the experiments of the hematopoietic cell compartment.

## Competing interests

The authors declare no competing interests.
