## [Transparent Peer Review file · Nature Communications]

CBP phosphorylation maintains intestinal homeostasis by supporting the stem cell niche through versican

Corresponding Author: Dr Ching-Chow Chen

Version 0:

Reviewer comments:

Reviewer #1

(Remarks to the Author)

This paper explores the role of CBP phosphorylation at Ser1383/1387 in regulating versican expression and maintaining the intestinal stem cell (ISC) niche. Using CBP knock-in mice with alanine substitutions (CBP^{AA}) at these phosphorylation sites, the study demonstrates that impaired phosphorylation results in ulcerative colitis (UC)-like symptoms, driven by altered CBP binding preferences. Specifically, increased CBP-p53 binding reduces versican levels, a large, extracellular matrix proteoglycan known for its role in cell adhesion, proliferation, migration, tissue remodeling. Reduction in versican disrupts ISC function, promoting colonic inflammation. These findings highlight the critical roles of CBP phosphorylation and versican in maintaining colonic health.

Overall, the study is intriguing and introduces a novel candidate, versican (Vcan), for exploration in ulcerative colitis (UC). However, it lacks critical rigor required for studies of this nature. Key issues include insufficient sample sizes to substantiate translational claims, failure to establish causation (relying primarily on associations), and the absence of a clear mechanism connecting Vcan, CBP, and intestinal epithelial cell (IEC) stem cell dysfunction. These gaps significantly temper the enthusiasm of this reviewer.

MAJOR:

1) The study lacks key rescue and proof-of-causation experiments. Restoring CBP phosphorylation (e.g., by introducing a phospho-mimetic CBP mutant at Ser1383/1387) could clarify whether phosphorylation alone rescues organoid formation and viability. Similarly, cell-based assays would help establish sufficiency. Introducing interventions such as versican supplementation or targeting ECM components could test whether reduced stemness is mediated by extracellular matrix alterations. Additional rescue experiments—such as supplementing CBP^{AA} mice with versican or phospho-mimetic CBP mutants—could confirm whether CBP phosphorylation or its downstream effects, like versican expression, are sufficient to reverse colitis and restore IEC stemness, independent of p53 deletion. Testing in vivo effects of versican replenishment (e.g., via systemic or localized delivery) in CBP^{AA} mice would further strengthen the case for versican as a therapeutic target for intestinal inflammation and stem cell maintenance. reviewer notes that the only reversal was Vcan addition to organoids.

2) the study currently lacks MoA elucidation, i.e., whether impaired CBP phosphorylation alters key inflammatory signaling pathways (e.g., NF- κ B or cytokine profiles) within the colon, contributing to the inflammatory phenotype. Including data on inflammatory cytokine levels or signaling pathways, which could connect CBP phosphorylation status with inflammatory responses. RNAseq could provide MOA guidance here.

3) only looks at a single time point, but it lacks a longitudinal analysis of disease progression or severity. This limits understanding of whether the UC-like phenotype worsens, stabilizes, or fluctuates over time, which is relevant for translational applications and understanding IBD's chronic nature

4) Th authors are pointing towards IEC driven phenotype but again would be nice to see more MOA... It's unclear whether the dysfunction arises from changes in cell-cell adhesion, permeability, or specific signaling pathways within IECs

5) How does impaired CBP phosphorylation affect epithelial cell signaling and communication with other colonic cells, such as fibroblasts or immune cells?

6) n = 3 seems particularly very low and lacks rigor in the patient studies (Fig 9)/ stem cell dysfunction (a key claim) did not get carried into this figure.

MINOR:

1) Abstract: expand CBP, IEC, define genotype [CBPAAp53 IEC]

Reviewer #2

(Remarks to the Author)

This manuscript by Lin and colleagues investigated the significance of the Ser1382/1386 phosphorylation of CREB-binding protein (CBP), a transcriptional coactivator of NF- κ B and p53, in the context of "colitis". The group previously reported that CBP phosphorylation at these sites determine its binding preference for NF- κ B or p53, thereby playing a role in controlling cell proliferation versus apoptosis. In this current study, the team characterized CBP(AA) knockin mice that do not have these serine sites for phosphorylation. They also used an IKK inhibitor that inhibits these phosphorylation events. In Figure 1-7, they show that, in the mouse intestinal epithelial cells, mouse enteroids, and HCT human cell lines, blocking these phosphorylation increased CBP to bind p53, reduced epithelial proliferation, caused a non-immune cell driven "spontaneous colitis" phenotype resembling human ulcerative colitis. Deleting epithelial p53 in CBP(AA) mice reverted the phenotypes. In Figure 8-9, they went on to show a reduced versican expressing compartment in CBP(AA) mice and its restoration by epithelial p53 deletion, and this loss of versican is also found in human UC patients. The data reporting changes in organoid culture and proximity ligation assay showing CBP-p53 interaction are well done. I have some major concerns about the interpretation of their data and the conceptualization of some findings.

Major concerns:

1. The manuscript tried to tie the phenotype to ulcerative colitis. The authors stated that "CBP(AA) mice exhibit a spontaneous colitis phenotype resembling clinical patients with ulcerative colitis (UC)". However, from the provided histology images, for example the very first one shown in Fig. 1f, no immune cell infiltration, glandular distortion, or any sign of epithelial death were seen in the tissue of CBP(AA) mice. The majority of colonic images from CBP(AA) mice provided in the following figure panels also do not strongly support a colitis in CBP(AA) mice, except in Fig. 2e, which was used to show a colon from the WT \rightarrow AA transplantation experiment. There in Fig. 2e (WT \rightarrow AA) does contain some chronic inflammation (basal lymphoplasmacytosis and mild glandular distortion). This raised the question why WT \rightarrow AA transplanted mice showed a stronger phenotype than intact AA mice? Therefore, the mis- or over-interpretation of these histopathology findings by the authors are a major concern. The group may want to include some GI pathology expertise, especially they want to sell the work as an IBD study.

2. The epithelial intrinsic versus versican-dependent mechanisms are confusing. While the first 7 figures established the defects in epithelial cells, which were supported by the epithelial intrinsic characterization of CBP-p53 interactions in enteroid and HCT cell line experiments. The epithelial intrinsic mechanism is also supported by reversion of the phenotype by epithelial deletion of p53. However, Fig. 8-9 switched the mechanism to versican expressing compartment, which is in the ECM or the "mucosal stromal" compartment. It is unclear how deleting p53 in the epithelial cells restored the versican expression in the ECM or the stroma. This versican perspective is especially confusing as the earlier figures indeed showing changed CBP-p53 interactions in the AA epithelial cells or AA enteroids (independent of versican).

3. In terms of to demonstrate that CBP(AA) binds more to p53 to regulate gene expression, Fig. 4 is considered a bit thin, without showing any chromatin level data.

4. Conceptually, Fig. 6 essentially used an inhibitor to show the similar effects observed in the CBP(AA) organoids. These data are cohesive and may be presented together to deliver the concept.

5. Fig. 3, the BrdU pulse chase, migration assays do not show an impressive difference.

Reviewer #3

(Remarks to the Author)

This report builds on previous work demonstrating that phosphorylation of the transcriptional coactivator CBP at Ser1382/1386 by IKK α enhances its interaction with the p65 NF- κ B subunit. Moreover, Ser1382/1386 phosphorylation inhibits the interaction of CBP with the tumour suppressor p53. Conversely, the unphosphorylated CBP preferentially binds p53 over p65. Here, to learn more about the in vivo significance of the regulatory mechanism, the authors have created a knockin mouse model where Ser1382/1386 of CBP are mutated to alanines. Analysis of these mice revealed the spontaneous development of a condition resembling ulcerative colitis (UC). Moreover, the authors convincingly demonstrate that this results from a defect in the renewal of intestinal stem cells rather than immune cells. RNA Seq analysis revealed that this is a consequence, at least in part, from the p53 dependent down regulation of an ECM component called versican in the CBP(AA) mice. In an organoid model, addition of versican rescued the defects seen upon CBP Ser1382/1386 mutation. Demonstrating the potential applicability of these finding to humans, intestinal tissues from UC patients were found to have

reduced level of CBP Ser1382/1386 phosphorylation, a decreased level of p65/CBP interactions, increased levels of CBP/p53 interactions and significantly reduced levels of versican.

Overall, I found the analysis of the UC phenotype of the CBP(AA) mice, together with the identification of versican and its regulation by p53, to be convincing. However, there were other aspects of the manuscript where I have major concerns.

Major Concerns

(1) Figures 5 & 6 rely heavily on an IKK inhibitor called CYL-19S as a means of reducing CBP Ser1382/1386 phosphorylation in cells and determining the effect of this on the interaction between CBP and p53 or p65, together with various phenotypic assays. There are a number of concerns with use of this inhibitor.

(a) CYL-19S is also an inhibitor of IKKbeta. Indeed this group has previously reported effects of this compound on p53 (including induction of p21 seen here) as a result of IKKbeta inhibition (DOI: 10.1111/j.1582-4934.2009.00712.x). Although in their previous report looking at CBP Ser1382/1386 phosphorylation they found that IKKbeta cannot phosphorylate these residues *in vitro*, there are likely to be many other CBP independent mechanisms by which IKKbeta inhibition might influence the effects seen here. Therefore, additional controls are required to support the data in Figure 5 in HCT116 cells, including both the effects on the CBP/p53 interaction, p21 induction and spheroid formation. First, the authors need to use siRNAs targeting IKKalpha or beta to confirm which of these kinases are responsible for the effects seen. Second, they should use a commercial IKKbeta inhibitor that does not inhibit IKKalpha and determine whether it has any effect.

(b) A further control that should be included is to determine whether CYL-19S has any effect on the CBP(AA) organoids. If the effects of CYL-19S are specifically through CBP Ser1382/1386 phosphorylation there should be no effect when these are mutated. Testing the effect of CYL-19S on CBP(AA) organoids that have been supplemented with versican (Figure 8) would be a good way of demonstrating that this compound is not affecting organoid growth through other, non-specific, mechanisms.

(c) In Figure 5A, there still seems to be a significant level of CBP Ser1382/1386 phosphorylation after CYL-19S treatment. This data needs to be quantified. Moreover, confirming the specificity of this signal through the use of CBP, IKKalpha and IKKbeta siRNAs is also required.

(d) I could find very little information on CYL-19S as it does not seem to be commercially available. There is a previous report on the synthesis of this compound from these authors (<https://doi.org/10.1093/carcin/bgh211>) but it is not cited in this current manuscript. Indeed I could not find a description of the source of CYL-19S in the methods. This previous report only performs limited analysis of the specificity of CYL-19S. Therefore, unless published elsewhere, the specificity of CYL-19S needs to be demonstrated using a large panel of recombinant kinases. Many companies offer this type of compound screening as a commercial service.

(2) Is versican also down regulated in the CBP(AA) organoids and in HCT116 cells treated with CYL-19S?

(3) One conclusion from Figure 8 is the CBP Ser1382/1386 mutation only affects a small subset of p53 regulated genes (including versican). One interpretation of this is that these therefore represent genes that are uniquely dependent on CBP as a regulator. Assuming that versican is also downregulated in CBP(AA) organoids and HCT116 cells (see (2) above), this apparent CBP dependence should be demonstrated through the use of a CBP siRNA and a CBP inhibitor.

(4) Does the RNA Seq data reveal any known NF-kB regulated target genes being affected by the CBP Ser1382/1386 mutation? The Venn diagram in Figure 8c suggests that there are 35 genes affected by the CBP Ser1382/1386 mutation in cells with wild type p53 but only 21 are shown in the heat map in Figure 8a. However, only one gene appears to be affected by the CBP (AA) mutation when p53 is mutated, suggesting that effects on NF-kB activity, at least in unstimulated cells, are minimal. In the discussion (lines 340-347) the authors discuss the previously described roles of NF-kB in IECs. They suggest that in patients with UC and lowered CBP Ser1382/1386 phosphorylation together with the CBP(AA) mice that NF-kB would be inert to environmental stimuli. Using the organoid model, can the authors demonstrate this by stimulating the cells with an NF-kB inducer such a TNF and then examining the effects on NF-kB regulated gene expression in cells with the CBP Ser1382/1386 mutation?

(5) Related to (4), I can't find any information on where the RNA Seq data was deposited. It would also be desirable to provide an Excel file of the RNA Seq data for the reviewers and as Supplementary data.

Other concerns

(6) Line 214 states 'Because increased CBP/p53 binding was demonstrated to dampen stemness in HCT116 IECs'. This is not the case. CYL-19S was shown to do this but as discussed above, it cannot currently be concluded that these effects are through CBP/p53 binding.

Reviewer #4

(Remarks to the Author)

In the present study, the authors followed up on their previous study which showed that phosphorylation of coactivator CBP by IKKa promotes NF-kB-dependent transcription while unphosphorylated CBP promotes p53-mediated transcription. Here the authors generated a CBPAA knock-in mouse model with impaired IKKa-dependent phosphorylation of CBP that resulted

in an Ulcerative Colitis (UC) phenotype, including shortened colon length, fecal occult blood, and stool consistency consistent with clinical UC. Histologically, there were infiltration of neutrophils and macrophages and increased CD4+ and C8+ T cells in mesenteric lymph nodes. Bone marrow chimera study suggested that the UC-like phenotype is mediated through Intestinal Epithelial Cells (IECs) which demonstrate decreased proliferation, migration, and organoid formation in CBPAA cells due to increased p53 activity. Predictably, an IKK inhibitor that decreased phosphorylation of CBP in wild-type IECs resulted in increased p53 activity, lower expression of stem cell marker ALDH1A1, and decreased organoid formation of HCT116 IECs, thereby recapitulating the CBPAA phenotype. The authors then deleted p53 from IECs in the CBPAA mouse and the UC phenotype resolved, indicating that these conditions are p53 dependent. RNA-seq showed that versican is down-regulated in CBPAA cells from colonic crypts, but deletion of p53 restores versican expression. In addition, CBPAA organoids cultured in exogenously added versican showed proliferation similar to wild-type counterparts. Tissue from UC patients show increased CPB/p53 binding, decreased CBP/p65 binding, and decreased versican expression.

Overall, the study provides strong evidence to support their major conclusions. However, there are some concerns that need to be addressed to make the study even more compelling.

Points to address:

1. The images presented in Figure 1f are not very clear to an untrained reader. It is unclear what has been scored on the right graph. This needs better explanation.
2. The materials and methods section indicates the use of 8-week old females (line 391) in the study but some of the figure legends indicate the use of 6-8 week old mice. Figure 2 clearly includes the use of male mice as well. The specifics of mice used need to be consistent throughout the presentation.
3. Figure 5: the HCT116 colon tumor cell line is referred to as IEC (intestinal epithelial cells). The data with this cell model does not seem to add much to the overall study which is otherwise focused on mouse IECs and UC. Moreover, referring HCT116 colon tumor cell line as IEC in the text makes the reading confusing because mouse intestinal cells are also referred to as IECs throughout. Consequently, this figure could be moved to supplemental section of the data presentation and these cells should be referred to as HCT116 or simply colon tumor cells to avoid confusion.
4. Figure 5a, 5b and 5g: the differences concluded for these figures are not clear. Quantitative data need to be presented to be convincing.
5. Figure 5c: Data for TNF α +/- CYL-19S also need to be included for comparison.
6. The source of the IKK α inhibitor, CYL-19S, was not described in the manuscript. This needs to be stated in the materials and methods section. It also needs to be stated how specific this inhibitor is toward IKK α . There is no clear information even in the authors' prior publication (Huang, et al., Mol Cell, 2007).
7. The source and nature of the recombinant versican used in Figure 8 was not described. Was this generated from mammalian expression system or from *E. coli*? This needs to be stated. How did the authors come to the specific dose of versican (100 ng/microL)? This needs to be explained.

Version 1:

Reviewer comments:

Reviewer #1

(Remarks to the Author)

The authors have made an effort to answer some but not all the critiques.

MAJOR CONCERNS

Figure Legends and Clarity:

Figure legends throughout the manuscript lack sufficient detail to interpret the data presented. Critical information needed to understand the panels is missing, forcing the reader to repeatedly consult the Methods section. Authors must significantly improve legend clarity and completeness to make their key findings more accessible and self-contained.

Figure 1I & 1K: Quantification and Inconsistencies in CBP Staining:

Panel K quantifies total CBP+ cells, yet images in panel I reveal absence of CBP at the top of the crypts. This may be due to CBP instability in cells lacking phosphorylation; however, such interpretations are speculative without corroborating images or dual-staining for pCBP and total CBP on serial or the same tissue sections. Moreover, it's unclear how quantification was derived—specifically, the method of thresholding used for defining “positive” cells based on integrated density is not described. The chosen Ulcerative Colitis (UC) field does not reflect the same features as in the mouse model, making translational claims tenuous.

Internal Inconsistency Regarding Inflammation:

Panel 1i shows prominent infiltration by neutrophils and macrophages, consistent with inflammation. However, the rebuttal document contradicts this, claiming no upregulation of inflammatory pathways in colonic gene expression. This discrepancy is not resolved and undermines the internal consistency of the manuscript.

Figure 1i & 1j: Overcropping of Panels:

The panels are cropped too tightly, raising concerns about selective presentation. Larger fields with insets should be included to demonstrate reproducibility and allow assessment of generalizability across the tissue.

Figure 3c: PLA Localization of p53-CBP Interaction:

PLA signals indicating interaction between p53 and CBP are predominantly extranuclear, which contradicts the model

proposed in the manuscript. This could reflect poor antibody specificity, background noise, or suboptimal staining protocols. Validation using negative controls (e.g., CBP^{-/-} mice) is strongly advised.

Organoid Immunofluorescence Imaging:

All IF images lack visible lumens, whereas the brightfield images show clear lumen structures. This discrepancy should be explained—are lumens collapsing in fixed samples, or is the imaging method insufficient to capture them?

Figure 4c: MTT Assay Interpretation Is Unclear:

The MTT readout is not intuitive. The Y-axis label lacks context, and the legend does not clarify the nature of the perturbation or the comparison being made. MTT reduction indicates loss of viability, yet no baseline or control is specified. For assessing organoid biogenesis, a growth assay over time, rather than endpoint viability, would be more appropriate.

Mechanistic Gaps in Vcan Role:

While the model points toward stem cell senescence (e.g., increased Gadd45 expression, p53 activation), and bone marrow transplant data suggest immune involvement, it remains unclear which cells produce Vcan and how it impacts the stem cell niche. These mechanistic gaps weaken the central hypothesis.

Figure 7: Rescue Experiments Lack Credibility:

While rescue experiments are commendable in principle, the methodology is problematic. The authors report dramatic phenotype reversal using DD/EE CBP mutants transfected via Lipofectamine into 1–5 cells/organoid. However, such low-efficiency transfection would not plausibly result in 100% rescue of all phenotypes. The discrepancy between representative images and quantification also raises concerns about data integrity.

Figure 8a: Questionable PLA Signal Localization:

As in Figure 3c, PLA signals are largely extranuclear, despite the proposed interaction (p53-CBP) occurring in the nucleus. This inconsistency must be addressed with proper controls.

Response to Reviewers Is Inadequate and Lacking Transparency:

The authors have paraphrased the reviewers' original comments rather than providing the full unedited critiques. This prevents an objective evaluation of whether concerns have been adequately addressed. Additionally, the manuscript contains numerous grammatical errors and inconsistencies between results and rebuttal statements, further undermining confidence in the revision.

MINOR CONCERNS

Language and Grammar:

The manuscript continues to suffer from grammatical errors and awkward phrasing, e.g., in the abstract (page 2, lines 24–27). A thorough language edit by a native English speaker or professional service is necessary.

Figure 2f (left panel, WT → AA):

This image is blurry and low-resolution. A higher-quality version should be provided to ensure proper interpretation.

Reviewer #2

(Remarks to the Author)

In my original point #1, I pointed out that there was no colitis in any of the AA tissue images. The authors responded that a new Fig. 1h was included along with macrophage and neutrophil staining panels in Fig. 1i, as well as TUNEL staining in Fig. 1j. Unfortunately, I am still not convinced there is inflammation in AA mouse colons. First, there is no inflammation in Fig. 1h. Second, the WT and AA colons shown in Fig. 1i looked the same. The Ly6G stained cells do not look like "neutrophils", as neutrophils are smaller than these stained cells. Third, the TUNEL stained cells in AA tissue do not look like apoptotic cells, instead they look quite healthy. Overall, I do not believe there is inflammation or colitis in these AA mouse colons. This was one of the major issues I raised in prior review. The authors may want to consult some GI pathologists.

My #3 point is to suggest more data, hopefully chromatin level results, to support CBP(AA) binds more to p53 to regulate gene expression. Instead of addressing the point using their mouse model, the authors responded by pointing to a study published in 2007.

Thanks for addressing #4.

The last point was about the BrdU pulse chase assay to show migration. The authors responded by adding more animal numbers. However, from the provided images, this reviewer failed to see any difference in terms of migration.

Reviewer #3

(Remarks to the Author)

The authors have satisfactorily addressed by concerns. I have no further comments.

Reviewer #4

(Remarks to the Author)

The authors have sufficiently addressed prior concerns of this reviewer. As such, this reviewer does not have any additional concerns.

Version 2:

Reviewer comments:

Reviewer #2

(Remarks to the Author)

For the original point #1: The authors dampened the claim of inflammatory phenotype in these AA mice, which improved the rigor of their experimental interpretations.

For the point regarding chromatin level evidence of CBP binding and regulation of p53, they showed a new result in panel Fig. 3f to demonstrate more p53 binding to p21 promoter. Although this indirectly showed an increased p53 activity in these AA tissues, the direct regulation of CBP on p53 was not demonstrated in this experiment. The n-number in this new experiment was 3 for each group. The panels of 3a, 3d, and 3e were also showing n=3 in all comparison, which did not seem to support a high confidence of these experimental results.

Overall, the paper has been improved but figure 3 remains to be a weak component of the work.

Version 3:

Reviewer comments:

Reviewer #2

(Remarks to the Author)

The newly revised Fig. 3a showed Ki67 and DAPI staining. The AA image on the right is overall dimmer than the WT image. The DAPI signal is brighter in the WT image than the DAPI signal in the AA image. The figure legend stated "The nuclei were counterstained with DAPI for normalization. Ki-67 expression was quantified as the number integrated density (IntDen) normalized to the DAPI IntDen within a fixed region". If the quantification was based on fluorescent intensity and the two images were clearly taken with different settings, I am concerned whether the authors carefully examined these images before including them in the manuscript.

Point-by-point response

Reviewer #1

Major:

1. *The study lacks key rescue and proof-of-causation experiments.*
 - (a) *Restoring CBP phosphorylation (e.g., by introducing a phospho-mimetic CBP mutant at Ser1383/1387) could clarify whether phosphorylation alone rescues organoid formation and viability.*
 - (b) *Similarly, cell-based assays would help establish sufficiency. Introducing interventions such as versican supplementation or targeting ECM components could test whether reduced stemness is mediated by extracellular matrix alterations.*
 - (c) *Additional rescue experiments—such as supplementing CBP^{AA} mice with versican or phospho-mimetic CBP mutants—could confirm whether CBP phosphorylation or its downstream effects, like versican expression, are sufficient to reverse colitis and restore IEC stemness, independent of p53 deletion. Testing in vivo effects of versican replenishment (e.g., via systemic or localized delivery) in CBP^{AA} mice would further strengthen the case for versican as a therapeutic target for intestinal inflammation and stem cell maintenance. reviewer notes that the only reversal was Vcan addition to organoids.*

Reply: The key rescue and proof-of-causation experiments have been conducted.

- (a) Restoring CBP phosphorylation in CBP^{AA} organoids by introducing a phospho-mimetic CBP DD or EE mutant at Ser1383/1387 have been conducted and demonstrated, also CBP^{AA} organoid formation and viability as well as IECs proliferation and versican are rescued (now Fig.7a-d). These provide key rescue and proof-of-causation that CBP phosphorylation alone maintains versican and stem cell niche to preserve IEC stemness. (p. 9, lines 269-273)
- (b) To establish the sufficiency of extracellular matrix on stemness, *ex vivo* rescue experiments by supplementing recombinant versican into cultured medium of CBP^{AA} organoids were performed. The findings of restoring IECs proliferation and organoid-forming efficiency not only strengthen versican and stem cell niche maintaining IECs stemness but also indicate that reduced stemness is mediated by extracellular matrix alteration (now Fig. 6h,i). Furthermore, gefitinib, an EGFR inhibitor, abrogated these rescues in CBP^{AA} organoids, suggesting that versican may trigger activation of the EGFR pathway (now Fig. 6h,j). (p. 8-9, lines 248-255)
- (c) We thank the reviewer for this insightful comment.

In the present study, we explore the role of a component of the extracellular matrix (ECM), particularly the involvement of versican in maintaining IEC stemness, which paves the way for new treatments for IBD. Our findings not only have valuable clinical implications—offering a novel potential therapeutic strategy for IBD—but also demonstrate how molecular-level insights can be translated into clinically significant applications. CBP phosphorylation is essential for maintaining versican and stem cell niche, thereby preserving IEC stemness and sustain intestinal homeostasis. The *in vivo* supplementation of versican to restore IEC stemness and reverse IBD is another task with great challenge that needs well preparation by consulting with glycan expert. It takes time.

We will contact with Dr. Chi-Huey Wong to arrange a face to face discussion. Dr. Wong is a former President of Academia Sinica in Taiwan and also a world-renowned expert in glycoscience. He is currently the Scripps Family Chair Professor of Chemistry at the Scripps Research Institute and a joint Distinguished Research Fellow at the Genomics Research Center of Academia Sinica.

2. *The study currently lacks MoA elucidation, i.e., whether impaired CBP phosphorylation alters key inflammatory signaling pathways (e.g., NF-κB or cytokine profiles) within the colon, contributing to the inflammatory phenotype. Including data on inflammatory cytokine levels or signaling pathways, which could connect CBP phosphorylation status with inflammatory responses. RNAseq could provide MOA guidance here.*

Reply:

As suggested, we have analyzed RNA-seq data to assess whether impaired CBP phosphorylation in CBP^{AA} mice alters key inflammatory signaling pathways (e.g., NF-κB or cytokine profiles) within the colon. There are no significant changes in the expression of key inflammatory signaling components, such as NF-κB target genes or pro-inflammatory cytokines in CBP^{AA} mice compared with CBP^{WT}, suggesting that the inflammatory phenotype observed in CBP^{AA} mice is not driven by conventional inflammatory signaling pathways. Instead, increased CBP binding to p53 leads to reduced versican that plays a central role as followings:

- Genetic deletion of p53 in IECs of CBP^{AA} mice reversed colonic histopathology, restored barrier integrity (now Fig. 5g,f) and expression of versican *in vivo* and *ex vivo* (now Fig. 6f, k). (p. 8, lines 217-219; p. 9, lines 260-263)

- Introducing phospho-mimetic CBP DD and EE mutants into CBP^{AA} organoids rescued the defective organoid formation and restored versican expression (now Fig. 7b,d). Similarly, supplementation with recombinant versican in CBP^{AA} organoids restored growth and proliferation (now Fig. 6h,i), reinforcing that versican is a functional downstream mediator of CBP phosphorylation to maintain intestinal homeostasis. (p. 9, Lines 271-273; p8-9, lines 248-251)

3. *only looks at a single time point, but it lacks a longitudinal analysis of disease progression or severity. This limits understanding of whether the UC-like phenotype worsens, stabilizes, or fluctuates over time, which is relevant for translational applications and understanding IBD's chronic nature.*

Reply:

In our pilot study, we did assess colitis progression in CBP^{AA} mice from 4 to 56 weeks of age by tracking stool consistency and occult blood (Fig. R1 in this response letter). The colitis severity peaked around 8-14 weeks and then partially resolved during adulthood.

All experiments in the present study were conducted using 8-week-old mice, both male and female CBP^{AA} mice showed comparable severity of colitis.

Fig. R1 Longitudinal assessment of colitis progression in CBP^{AA} mice from 4 to 56 weeks of age. Dynamics of colitis severity in male and female CBP^{AA} mice were shown (a), which were assessed by combination of individual weekly stool occult blood and stool consistency scores (b-h).

4. The authors are pointing towards IEC driven phenotype but again would be nice to see more MOA... It's unclear whether the dysfunction arises from changes in cell-cell adhesion, permeability, or specific signaling pathways within IECs.

Reply:

The mechanisms underlying IEC dysfunction in CBP^{AA} mice like cell-cell adhesion, barrier permeability, *specific signaling pathways* are all examined.

Cell-cell adhesion: Reduced E-cadherin expression in colonic epithelial cells of CBP^{AA} mice compared with CBP^{WT} is revealed and that is restored in $CBP^{AA}p53^{\Delta IEC}$ mice (now Fig. 6g). Similarly, in CBP^{AA} organoids, E-cadherin expression is diminished and rescued by supplementation with recombinant versican as well as overexpressed with phospho-mimetic CBP DD or EE mutants (now Fig. 6j, 7e). These findings suggest that impaired CBP phosphorylation and subsequent suppression of epithelial versican compromise cell-cell adhesion in IECs. (p. 8, lines 245-246; p. 8-9, lines 248-252; p.9, lines 273-274)

Barrier function: Employing FITC-dextran gavage, increased intestinal permeability in CBP^{AA} mice and reversed in $CBP^{AA}p53^{\Delta IEC}$ mice were originally shown (now Fig. 1f, 1g, 5f), supporting an IEC-intrinsic defect. (p. 4, lines 96-98; p. 8, lines 217-219)

Specific signaling pathways: Rescued functions of CBP^{AA} organoids by supplementing with

recombinant versican are shown, the effects of which were abrogated by the treatment with EGFR inhibitor, gefitinib (now Fig.6h-j), suggesting the involvement of EGFR pathway in the CBP-versican signaling axis. (p. 9, lines 253-255; p. 11, lines 349-351)

5. *How does impaired CBP phosphorylation affect epithelial cell signaling and communication with other colonic cells, such as fibroblasts or immune cells?*

Reply:

In the present study, we revealed that impaired CBP phosphorylation in CBP^{AA} mice leads to increased CBP/p53 binding, impaired IEC proliferation and reduced IEC stemness through down-regulating chondroitin sulfate proteoglycan versican and stem cell niche, resulting in a UC-like phenotype. All these findings were restored in CBP^{AA}p53^{ΔIEC} mice (now Figs.3-6).

Versican is a matricellular protein playing a key role not only in epithelial integrity but also in modulating the extracellular matrix composition and cytokine milieu. Its crosstalk with adjacent stromal and immune cells through paracrine signaling in CBP^{WT} mice is possible, like preventing luminal antigen exposure that triggers immune cell infiltration; by contrast, CBP^{AA} mice show increased F4/80⁺ and Ly6G⁺ cells (now Fig. 1i). These highlight that epithelial CBP phosphorylation status can shape the tissue microenvironment not only through stem cell regulation and barrier maintenance, but also by influencing downstream stromal and immune responses. (p. 4, lines 100-102)

6. *n = 3 seems particularly very low and lacks rigor in the patient studies (Fig 9)/stem cell dysfunction (a key claim) did not get carried into this figure.*

Reply:

(1) We have expanded our immunohistochemical analysis using additional colonic biopsy samples to n = 4-7 for normal controls and n = 9-10 for UC patients (now Fig. 8b-d). These expanded data strengthen the statistical power and reinforce our original observations. (p. 31, lines 1036-1038).

(2) Additionally, we performed Ki-67 staining and observed reduced Ki-67⁺ staining in epithelium of UC patients compared to non-IBD controls (now Fig. 8c), consistent with the stem cell dysfunction. (p. 9, lines 281)

(3) To further support the translational relevance of CBP phosphorylation in human, we performed additional IHC staining for IKKα. Consistent with our previous findings demonstrating that IKKα phosphorylates CBP at Ser1382/1386 to regulate cell fate decisions (Mol. Cell 2007;26(1):75-87), we observed reduced IKKα expression in the colonic sections of UC patients (now Fig. 8d), which may account for the decreased CBP phosphorylation detected in these samples. (p. 9, lines 281-282)

In our pilot study, CBP phosphorylation sites in colonic biopsies from UC patients were sequenced and there were no point mutations at the corresponding serine residues (Fig.R2). This suggests that reduced CBP phosphorylation in human UC is due to reduced IKKα expression instead of genetic mutations.

(4) The sample number of UC patients for CBP/p53 proximity ligation assay (PLA) remains at n = 3 (Fig. 8a), since the results are consistent with those in Fig. 8b-d. Furthermore, the quality of these PLA figures were as prominent as those in Figs 3c, 5h and Supplemental Fig. 5b.

MINOR:

- 1) *Abstract: expand CBP, IEC, define genotype [CBP^{AA}p53^{ΔIEC}]*

Reply: We have now expanded CBP (CREB-binding protein), extracellular matrix (ECM) and defined genotype of CBP^{AA}p53^{ΔIEC} mice (CBP^{AA} mice with deletion of p53 in the intestinal epithelium) in the Abstract.

UC p't #1	GGCGTGAGGGGCATCAGAGCCGATTCTTGGACGTGCATTCCAAAAAGCAGACATCCACGCCGTCAATTCCTCAAAGCAAACAGAGCTTTGGTTCGATATGG GAAAGATTGAGACATTTCCCGAGAATCCACAAACC
UC p't #2	GGGCTGAAGGCATCAGAGCGTATTCTTGGACGTGCATTCCAAAAAGCAGACATCCACGCCGTCAATTCCTCAAAGCAAACAGAGCTTTGGTTCGATATGG GAAAGATTGAGACATTTCCCGAGAATCCACAAACC
UC p't #3	CGAGTAGGGCATCAGAGCCGATTCTTGGACGTGCATTCCAAAAAGCAGACATCCACGCCGTCAATTCCTCAAAGCAAACAGAGCTTTGGTTCGATATGG GAAAGATTGAGACATTTCCCGAGAATCCACAAACC
UC p't #4	GGGGTTAAGGGCATCAGAGCCGATTCTTGGACGTGCATTCCAAAAAGCAGACATCCACGCCGTCAATTCCTCAAAGCAAACAGAGCTTTGGTTCGATATGG GAAAGATTGAGACATTTCCCGAGAATCCACAAACC
UC p't #5	GGTGGGGGCATCAGAGCGTATTCTTGGACGTGCATTCCAAAAAGCAGACATCCACGCCGTCAATTCCTCAAAGCAAACAGAGCTTTGGTTCGATATGG AAGATTGAGACATTTCCCGAGAATCCACAAACC
UC p't #6	GGGATAGGGGCATCAGAGCGTATTCTTGGACGTGCATTCCAAAAAGCAGACATCCACGCCGTCAATTCCTCAAAGCAAACAGAGCTTTGGTTCGATATGG GAAAGATTGAGACATTTCCCGAGAATCCACAAACC
UC p't #7	GCGGGGGGGGCATCAGAGCCGATTCTTGGACGTGCATTCCAAAAAGCAGACATCCACGCCGTCAATTCCTCAAAGCAAACAGAGCTTTGGTTCGATATGG GAAAGATTGAGACATTTCCCGAGAATCCACAAACC
UC p't #8	GGTTAAGGGCATCAGAGCGTATTCTTGGACGTGCATTCCAAAAAGCAGACATCCACGCCGTCAATTCCTCAAAGCAAACAGAGCTTTGGTTCGATATGG AAGATTGAGACATTTCCCGAGAATCCACAAACC
UC p't #9	GGGGTTAGGGGCATCAGAGCCGATTCTTGGACGTGCATTCCAAAAAGCAGACATCCACGCCGTCAATTCCTCAAAGCAAACAGAGCTTTGGTTCGATATGG GAAAGATTGAGACATTTCCCGAGAATCCACAAACC
UC p't #10	GGGCTAGGGCATCAGAGCGTATTCTTGGACGTGCATTCCAAAAAGCAGACATCCACGCCGTCAATTCCTCAAAGCAAACAGAGCTTTGGTTCGATATGG AAGATTGAGACATTTCCCGAGAATCCACAAACC
UC p't #11	GGCTGAGGGGCATCAGAGCGTATTCTTGGACGTGCATTCCAAAAAGCAGACATCCACGCCGTCAATTCCTCAAAGCAAACAGAGCTTTGGTTCGATATGG GAAAGATTGAGACATTTCCCGAGAATCCACAAACC
UC p't #12	GGGGTAAGGGCATCAGAGCCGATTCTTGGACGTGCATTCCAAAAAGCAGACATCCACGCCGTCAATTCCTCAAAGCAAACAGAGCTTTGGTTCGATATGG GAAAGATTGAGACATTTCCCGAGAATCCACAAACC

Ser1382 Ser1386

GGTTTGTGGATTCTGGGGAAATGTCTGAATCTTTC

The reverse complement is:

GAAAGATTGAGACATTTCCCGAGAATCCACAAACC

Fig. R2 Blood sample sequencing of UC patients. Genomic DNA was extracted from peripheral blood samples of ulcerative colitis (UC) patients. The obtained sequence was aligned to the human reference genome (NCBI RefSeq: NG_009873.1) and showed 100% identity, indicating the absence of point mutations.

Reviewer #2

Major concerns:

1. *The manuscript tried to tie the phenotype to ulcerative colitis.*

(a) *The authors stated that CBP(AA) mice exhibit a spontaneous colitis phenotype resembling clinical patients with ulcerative colitis (UC). However, from the provided histology images, for example the very first one shown in Fig. 1f, no immune cell infiltration, glandular distortion, or any sign of epithelial death were seen in the tissue of CBP(AA) mice.*

Reply:

To address this concern, we have replaced a higher-quality representative image that more clearly illustrates the phenotype in Fig. 1 (now Fig. 1h, AA). Additionally, we have included a TUNEL assay to provide a quantitative evidence of epithelial cell apoptosis (now Fig. 1j). Increased infiltration of F4/80⁺ macrophages and Ly6G⁺ neutrophils in CBP^{AA} colons were shown (now Fig. 1i). The observed increase in immune cell infiltration along with the increased epithelial cell death support the UC-like phenotype in CBP^{AA} mice. (p. 4, lines 100-102)

The scoring criteria in histological assessment have been expanded in the revised Methods section under "Histology". (p.15, lines 473-481)

(b) *The majority of colonic images from CBP(AA) mice provided in the following figure panels also do not strongly support a colitis in CBP(AA) mice, except in Fig. 2e, which was used to show a colon from the WT→AA transplantation experiment. There in Fig. 2e (WT→AA) does contain some chronic inflammation (basal lymphoplasmacytosis and mild glandular distortion). This raised the question why WT→AA transplanted mice showed a stronger phenotype than intact AA mice?*

Reply:

Prior to chimera experiments, the recipient mice were irradiated (Fig. 2a), which induced γ -irradiation-mediated intestinal injury. These explain why AA transplanted mice showed a stronger phenotype than intact AA mice, because the combination of intestinal injury in intact CBP^{AA} mice plus γ -irradiation occurs in the WT→AA group (now Fig. 2f). (p. 16-17, lines 528-533)

2. *The epithelial intrinsic versus versican-dependent mechanisms are confusing. While the first 7 figures established the defects in epithelial cells, which were supported by the epithelial intrinsic characterization of CBP-p53 interactions in enteroid and HCT cell line experiments. The epithelial intrinsic mechanism is also supported by reversion of the phenotype by epithelial deletion of p53. However, Fig. 8-9 switched the mechanism to versican expressing compartment, which is in the ECM or the "mucosal stromal" compartment. It is unclear how deleting p53 in the epithelial cells restored the versican expression in the ECM or the stroma.*

Reply: The epithelial intrinsic versus versican-dependent mechanisms are clarified.

Epithelial origin of versican was examined by employing *ex vivo* 3D organoids. Consistent with the expression pattern observed in colon tissues (Fig. 6f), versican is also downregulated in CBP^{AA}p53-floxed organoids and restored in CBP^{AA}p53^{ΔIEC} organoids with IEC-specific p53 deletion (Fig. 6k), confirming its epithelial origin. Furthermore, replenishing recombinant versican protein into the culture medium of CBP^{AA}p53-floxed organoids (AA + Vcan) restored the organoid-forming efficiency and size, as well as the number of Ki-67⁺ cells to levels comparable to those in CBP^{WT}p53-floxed organoids (Fig. 6h, i). These suggest that epithelial-derived versican is released into the ECM or the "mucosal stromal" to regulate ISCs. (p. 9, lines 259-263; p. 11, lines 333-336)

3. *In terms of to demonstrate that CBP(AA) binds more to p53 to regulate gene expression, Fig. 4 is considered a bit thin, without showing any chromatin level data.*

Reply: In our Mol. Cell paper (2007;26:75-87), we have demonstrated that phosphorylation of CBP at Ser1382/1386 by IKK α enhances NF- κ B, while suppressing p53-mediated gene expression, thereby promoting cell growth. However, mutation of serines to alanines, which impairs CBP phosphorylation, increases its recruitment to p53- instead of NF- κ B-responsive promoters, like p21. In the present study, we show that our previous mechanistic findings at the molecular level convey translational significance in CBP^{AA} mice and UC patients.

4. *Conceptually, Fig. 6 essentially used an inhibitor to show the similar effects observed in the CBP(AA) organoids. These data are cohesive and may be presented together to deliver the concept.*

Reply: Original Fig. 3c-e is now Fig. 4.

Since we have generated *ex vivo* CBP^{WT} and CBP^{AA} organoids to deliver our concept (now Fig. 4), it is not necessary to treat CBP^{WT} organoids with inhibitor (CYL-19s) that mimic CBP^{AA} status like HCT116 did. Therefore, the original Fig. 6 is now deleted.

5. *Fig. 3, the BrdU pulse chase, migration assays do not show an impressive difference.*

Reply:

Increasing the n number up to 8 in CBP^{WT} and 6 in CBP^{AA} mice at 96 h is further conducted to show an impressive difference (Fig. 3b). (p. 29, lines 941-944).

Reviewer #3

Major Concerns

1. *In Figures 5 & 6, there are a number of concerns with use of an IKK inhibitor, CYL-19S as a means of reducing CBP Ser1382/1386 phosphorylation in cells and determining the effect of this on the interaction between CBP and p53 or p65, together with various phenotypic assays.*

(a) *CYL-19S is also an inhibitor of IKKbeta reported by authors (DOI: 10.1111/j.1582-4934.2009.00712.x). Although this group previously found that IKKbeta cannot phosphorylate CBP Ser1382/1386 in vitro, there are likely to be many other CBP independent mechanisms by which IKKbeta inhibition might influence the effects seen here. Therefore, additional controls are required to support the data in Figure 5 in HCT116 cells, including both the effects on the CBP/p53 interaction, p21 induction and spheroid formation.*

First, the authors need to use siRNAs targeting IKKalpha or beta to confirm which of these kinases are responsible for the effects seen. Second, they should use a commercial IKKbeta inhibitor that does not inhibit IKKalpha and determine whether it has any effect.

(b) *A further control that should be included is to determine whether CYL-19S has any effect on the CBP(AA) organoids. If the effects of CYL-19S are specifically through CBP Ser1382/1386 phosphorylation there should be no effect when these are mutated. Testing the effect of CYL-19S on CBP(AA) organoids that have been supplemented with versican (Figure 6) would be a good way of demonstrating that this compound is not affecting organoid growth through other, non-specific, mechanisms.*

(c) *In Figure 5A, there still seems to be a significant level of CBP Ser1382/1386 phosphorylation after CYL-19S treatment. This data needs to be quantified. Moreover, confirming the specificity of this signal through the use of CBP, IKKalpha and IKKbeta siRNAs is also required.*

(d) *I could find very little information on CYL-19S as it does not seem to be commercially available. There is a previous report on the synthesis of this compound from these authors (<https://doi.org/10.1093/carcin/bgh211>) but it is not cited in this current manuscript. Indeed I could not find a description of the source of CYL-19S in the methods. This previous report only performs limited analysis of the specificity of CYL-19S. Therefore, unless published elsewhere, the specificity of CYL-19S needs to be demonstrated using a large panel of recombinant kinases. Many companies offer this type of compound screening as a commercial service.*

Reply (a) to (d): CYL-19s synthesized by our collaborator, Prof. Cherng-Chyi Tzeng, a medicinal chemist is now cited as reference 29 and its non-specific mechanisms have been excluded.

(1) We have now included the proper citation of our previous work describing synthesis and initial characterization of CYL-19s as Ref. 29 (Carcinogenesis. 2004;25(10):1925-34), and this information has also been added to the Methods section. (p.13, 418-419)

(2) As suggested, we have tested the effect of CYL-19s on CBP^{AA} organoids that have been supplemented with versican and demonstrated that this compound is not affecting organoid growth and Ki-67⁺ cells (Fig. R3), confirming that CYL-19S specifically acts through reducing CBP Ser1383/1387 phosphorylation without non-specific mechanisms.

Since we have generated *ex vivo* primary CBP^{WT} and CBP^{AA} organoids (now Fig. 4), in line with the suggestion of Reviewer #4, HCT116 cells treated with CYL-19s that mimic CBP^{AA} status (originally Figure 5) is now moved to Supplementary Figure 5.

Fig. R3 CBP^{AA} organoids are not affected by CYL-19s treatment. Organoids were collected on day 6. The experimental groups included: CBP^{WT}p53^{f/f} (WT), CBP^{AA}p53^{f/f} (AA), CBP^{AA}p53^{f/f} organoids treated with 100 ng/μL versican from day 0 (AA + Vcan), and CBP^{AA}p53^{f/f} organoids treated with 100 ng/μL versican from day 0 and 5 μM CYL-19s added on day 3 (AA + Vcan + CYL-19s). **a** Microscopy images of day 6 cultured organoids (upper panel, 10X; lower panel, 4X). Scale bar, 100 μm. **b** Representative immunofluorescence staining of Ki-67 in colon sections. Nuclei were counterstained with DAPI. Scale bar, 100 μm. Image shown is a reused panel from Fig. 6h and 6i of the manuscript, provided here to facilitate direct comparison with the additional data.

2. Is versican also down regulated in the CBP(AA) organoids and in HCT116 cells treated with CYL-19S?

Reply:

- (1) Versican is also downregulated in CBP^{AA} organoids compared with CBP^{WT} organoids (now Fig. 6k).
- (2) HCT116 cells treated with CYL-19S also show downregulation of versican as well as Ki-67⁺ cells and E-cadherin expression (right Fig.R4), consistent with the results of CBP^{AA} organoids (Fig.6i-k).

Fig. R4 CYL-19s downregulated Ki-67, versican, and E-cadherin expression in HCT116 cells. HCT116 cells were treated with 10 μM CYL-19s for 24 h. Representative immunofluorescence images of Ki-67, versican and E-cadherin expression. Nuclei were counterstained with DAPI (blue). Scale bars, 20 μm.

3. One conclusion from Figure 8 is the CBP Ser1382/1386 mutation only affects a small subset of p53 regulated genes (including versican). One interpretation of this is that these therefore represent genes that are uniquely dependent on CBP as a regulator. Assuming that versican is also downregulated in CBP(AA) organoids and HCT116 cells (see (2) above), this apparent CBP dependence should be demonstrated through the use of a CBP siRNA and a CBP inhibitor.

Reply:

We employed CBP^{AA} organoids overexpressing phospho-mimetic CBP DD or EE mutant at Ser1383/1387 to restore CBP phosphorylation in CBP^{AA} organoid and organoids formation and versican expression are rescued, clarifying the unique dependence of CBP phosphorylation in regulating downstream versican expression (Fig. 7), as reply to Reviewer #1 (comment 1a). (p. 9, lines 269-273)

4. Does the RNA Seq data reveal any known NF-κB regulated target genes being affected by the CBP Ser1382/1386 mutation? The Venn diagram in Figure 8c suggests that there are 35 genes affected

by the CBP Ser1382/1386 mutation in cells with wild type p53 but only 21 are shown in the heat map in Figure 8a. However, only one gene appears to be affected by the CBP (AA) mutation when p53 is mutated, suggesting that effects on NF- κ B activity, at least in unstimulated cells, are minimal.

In the discussion (lines 340-347) the authors discuss the previously described roles of NF- κ B in IECs. They suggest that in patients with UC and lowered CBP Ser1382/1386 phosphorylation together with the CBP(AA) mice that NF- κ B would be inert to environmental stimuli. Using the organoid model, can the authors demonstrate this by stimulating the cells with an NF- κ B inducer such a TNF and then examining the effects on NF- κ B regulated gene expression in cells with the CBP Ser1382/1386 mutation?

Reply:

To address whether CBP Ser1382/1386 phosphorylation influences NF- κ B signaling, CBP^{WT} and CBP^{AA} organoids were respectively stimulated with TNF- α and abundant CBP/p65 binding was only observed in CBP^{WT} organoids (Supplementary Fig. 8c, lower left panel), indicating the activation of NF- κ B pathway. In contrast, CBP/p65 binding was not seen in CBP^{AA} organoids (Supplementary Fig. 8c, lower right panel), indicating that CBP^{AA} organoids do not respond to pro-inflammatory cues. These findings suggest that impaired phosphorylation of CBP renders IECs inert to NF- κ B-mediated transcriptional activation upon environmental stimulation. (p. 11-12, lines 356-362)

Furthermore, RNA-seq analysis did not reveal significant differences between CBP^{WT} and CBP^{AA} mice in canonical NF- κ B target gene expression under basal conditions, consistent with the idea that impaired phosphorylation of CBP affects inducibility rather than baseline NF- κ B activity. Therefore, reduced CBP phosphorylation dampens stimulus-induced NF- κ B activation, providing a potential explanation for inert response in CBP^{AA} mice to environmental stimuli and possibly in UC patients harboring low CBP phosphorylation.

The discrepancy in gene numbers between the Venn diagram (original Fig. 8c) and the heat map (original Fig. 8a) was due to employing inconsistent threshold criteria during the initial analysis. To address this, we have reanalyzed the RNA-seq data with consistent threshold parameters: $|\log_2FC| > 0.5$ and $p_{adj} < 0.05$. The revised data show the correct number of 13 genes in both Venn diagram and heat map (now Fig. 6a, c), and the final gene set is consistent with the applied threshold. (p. 30, lines 988, 991).

5. *Related to (4), I can't find any information on where the RNA Seq data was deposited. It would also be desirable to provide an Excel file of the RNA Seq data for the reviewers and as Supplementary data.*

Reply: The RNA sequencing data reported in this study have been deposited in the Gene Expression Omnibus (GEO) database under accession codes: **GSE295556** (<https://www.ncbi.nlm.nih.gov/geo/query/acc.cgi?acc=GSE295556>). The following secure token has been created to allow review while the record remains private: **qjuveqwkhyjbyd**.

6. *Line 214 states "Because increased CBP/p53 binding was demonstrated to dampen stemness in HCT116 IECs". This is not the case. CYL-19S was shown to do this but as discussed above, it cannot currently be concluded that these effects are through CBP/p53 binding.*

Reply:

"Because increased CBP/p53 binding was demonstrated to dampen stemness in HCT116 IECs" is now revised into "Since increased CBP/p53 binding suppressed sphere formation and reduced stemness markers in HCT116 cells". (p. 7, lines 195-196)

Reviewer #4

Points to address:

1. *The images presented in Figure 1f are not very clear to an untrained reader. It is unclear what has been scored on the right graph. This needs better explanation.*

Reply:

We have replaced a higher-quality representative image that more clearly illustrates the phenotype in Fig. 1 (now Fig. 1h, AA). Additionally, we have included a TUNEL assay to provide a quantitative evidence of epithelial cell apoptosis (Fig. 1j). (p. 4, lines 100-102)

The scoring criteria in histological assessment have been expanded in the revised Methods section under "Histology". (p.15, lines 473-481)

2. *The materials and methods section indicates the use of 8-week-old females (line 391) in the study but some of the figure legends indicate the use of 6–8-week-old mice. Figure 2 clearly includes the use of male mice as well. The specifics of mice used need to be consistent throughout the presentation.*

Reply:

All experiments in the present study employed female mice except the bone marrow chimera experiments shown in Figure 2. Dr. Chien-Kuo Lee (Graduate Institute of Immunology, College of Medicine, National Taiwan University) kindly provided us congenic CD45.1 (B6.SJL-Ptprca pepc β /BoyJ) mice in both sexes. Due to the limited availability of CD45.1 female mice, male mice were also included. We have revised the legend of Figure 2 to explicitly state the use of both sexes. The results shown in the new Figure 2 combine both male and female mice. To further address the concern about sex-related issues, we present separate data in Supplementary Figure 2. All readouts, including stool scores, colon length, histological damage, and immune cell infiltration are consistent between male and female mice, supporting that the colitis phenotype is not sex-dependent. (p.28, lines 923-927)

3. *Figure 5: the HCT116 colon tumor cell line is referred to as IEC (intestinal epithelial cells). The data with this cell model does not seem to add much to the overall study which is otherwise focused on mouse IECs and UC. Moreover, referring HCT116 colon tumor cell line as IEC in the text makes the reading confusing because mouse intestinal cells are also referred to as IECs throughout. Consequently, this figure could be moved to supplemental section of the data presentation and these cells should be referred to as HCT116 or simply colon tumor cells to avoid confusion.*

Reply:

As suggested, Figure 5 regarding HCT116 colon tumor cell line is now moved to supplemental Fig. 5. Also, we now refer this cell line as HCT116 instead of IEC to avoid confusing.

4. *Figure 5a, 5b and 5g: the differences concluded for these figures are not clear. Quantitative data need to be presented to be convincing.*
5. *Figure 5c: Data for TNF α +/- CYL-19S also need to be included for comparison.*

Reply (4-5):

As suggested, we have moved CYL-19s-related HCT116 cell data (original Fig. 5) to Supplementary Fig. 5. Since we have already generated CBP^{AA}p53 ^{Δ IEC} mice and performed *ex vivo* organoid assays to get core findings in this manuscript, we believe it is not essential to further quantify HCT116 data.

6. *The source of the IKK α inhibitor, CYL-19S, was not described in the manuscript. This needs to be stated in the materials and methods section. It also needs to be stated how specific this inhibitor*

is toward IKK α . There is no clear information even in the authors' prior publication (Huang, et al., *Mol Cell*, 2007).

Reply:

As stated in the reply to Reviewer #3, point 1, we have now included the proper citation of our previous work describing synthesis and initial characterization of CYL-19s as Ref. 29 (*Carcinogenesis*. 2004 Oct;25(10):1925-34), and this information has also been added to the Methods section. (p.13, 418-419)

The size and number of organoids formation and Ki-67⁺ IECs in CBP^{AA} organoids supplemented with versican are not affected by CYL-19s (Fig. R3), confirming that CYL-19s specifically acts through reducing CBP Ser1383/1387 phosphorylation without non-specific mechanisms.

Our prior work (*Mol Cell*. 2007 Apr 13;26(1):75-87) demonstrated that CYL-19s effectively abolished TNF- α -induced CBP phosphorylation at Ser1382/1386 in HeLa cells (mentioned in p.6, lines 175-176).

7. *The source and nature of the recombinant versican used in Figure 8 was not described. Was this generated from mammalian expression system or from E. coli? This needs to be stated. How did the authors come to the specific dose of versican (100 ng/microL)? This needs to be explained.*

Reply:

The recombinant mouse versican protein (Abxexa, abx069663) used in original Fig. 8 (now Fig. 6) was produced using a prokaryotic (*E. coli*) expression system and now clarified in the Methods section. (p.13, lines 422-423)

To determine the optimal concentration, we first follow a recent study that employed 100 ng/mL for in vitro treatment (*Circulation*. 2024;149:1004–1015; now reference 52). Based on this, we performed a preliminary dose-response test from 50, 100, to 200 ng/mL. The concentration of 50 ng/mL already elicited a detectable biological response, and 100 ng/mL induced a more robust effect, and no further enhancement was observed at 200 ng/mL. Therefore, we selected 100 ng/mL as the optimal working concentration for the experiments presented. (p.13-14. Lines 423-426)

Point-by-point response

Response to Reviewer #1:

MAJOR CONCERNS (1-11)

1. *Figure Legends and Clarity: Figure legends throughout the manuscript lack sufficient detail to interpret the data presented. Critical information needed to understand the panels is missing, forcing the reader to repeatedly consult the Methods section. Authors must significantly improve legend clarity and completeness to make their key findings more accessible and self-contained.*

Reply 1: We thank Reviewer's important suggestions and apologize for not providing sufficient details in the prior figure legends.

In the current revised form, sufficient details highlighted in red have been added into all the figure legends to interpret the data presented throughout the manuscript. We have significantly improved the legend clarity and completeness to make our key findings more accessible and self-contained. Thank you.

2. *Figure 1l & 1k: Quantification and Inconsistencies in CBP Staining: Panel k quantifies total CBP+ cells, yet images in panel l reveal absence of CBP at the top of the crypts. This may be due to CBP instability in cells lacking phosphorylation; however, such interpretations are speculative without corroborating images or dual-staining for pCBP and total CBP on serial or the same tissue sections. Moreover, it's unclear how quantification was derived—specifically, the method of thresholding used for defining “positive” cells based on integrated density is not described. The chosen Ulcerative Colitis (UC) field does not reflect the same features as in the mouse model, making translational claims tenuous.*

Reply 2: We sincerely thank the Reviewer's valuable comments to point out the issues regarding CBP staining, quantification, and translational relevance.

- (1) To address the concern about absence of CBP at the top of the crypts in CBP^{AA} mouse colons (original panel 1k not panel 1l), we have now employed frozen sections of CBP^{WT} and CBP^{AA}

colons to perform immunofluorescence (IF) staining instead of IHC by formalin-fixed paraffin-embedded (FFPE) (now Fig. 1i). CBP stainings are prominent in both genotypes (now Fig. 1i, lower panels), and pCBP staining is still detected in CBP^{WT} colons but not CBP^{AA} colons (now Fig. 1i, upper panels), similar to our original IHC results (original Fig. 1k, upper panels). These results indicate that CBP is not destabilized in cells lacking phosphorylation.

IF staining can better preserve antigenicity compared to IHC, where suboptimal antigen retrieval may result in uneven staining. As our custom-made anti-pCBP antibody and the commercial anti-CBP antibody are both rabbit-derived, dual staining is not feasible due to potential cross-reactivity.

- (2) Regarding quantification, our original analysis was performed using StrataQuest software with fixed ROIs. In this revised version, we have re-quantified the data using ImageJ with fixed ROIs and integrated density thresholding. Detailed descriptions are now added to the legends of Fig. 1i and 1j (p.31, lines 966-973).
- (3) We acknowledge the Reviewer's concern regarding the UC field selection. We have carefully reviewed our data and noted that the same feature of reduced pCBP expression is detected in both human UC colon tissues and CBP^{AA} colons without changes in total CBP level. This finding,

together with the similar results of increased CBP–p53 binding, reduced versican expression, and fewer Ki-67⁺ proliferative IECs in both colons (Fig. 8a–c; Fig.3c, 5h, 5i, 6f), collectively support the translational relevance of the current study. We are grateful to the Reviewer for prompting us to clarify and better substantiate this point.

- 3. Internal Inconsistency Regarding Inflammation:** Panel 1i shows prominent infiltration by neutrophils and macrophages, consistent with inflammation. However, the rebuttal document contradicts this, claiming no upregulation of inflammatory pathways in colonic gene expression. This discrepancy is not resolved and undermines the internal consistency of the manuscript.

Reply 3:

We sincerely thank the Reviewer for highlighting this important point regarding potential internal inconsistency between histological observations and gene expression data.

- (1) In response to this concern along with the related comments from the Reviewer #2, we have consulted two gastrointestinal pathologists—one veterinary pathologist at our Laboratory Animal Center and one attending physician specializing in gastrointestinal pathology at the Department of Pathology, NTU Hospital. Consistent with the Reviewer #2's assessment, both experts confirmed that CBP^{AA} mice under basal conditions has no colonic inflammation. To avoid overinterpretation, we have therefore described the lesions in CBP^{AA} mice as colonic abnormalities throughout the manuscript (p.4, lines 101-106), and concomitantly removed two original Fig. 1i and 1j from prior revised form shown in the right which presented F4/80⁺ and Ly6G⁺, and TUNEL assay.

- (2) In bone marrow chimera experiments (Fig. 2), chimeric CBP^{AA} mice exhibit substantial immune cell infiltration in all layers of the bowel wall, severe crypt distortion with the loss of entire crypts, and moderate mucosal hyperplasia, accompanied by increased histological scores and elevated mesenteric lymph node (mLN) cellularity (Fig. 2f, g, h). These mice also showed greater infiltration of macrophages (F4/80⁺) and neutrophils (Ly6G⁺) than chimeric WT mice did (AA→WT) (Fig. 2i, j), in which minimal levels of infiltration markers were observed, likely because of irradiation prior to the chimera experiments (p. 5, lines 124-131). These results indicate that chimeric CBP^{AA} mice exhibit colonic inflammation driven by nonhematopoietic cells, such as epithelial cells.
- (3) Regarding RNA-seq analysis, we would like to clarify that these data were generated from isolated colonic crypts enriching epithelial cells without inclusion of the lamina propria, which contains the primary source of inflammatory cells. Therefore, the lack of upregulation in inflammatory pathways in this dataset is expected. We have revised the text to make this methodological detail more explicit (p.8, lines 243-244).

We are grateful to the Reviewer for prompting us to carefully reassess the presentation of our data and ensure the internal consistency between histology and colonic gene expression.

- 4. Figure 1i & 1j: Overcropping of Panels:** The panels are cropped too tightly, raising concerns about selective presentation. Larger fields with insets should be included to demonstrate reproducibility and allow assessment of generalizability across the tissue.

Reply 4: We sincerely thank the Reviewer's helpful suggestion regarding image cropping and the importance of showing larger fields with insets to assess reproducibility and generalizability.

As stated in Reply 3, we have removed the original Fig. 1i and 1j (F4/80⁺, Ly6G⁺, and TUNEL panels) to

avoid overinterpretation and retained F4/80⁺ and Ly6G⁺ stainings in bone marrow chimera experiments by larger fields with insets (now Fig. 2i and 2j). The lower panels in these two figures clearly show greater infiltration of macrophages (F4/80⁺) and neutrophils (Ly6G⁺) in chimeric CBP^{AA} mice compared to those in chimeric WT mice, thereby demonstrating the reproducibility and generalizability of our findings across the examined tissues.

5. Figure 3c: PLA Localization of p53-CBP Interaction: PLA signals indicating interaction between p53 and CBP are predominantly extranuclear, which contradicts the model proposed in the manuscript. This could reflect poor antibody specificity, background noise, or suboptimal staining protocols. Validation using negative controls (e.g., CBP^{-/-} mice) is strongly advised.

Reply 5: We sincerely thank the Reviewer for raising this important point regarding the localization and specificity of PLA signals for p53–CBP interactions.

As suggested, we have validated the assay using multiple negative controls, including CBP^{AA} mouse tissues, UC patient samples, and HCT116 cells. As shown in Fig. R1, no PLA signals were detected in any of these negative controls, supporting the specificity of the PLA signals shown in Fig. 3c and Fig. 8a. In these figures, the majority of PLA signals are observed within the nuclei, with some located immediately adjacent to these signals. In UC patient tissues (Fig. 8a, right), some PLA signals are also observed at more distant extranuclear locations.

We note that the mouse colonic sections in Fig. 3c were cryopreserved, whereas those in UC patients in Fig. 8a were formalin-fixed paraffin-embedded (FFPE) because of the lack of frozen human tissues. Therefore, more distant extranuclear background in UC patient tissues may be attributed to technical limitations of FFPE processing, because a similar background signal is also present in the adjacent normal tissues (Fig. 8a, left).

Importantly, enhanced CBP–p53 PLA signals were specifically in CBP^{AA} mice and UC patient tissues, but not the corresponding negative controls, supporting the conclusion that CBP–p53 interactions are increased in these pathological contexts. We have taken care to present these data clearly, accompanied by appropriate validation, as they represent an important part of our mechanistic findings.

6. Organoid Immunofluorescence Imaging: All IF images lack visible lumens, whereas the brightfield images show clear lumen structures. This discrepancy should be explained—are lumens collapsing in fixed samples, or is the imaging method insufficient to capture them?

Reply 6: We thank the Reviewer for pointing out this important observation regarding the appearance of lumens in our organoid immunofluorescence images.

As described in *Current Protocols in Mouse Biology* (2013 Dec 19; 3(4):217–240), we followed a whole-

mount staining procedure, and images were acquired by confocal Z-stacking to generate maximum intensity projections. This approach optimizes the visualization of fluorescence signals but sometimes may obscure three-dimensional architecture including lumens.

To address this, we have now included the representative single optical sections along with full Z-stack reconstructions adapted from the AA+Vcan panel in Fig.6q, in which the lumen is clearly visible. These additions clarify that lumen structures are preserved in fixed samples.

Fig. R2 Representative optical sections and a Z-stack projection of an organoid stained by immunofluorescence for versican adapted from Fig.6q AA+Vcan. Left panels: Twelve individual optical sections (1–12) acquired by confocal microscopy from a whole-mount-stained organoid, showing localization of versican (green) and nuclei (blue). Lumen structures are indicated by arrows. Right panel: Corresponding Z-stack maximum intensity projection clearly reveals the central lumen.

7. *Figure 4c: MTT Assay Interpretation Is Unclear: The MTT readout is not intuitive. The Y-axis label lacks context, and the legend does not clarify the nature of the perturbation or the comparison being made. MTT reduction indicates loss of viability, yet no baseline or control is specified. For assessing organoid biogenesis, a growth assay over time, rather than endpoint viability, would be more appropriate.*

Reply 7: We sincerely thank the Reviewer for the helpful comments regarding the interpretation and presentation of the MTT readout. We apologize for the lack of clarity in the original Fig. 4c and have revised it according to the Reviewer’s guidance to improve the presentation.

The MTT assay was performed following the established method described in Cell Death & Disease (2014, vol. 5, e1228). In the current revised manuscript, we have clarified the Y-axis label to indicate cell viability normalized to CBP^{WT} organoids (Methods, p. 20, lines 641-642; Fig. 4c legend, p.32-33, lines 1026-1027). Consistent with Ki-67 staining (Fig. 4b), the MTT assay also revealed reduced proliferative capacity in CBP^{AA} organoids compared to those in CBP^{WT} organoids. These two independent assays corroboratively indicate impaired proliferation of CBP^{AA} organoids. We also measured organoid formation efficiency, which reflects the initiation of new organoids, and organoid diameter, which reflects subsequent growth. Both measurements indicate the evaluation of organoid biogenesis (*Nat Commun.* 2023 Sep 27;14(1):6016.) (Fig. 4d; p.7, lines 215–216).

8. *Mechanistic Gaps in Vcan Role: While the model points toward stem cell senescence (e.g., increased Gadd45 expression, p53 activation), and bone marrow transplant data suggest immune involvement, it remains unclear which cells produce Vcan and how it impacts the stem cell niche. These mechanistic gaps weaken the central hypothesis.*

Reply 8: We sincerely thank the Reviewer for the thoughtful comments on the mechanistic aspects of versican (Vcan) production and its impact on the stem cell niche.

- (1) We appreciate the Reviewer’s insights regarding stem cell senescence markers such as increased Gadd45 expression and p53 activation. Gadd45 was beyond the scope of the current study. The central hypothesis of our study is that impaired CBP phosphorylation enhances CBP–p53 interaction, thereby repressing versican (Vcan) expression.
- (2) Our bone marrow chimera experiments, together with evaluation of the differentiation potential of naïve T cells and T_H subsets, support the conclusion that the colonic abnormalities (originally

described as colitis) in CBP^{AA} mice are attributed to impaired CBP phosphorylation in IECs *per se*, rather than to the regulation of baseline immune homeostasis (p.5, lines 129-131 and 133-134; p. 5-6, lines 145-147).

- (3) We provide multiple complementary lines of evidence to demonstrate that versican (Vcan) is produced by intestinal epithelial cells (IECs) and regulated by epithelial CBP-p53 binding:
In addition to the RNA-seq and qPCR results (Fig. 6a, d, e), the epithelial-intrinsic defect and reduced versican expression observed in CBP^{AA} mice were reversed by epithelial-specific p53 deletion (Fig. 6f), suggesting that versican is derived primarily from epithelial cells. Consistent with these tissue-level changes (Fig. 6f), versican expression was also reduced in organoids derived from the crypts of CBP^{AA}p53-floxed mice and restored in those derived from CBP^{AA}p53^{ΔIEC} mice (Fig. 6q, r), further confirming its epithelial origin (p.9, lines 272-277).
- (4) The number of Lgr5⁺ stem cells in the IECs did not differ between Lgr5-CBP^{AA} mice and Lgr5-CBP^{WT} mice (Supplementary Fig. 6). However, intestinal stem cell function was compromised in CBP^{AA} mice, as evidenced by the reduced organoid formation, decreased number of Ki-67⁺ IECs, and reduced versican expression. The unchanged number of Lgr5⁺ stem cells suggests that the observed impairment in stemness may be attributed to a disruption in the stem cell niche, likely caused by alteration in the extracellular matrix (ECM), rather than a direct loss of the stem cell population. Currently, epithelial-derived versican, an ECM, may contribute to the local regulation of the stem cell niche, as replenishing versican into the culture medium of CBP^{AA} organoids restored organoid formation, the organoid diameter and the number of Ki-67⁺ cells to levels comparable to those of CBP^{WT} organoids (Fig. 6j–p; p. 9, lines 263-267 and Discussion, p. 11, lines 351-360), supporting the role of versican in modulating stem cell niche environment to promote stem cell function and regulating intestinal epithelial cell behavior.

The loss of CBP–p53 binding in CBP^{AA}p53^{ΔIEC} mice restored versican expression and IEC stemness to prevent colonic abnormalities. In contrast, impaired CBP phosphorylation in CBP^{AA} mice increases CBP–p53 binding, leading to the downregulation of versican expression and contributing to colonic abnormalities (Figs. 5–6). An analysis of the versican gene promoter and enhancer regions using the PROMO and EPD databases did not reveal classical p53-binding sites. The precise mechanism by which CBP–p53 binding negatively regulates versican expression in IECs seems to be indirect and remains to be investigated.

9. *Figure 7: Rescue Experiments Lack Credibility: While rescue experiments are commendable in principle, the methodology is problematic. The authors report dramatic phenotype reversal using DD/EE CBP mutants transfected via Lipofectamine into 1–5 cells/organoid. However, such low-efficiency transfection would not plausibly result in 100% rescue of all phenotypes. The discrepancy between representative images and quantification also raises concerns about data integrity.*

Reply 9: We thank the Reviewer for raising these important points regarding the rescue experiments and data interpretation.

- (1) We performed gain-of-function experiments using phosphomimetic CBP mutants to further determine whether CBP phosphorylation affects versican levels. Single cells dissociated from crypt-derived CBP^{AA}p53^{f/f} (AA) organoids were transfected with either a control vector expressing EGFP alone (pCMV-puro-IRES2-EGFP), or constructs encoding HA-tagged CBP phospho-mimetic mutants (DD or EE; HA-tagged and co-expressing EGFP) using Lipofectamine™ Stem Transfection Reagent. Transfected cells were identified by EGFP fluorescence. Successful delivery of HA-tagged CBP mutants into AA organoids was confirmed by HA-CBP expression detected by Western blotting (Supplementary Fig. 5). Compared with the vector control, AA

organoids overexpressing DD or EE showed restored versican expression and a reversal of AA phenotype, as evidenced by the rescued organoid-forming efficiency and organoid diameter (Fig. 7a–b). Additionally, these organoids exhibited increased IEC proliferation and E-cadherin expression (Fig. 7c–d). These findings indicate that CBP phosphorylation plays important roles in maintaining versican expression and supporting the intestinal stem cell niche, thereby contributing to the preservation of epithelial homeostasis in the colon. Despite as few as 1-5 EGFP⁺ cells per AA organoid, overexpression of DD or EE upregulated versican expression in AA organoids (Fig. 7a, e). This finding parallels the concept that localized alterations within the stem cell niche can have broad effects on epithelial structure and function. We have now added detailed descriptions of the experimental design into the main text and the legend of Fig. 7 to ensure clarity for readers (p.9-10, lines 282-299 and p.34-35, lines 1098-1104).

- (2) Regarding the discrepancy between representative images and quantification, we apologize for any misunderstanding caused by this important issue.

In addition to the updated representative images (Fig.7b, left, upper panels), low-magnification (4x) images were included to provide a broader view of organoids (Fig.7b, left, lower panels), which clearly show the rescue of organoid formation after DD/EE transfection. Quantification was performed by analyzing all organoids in the entire field of view, including organoids of varying sizes to ensure unbiased quantification. The quantitative data reveal a significant difference between the vector and DD or EE group regarding relative organoid diameter and formation efficiency (Fig.7b, right).

- 10.** *Figure 8a: Questionable PLA Signal Localization: As in Figure 3c, PLA signals are largely extranuclear, despite the proposed interaction (p53-CBP) occurring in the nucleus. This inconsistency must be addressed with proper controls.*

Reply 10: We appreciate the Reviewer's continued attention to the localization of PLA signals. As Reply 5, we have thoroughly addressed this concern by including appropriate negative controls to demonstrate the specificity of the PLA signals. These negative controls support our conclusion that the enhanced CBP–p53 interaction occurs predominantly in the nucleus, with some extranuclear signals in UC patients (Fig.8a) potentially due to technical limitations as previously discussed.

- 11.** *Response to Reviewers Is Inadequate and Lacking Transparency: The authors have paraphrased the reviewers' original comments rather than providing the full unedited critiques. This prevents an objective evaluation of whether concerns have been adequately addressed. Additionally, the manuscript contains numerous grammatical errors and inconsistencies between results and rebuttal statements, further undermining confidence in the revision.*

Reply 11: We thank the Reviewer for the feedback regarding our prior response to Reviewer's comments raised on Dec. 31, 2024. In those responses, only Major Comment 1 was subdivided into 1a, 1b, and 1c to improve clarity, as those remarks were interrelated. All other comments were addressed in original form in our point-by-point response on April 30, 2025 revision.

In the current revision, we have included the full, unedited text of the Reviewer's comments to ensure transparency and to demonstrate that all concerns have been adequately addressed. Additionally, we have carefully re-checked the manuscript for grammatical accuracy and ensured consistency between all results and rebuttal statements, and have sent it to Springer Nature Editing Service for professional English language editing.

MINOR CONCERNS (12-13)

12. *Language and Grammar: The manuscript continues to suffer from grammatical errors and awkward phrasing, e.g., in the abstract (page 2, lines 24–27). A thorough language edit by a native English speaker or professional service is necessary.*

Reply 12: We thank the Reviewer for highlighting the importance of clear language and grammar. The manuscript has been already undergone professional English language editing by Springer Nature Editing Service, and the certificate was issued on August 20, 2025 and may be verified on the SNAS website using the verification code E4F9-7033-AEC4-0C87-D582 as below.

13. *Figure 2f (left panel, WT→AA): This image is blurry and low-resolution. A higher-quality version should be provided to ensure proper interpretation.*

Reply 13: The images in Fig. 2f now have been replaced with a higher-resolution version, re-acquired by a board-certified veterinary pathologist at our Animal Center, College of Medicine, National Taiwan University, to ensure optimal image clarity and correct the histopathological interpretation.

Response to Reviewer #1's prior major comment 6 dated Dec. 31, 2024 following our very first submission on October 1st, 2024

6. *n = 3 seems particularly very low and lacks rigor in the patient studies (Fig 9).*

Reply: We thank the Reviewer for the important comment regarding sample size in the patient studies. We have now increased the number of UC patient samples to n = 7 for CBP-p53 proximity ligation assay (PLA) (now shown in Fig. 8a; original Fig.9a in the first submission). Because of a temporary shortage of reagents from the distributor, Sigma Aldrich, earlier this year, we were unable to expand the sample number in the initial revision on April 30, 2025. We appreciate the Reviewer's understanding and believe this expanded dataset strengthens the robustness of our findings.

Response to Reviewer #2:

1. *In my original point #1, I pointed out that there was no colitis in any of the AA tissue images. The authors responded that a new Fig. 1h was included along with macrophage and neutrophil staining panels in Fig. 1i, as well as TUNEL staining in Fig. 1j. Unfortunately, I am still not convinced there is inflammation in AA mouse colons. First, there is no inflammation in Fig. 1h. Second, the WT and AA colons shown in Fig. 1i looked the same. The Ly6G stained cells do not look like "neutrophils", as neutrophils are smaller than these stained cells. Third, the TUNEL stained cells in AA tissue do not look like apoptotic cells, instead they look quite healthy. Overall, I do not believe there is inflammation or colitis in these AA mouse colons. This was one of the major issues I raised in prior review. The authors may want to consult some GI pathologists.*

Reply 1: We sincerely thank the Reviewer for the detailed and professional feedback.

- (1) As suggested, we consulted Yi-Ting Tsai, a veterinary pathologist at Laboratory Animal Center, College of Medicine, NTU, and Dr. Jia-Huei Tsai, a gastrointestinal pathologist at the Department of Pathology, National Taiwan University Hospital. Both experts concur with the Reviewer #2's assessment that the histological evaluation of CBP^{AA} mice revealed mild pathological changes in the distal colon, including localized low-grade immune cell infiltration in the mucosa and submucosa as well as mild crypt architectural irregularities (Fig. 1h, right panels and Supplementary Fig. 1e). Since these changes do not meet the criteria for colitis, we therefore described these observed lesions as colonic abnormalities (p. 4, lines 101-106), and concomitantly removed two original Fig. 1i and 1j presenting F4/80+ and Ly6G+, and TUNEL assay from prior revised form.
- (2) In bone marrow chimera experiments (Fig. 2), chimeric CBP^{AA} mice exhibit substantial immune cell infiltration in all layers of the bowel wall, severe crypt distortion with the loss of entire crypts, and moderate mucosal hyperplasia, accompanied by increased histological scores and elevated mesenteric lymph node (mLN) cellularity (Fig. 2f, g, h). These mice also showed greater infiltration of macrophages (F4/80⁺) and neutrophils (Ly6G⁺) than chimeric WT mice did (AA→WT) (Fig. 2i, j), in which minimal levels of infiltration markers were observed, likely because of irradiation prior to the chimera experiments (p. 5, lines 124-131). These results indicate that chimeric CBP^{AA} mice exhibit colonic inflammation driven by nonhematopoietic cells, such as epithelial cells.

We appreciate the Reviewer's comments, which have helped us improve the clarity and rigor of our study.

2. *My #3 point is to suggest more data, hopefully chromatin level results, to support CBP(AA) binds more to p53 to regulate gene expression. Instead of addressing the point using their mouse model, the authors responded by pointing to a study published in 2007.*

Reply 2: We thank the Reviewer for the valuable suggestion regarding additional chromatin-level evidence to support enhanced CBP(AA) binding to p53. We have now completed the chromatin level experiments and further confirmed that CBP^{AA} preferentially binds to p53 to regulate gene expression, and the results showed increased binding of p53 to the p21 promoter (now Fig. 3f; p. 6, lines 171-173). During the prior revision, the Cut&Run assays were still being optimized, therefore chromatin-level data were not available yet. We appreciate the Reviewer's understanding and believe the new data further strengthen the mechanistic insights of our study.

3. *Thanks for addressing #4.*

Reply 3: We sincerely thank the Reviewer for the appreciation of our *ex vivo* CBP^{WT} and CBP^{AA} organoid models, which support and deliver the key concepts of our study, thus the figures of CBP^{WT} organoids treated with inhibitor CYL-19s as employed in HCT116 cells that effectively mimic CBP^{AA} status were deleted. We appreciate the Reviewer's constructive comments.

4. *The last point was about the BrdU pulse chase assay to show migration. The authors responded by adding more animal numbers. However, from the provided images, this reviewer failed to see any difference in terms of migration.*

Reply 4: We thank the Reviewer for the helpful comment. To better illustrate the migration trend, we have updated Fig. 3b with higher-resolution BrdU images, including enlarged insets and yellow dashed lines to indicate the direction of cell migration. Hope these improvements clarify the migration differences.

Point-by-point response

Response to reviewer #2:

1. *For the original point #1: The authors dampened the claim of inflammatory phenotype in these AA mice, which improved the rigor of their experimental interpretations.*

Reply 1:

We sincerely appreciate the Reviewer's professional comments, which guided us to temper the claim of an inflammatory phenotype in CBP^{AA} mice and improved the clarity and rigor of our study. We also gained valuable insights from these suggestions.

2. *For the point regarding chromatin level evidence of CBP binding and regulation of p53, they showed a new result in panel Fig. 3f to demonstrate more p53 binding to p21 promoter. Although this indirectly showed an increased p53 activity in these AA tissues, the direct regulation of CBP on p53 was not demonstrated in this experiment. The n-number in this new experiment was 3 for each group. The panels of 3a, 3d, and 3e were also showing n=3 in all comparison, which did not seem to support a high confidence of these experimental results. Overall, the paper has been improved but figure 3 remains to be a weak component of the work.*

Reply 2: We sincerely thank the Reviewer for the constructive comments.

- (a) In addition to p53, we also use anti-CBP antibody to perform CUT&RUN assay. The results reveal that both CBP- and p53-precipitated chromatin are enriched at the p21 promoter (Fig. 3f, g), supporting that enhanced recruitment of CBP by p53 regulates p21 expression in CBP^{AA} mice (p.6, lines 171–173). However, CBP enrichment at the p21 promoter is absent in CBP^{AA}p53^{ΔIEC} mice, accompanying a reduction in p21 expression (Supplementary Fig. 5), in which CBP–p53 binding is lost (Fig. 5h, right two panels). These results confirm that direct regulation of CBP on p53 is enhanced at the chromatin level by impaired CBP phosphorylation in CBP^{AA} mice, providing further evidence that differential phosphorylation of CBP regulates binding to the p53 locus and subsequent p21 expression (p.8, lines 240–246). We believe these new data strengthen the mechanistic insights of our study. We have now increased the n-number of CBP^{AA} mice to 5 for each group to support a high confidence of the results of chromatin level experiments (Fig.3f, g and Supplementary Fig. 5a).
- (b) Also, we have now increased the sample size in Fig. 3a to 8 for all comparisons. In Fig. 3d and Fig. 3e, sample sizes were 5 for CBP^{WT} and 6 for CBP^{AA}, thereby strengthening the confidence in these results. In Fig. 3a, we replaced IHC with Ki-67 immunofluorescence (IF) staining on frozen colon sections from CBP^{WT} and CBP^{AA} mice, which provides much clearer and higher-resolution signals.

Point-by-point response

Response to reviewer #2:

1. *The newly revised Fig. 3a showed Ki67 and DAPI staining. The AA image on the right is overall dimmer than the WT image. The DAPI signal is brighter in the WT image than the DAPI signal in the AA image. The figure legend stated "The nuclei were counterstained with DAPI for normalization. Ki-67 expression was quantified as the number integrated density (IntDen) normalized to the DAPI IntDen within a fixed region". If the quantification was based on fluorescent intensity and the two images were clearly taken with different settings, I am concerned whether the authors carefully examined these images before including them in the manuscript.*

Reply 1:

We sincerely thank the reviewer for the careful and insightful assessment, and we apologize for the confusion caused by the original selection of representative images. We have addressed this concern as follows:

1. **Updated Representative Images:** We appreciate the reviewer's careful attention to the image presentation in Fig.3a. To address this concern, we re-examined our complete dataset from the entire cohort (n = 8 mice per group; Fig. R1). We retained the original WT representative image (WT-03) and replaced the AA image (AA-08) with AA-07. We believe that this updated representative pair (WT-03 and AA-07; highlighted in the red box) offers a more comparable DAPI intensity profile, thereby ensuring a fair and balanced visual comparison between the WT and AA groups (Fig. R2).
2. **Clarification of Imaging Settings and Data Consistency:** To address the concerns regarding acquisition settings for Fig. 3a, we have now provided a comprehensive summary of imaging parameters (Laser Power and Detector Gain) for the entire cohort in Table R1. To maintain consistent visualization of Ki-67 staining across the manuscript, the Ki-67 signal in Fig. 3a has been displayed using a green pseudo-color to match Fig. 5i. Specifically, Fig. 3a was detected using Alexa Fluor 594 (red fluorescence, 561 nm channel), but the signal was pseudo-colored green to ensure consistency with Fig. 5i, which was acquired using Alexa Fluor 488 (green fluorescence).

After consultation with imaging specialists at the First Core Labs, National Taiwan University College of Medicine, variations in DAPI intensity are commonly attributable to unavoidable technical factors such as tissue section thickness.

As shown in Table R1, the Detector Gain for each channel was individually optimized to ensure appropriate dynamic range and to prevent signal saturation. Importantly, even in cases where AA samples required higher Detector Gain (up to 700 V), the Ki-67 signal remained consistently weaker than that in WT controls. This strongly supports the conclusion that the observed reduction reflects a genuine biological phenomenon rather than an imaging artifact.

Furthermore, the reduction of Ki-67 in the AA group was consistently observed in our earlier IHC staining (Fig. 3a in NCOMMS-24-62813B-Z). We expanded the cohort size (n = 8) when transitioning to immunofluorescence staining to ensure consistency with the experimental conditions in Fig. 5i (NCOMMS-24-62813C).

3. Data Accessibility and Transparency: To ensure full transparency, all raw confocal image files (.czi) corresponding to Fig. 3a, along with all other imaging data contributing to quantification in this study, have been deposited in the EBI BioImage Archive. Additionally, all processed figures and the detailed parameter summary table are available via Figshare.

- **EBI BioImage Archive (Accession: S-BIAD2810):**

<https://www.ebi.ac.uk/biostudies/bioimages/studies/S-BIAD2810?key=6d2c7865-f2a6-42c3-9667-c29da769d403>

- **Figshare (Table R1 & Source data):** <https://figshare.com/s/94654a07dcdca6cb64e>

Fig. R1 Full dataset of Ki-67/DAPI staining for Figure 3a. Top panels: Wild-type (WT) group (n = 8 mice). Bottom panels: AA group (n = 8 mice). The blue box indicates the original AA representative image (AA-08). The red boxes highlight the updated representative pair selected for the revised manuscript: WT-03 (which remains unchanged from the original version due to its high structural integrity) and AA-07 (the newly selected image to provide a more balanced DAPI intensity for visual comparison). Scale bar = 100 μ m. All raw confocal image files are available in the EBI BioImage Archive (S-BIAD2810).

Fig. R2 Original and revised representative images for Figure 3a.

Table R1. Individual acquisition parameters for all replicates in Figure 3a.

Sample ID	Group	Ki-67		DAPI	
		Attenuation: Laser 561nm	Detector Gain	Attenuation: Laser 405nm	Detector Gain
WT-01	WT	0.6	650	0.75	540
WT-02	WT	0.6	650	0.75	550
WT-03	WT	0.6	650	0.75	550
WT-04	WT	0.55	660	0.7	550
WT-05	WT	0.55	660	0.7	550
WT-06	WT	0.5	650	0.75	520
WT-07	WT	0.5	650	0.75	520
WT-08	WT	0.5	650	0.75	520
AA-01	AA	0.6	650	0.7	550
AA-02	AA	0.65	630	0.7	550
AA-03	AA	0.65	600	0.7	550
AA-04	AA	0.65	600	0.75	540
AA-05	AA	0.65	700	0.75	530
AA-06	AA	0.65	700	0.75	530
AA-07	AA	0.65	700	0.75	505
AA-08	AA	0.6	700	0.75	530